# Expressivity-Preserving GNN Simulation

**Fabian Jogl**
Machine Learning Research Unit
Center for Artificial Intelligence and Machine Learning
TU Wien, Vienna, Austria
`fabian.jogl@tuwien.ac.at`

**Maximilian Thiessen**
Machine Learning Research Unit
TU Wien, Vienna, Austria
`maximilian.thiessen@tuwien.ac.at`

**Thomas Gärtner**
Machine Learning Research Unit
TU Wien, Vienna, Austria
`thomas.gaertner@tuwien.ac.at`

## Abstract

We systematically investigate *graph transformations* that enable standard message passing to *simulate* state-of-the-art graph neural networks (GNNs) without loss of expressivity. Using these, many state-of-the-art GNNs can be implemented with message passing operations from standard libraries, eliminating many sources of implementation issues and allowing for better code optimization. We distinguish between weak and strong simulation: weak simulation achieves the same expressivity only after several message passing steps while strong simulation achieves this after every message passing step. Our contribution leads to a direct way to translate common operations of non-standard GNNs to graph transformations that allow for strong or weak simulation. Our empirical evaluation shows competitive predictive performance of message passing on transformed graphs for various molecular benchmark datasets, in several cases surpassing the original GNNs.

## 1 Introduction

We systematically investigate which variants of non-standard message passing graph neural networks can be implemented with standard message passing graph neural networks (MPNNs) on a transformed graph without losing expressivity. While MPNNs are a very popular type of graph neural networks (GNNs), they are not able to represent every function on the space of graphs [Xu et al., 2019, Morris et al., 2019] due to their limited *expressivity*: for all MPNNs, there are pairs of non-isomorphic graphs which always have the same embedding. A common approach to create more expressive GNNs is to change the message passing function of MPNNs. If a GNN is more expressive than MPNNs by adapting the message passing function, we call this *non-standard message passing*. Examples of this are message passing variants that operate on subgraphs [Frasca et al., 2022, Bevilacqua et al., 2021] or tuples of nodes [Morris et al., 2019, 2020, 2022]. It is known that some variants of non-standard message passing can be implemented with MPNNs on a transformed graph without losing expressivity as demonstrated by Morris et al. [2019, 2020, 2022] and Qian et al. [2022]. We formalize this as an MPNN *simulating* a GNN: a GNN can be simulated if there exists a graph transformation such that combining an MPNN with that graph transformation is at least as expressive as the GNN. As simulation only requires graph transformations as pre-processing, it allows the use of off-the-shelf MPNN implementations which simplifies and speeds up the implementation of GNNs. Another advantage is that simulation is programming framework agnostic: only the graphs needs to be transformed—independently of how the MPNN is implemented. Finally, simulation allows to easily exchange GNNs in existing workflows without requiring changes to the model. Despite

37th Conference on Neural Information Processing Systems (NeurIPS 2023).

the benefits of simulation, this approach has not been thoroughly investigated: there is currently no formal definition of simulation and it is not known which GNNs can be simulated. In this paper, we define simulation and provide sufficient criteria for non-standard message passing to be simulated.

**Related Work.** Xu et al. [2019] and Morris et al. [2019] proved that MPNNs have limited expressivity. This lead to the development of new GNNs that have higher expressivity than MPNNs. Such GNNs often operate on structures that differ from graphs, for example (1) Morris et al. [2019, 2020, 2022] proposed GNNs operating on $k$-tuples of nodes, (2) Bodnar et al. [2021a,b] proposed GNNs on topological structures, and (3) Bevilacqua et al. [2021] proposed GNNs operating on subgraphs. The idea of implementing non-standard message passing by standard message passing dates back to at least Otto [1997] who showed that instead of performing the higher-order message passing of $k$-WL it is possible to use 1-WL (classical message passing) on a transformed structure [Grohe et al., 2021]. To the best of our knowledge, the first GNN that has been implemented through a graph transformation is $k$-GNN [Morris et al., 2019], which is a generalization from $k$-WL to GNNs. Already Morris et al. [2019] refer to this concept as *simulation*. In follow up work, Morris et al. [2020] and Morris et al. [2022] implemented similar GNNs with non-standard message passing as MPNNs together with graph transformations. Similarly, $k$-OSWL [Qian et al., 2022] was designed to utilize WL. However, none of these papers have formalized the idea of simulation for GNNs (or WL) explicitly. To this end, we [Jogl et al., 2022a,b] showed that CW Networks [Bodnar et al., 2021a], DS [Bevilacqua et al., 2021], DSS [Bevilacqua et al., 2021], and $\delta$-$k$-GNNs/WL [Morris et al., 2020] can be implemented as an MPNN on a transformed graph. A similar high-level idea was proposed in a positional paper by Veličković [2022]. In parallel, Hajij et al. [2023] introduced combinatorial complexes that generalize structures such as graphs or cell complexes and showed that computations on these complexes can be realized as message passing on graphs. While simulation has been used and advocated in the past there is no unified solution to obtaining graph transformations, theoretical justification, or thorough empirical evaluation. We fill this gap and find many GNNs can be simulated by MPNNs on transformed graphs. More details can be found in Appendix A and Appendix B.

**Our Approach.** We introduce *simulation* of non-standard message passing, which formalizes the idea of implementing a GNN, which uses non-standard message passing, through an MPNN on a transformed graph. A message passing algorithm can be *strongly simulated* if an MPNN together with a graph transformation can achieve the same expressivity in *every* iteration of message passing. To prove that many GNNs can be strongly simulated, we define the class of *augmented message passing* (AMP) algorithms which contains many common GNNs. We prove that all AMP algorithms can be strongly simulated and present a meta algorithm to generate the necessary graph transformations. This allows us to show that eleven recently proposed GNNs can be strongly simulated. Specifically, the GNNs by Morris et al. [2019, 2020, 2022] and Qian et al. [2022] perform AMP. This gives an additional mathematical justification to their approach by proving that their GNNs can be implemented with MPNNs without losing expressivity over an implementation with non-standard message passing.

Furthermore, we investigate three constructions that demonstrate the limits of strong simulation: time dependent neighborhoods, nested aggregations, and non-pairwise message passing. We prove that these constructions either cannot be strongly simulated efficiently or cannot be strongly simulated at all. However, if we are only interested in the MPNN achieving the same expressivity as the GNN with non-standard message passing after a sufficient number of layers, it is possible to implement all three constructions with an MPNN. We call this *weak simulation*: a message passing algorithm can be weakly simulated if there exists a non-negative integer $\zeta$ such that an MPNN together with a graph transformation can achieve the same expressivity as the message passing algorithm in one iteration after every $\zeta$ iterations of message passing. We show that Message Passing Simplicial Networks [Bodnar et al., 2021b], CW Networks [Bodnar et al., 2021a], DSS [Bevilacqua et al., 2021], and $K$-hop message passing [Feng et al., 2022] can be weakly simulated. Finally, we evaluate a representative set of graph transformation empirically. Our graph transformations lead to competitive performance compared to the simulated algorithms and often lead to more accurate predictions.

**Main Contributions.** We introduce the concept of strong and weak simulation (Section 3) of message passing. This generalizes existing ideas behind the implementation of several GNNs [Morris et al., 2019, 2020, 2022, Qian et al., 2022]. We provide an automated way of proving that a GNN can be simulated and deriving the necessary graph transformations. We prove that there exist architectures that cannot be strongly simulated (Section 4) but only weakly simulated (Section 5). Our empirical evaluation (Section 6) demonstrates that simulation achieves competitive performance.

## 2 Background

A *graph* $G = (V, E, F)$ is a triple $G = (V, E, F)$ consisting of a set of vertices $V$, a set of directed edges $E \subseteq \{(x, y) \mid x, y \in V, x \neq y\}$, and a function $F$ that assigns a feature to every vertex and edge. For a set of objects $U$ and integer $l > 0$, we denote by $2^U = \{X \mid X \subseteq U\}$ the powerset of $U$, i.e., the set of all subsets of $U$, and by $U^l = \{(u_1, \dots, u_l) \mid u_1, \dots, u_l \in U\}$ the set of all tuples of length $l$ built from $U$. In this paper we work on a generalization of graphs we call *relational structures* that use a generalized notion of *neighborhood*. For a set of objects $U$, we call any function $\mathcal{N} : U \to 2^{(U^\ell)}$ a *neighborhood function*. A neighborhood function assigns to every object a set of tuples of length $\ell$ which we call its neighbors. We use $\ell(\mathcal{N})$ to denote the length of these tuples. We call any tuple $(w, u)$ a *directed edge* where $u \in U$ and $w \in \mathcal{N}(u)$ (the *in-neighbours*) with $\ell(\mathcal{N}) = 1$. For an integer $k > 0$, we use $[\![k]\!]$ to denote the set $\{1, \dots, k\}$ and $\{\!\{\cdot\}\!\}$ to denote a multiset.

**Definition 2.1.** (Relational Structure) Let $k \geq 0$ be an integer, $U$ be a set of objects, and let $\mathcal{N}_1, \dots, \mathcal{N}_k$ be neighborhood functions over $U$. Furthermore, let $F$ be a function such that $F(u)$ assigns a feature to every $u \in U$ and $F^i((x, u))$ assigns a feature to every directed edge with $i \in [\![k]\!]$ and $x \in \mathcal{N}_i(u)$ with $\ell(\mathcal{N}_i) = 1$. Then, the tuple $X = (U, \mathcal{N}_1, \dots \mathcal{N}_k, F)$ is a *relational structure*.

If it is clear from context from which neighborhood $\mathcal{N}_i$ an edge $(x, u)$ is, we will simply write its features as $F((x, u))$ instead of $F^i((x, u))$. If the number of neighborhoods is important we call such a structure a $k$-relational structure. Note, that a graph $G = (V, E, F)$ is a special case of a 1-relational structure with only one neighborhood $\mathcal{N}_G(v) = \{w \mid (w, v) \in E_G\}$, i.e., $G = (V, \mathcal{N}_G, F)$. For graphs $G, H$ we refer to their vertices, edges, and features by adding the graph as a subscript. We say that $G, H$ are *isomorphic* if there exists an edge and feature preserving bijective function $\alpha : V_G \to V_H$ between their vertices: for every vertex $v \in V_G$ it holds that $F_G(v) = F_H(\alpha(v))$ and for every $x, y \in V_G$, it holds that $(x, y) \in E_G$ if and only if $(\alpha(x), \alpha(y)) \in E_H$ with $F_G((x, y)) = F_H((\alpha(x), \alpha(y)))$. No polynomial time algorithm is known which

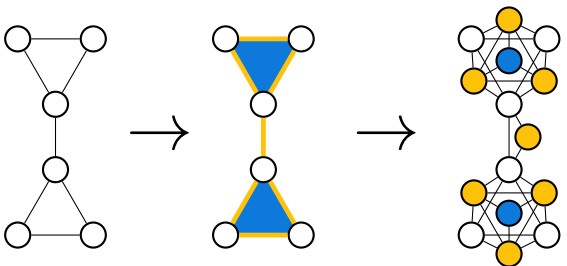

Figure 1: Left: graph. Center: regular cell complex built from the graph through a graph-to-structure encoding [Bodnar et al., 2021a]. Vertices correspond to 0-dimensional cells, edges to 1-dimensional cells ( yellow ) and induced cycles to 2-dimensional cells ( blue ). Right: a graph created by structure-to-graph encoding the regular cell complex to a graph. Vertices corresponds to cells as indicated by color.

decides whether two graphs are isomorphic [Babai, 2016]. However, heuristics such as those based on *colorings* and *color update functions* allow us to distinguish many pairs of non-isomorphic graphs. Furthermore, they allow us to model the ability of GNNs to distinguish graphs [Xu et al., 2019, Morris et al., 2019]. For a relational structure over a set of objects $U$, a *coloring* is a function $c : U \to \chi$ where $\chi$ is a known set of colors. Suppose we have two disjoint sets $U$ and $U'$ with two colorings $c$ over $U$ and $c'$ over $U'$. Then, we define the joint coloring $c \cup c'$ for every $x \in (U \cup U')$ as $c(x)$ if $x \in U$ and $c'(x)$ otherwise. For two colorings $c$ and $d$ defined over the same set of objects $U$, we say that $c$ *refines* $d$ if for every pair of objects $u, r \in U$ it holds that $c_u = c_r \Rightarrow d_u = d_r$. Let $U, U'$ be sets of objects with $U' \subseteq U$. Let $c$ be a coloring of $U$ and $d$ a coloring of $U'$. Then, $c$ refines $d$ if for every pair of objects $u, r \in U'$ it holds that $c_u = c_r \Rightarrow d_u = d_r$.

**Definition 2.2.** (Color Update Function) Let $k \geq 1$ be an integer and $\mathcal{X}$ a set of $k$-relational structures. A color update function takes any $X = (U, \mathcal{N}_1, \dots, \mathcal{N}_k, F) \in \mathcal{X}$, a coloring $c$ of $U$, and a $u \in U$ and outputs a new color $\phi(u, X, c)$ of $u$. We denote the new coloring of the whole set $U$ as $\phi(X, c)$.

We assume color update functions $\phi$ to be computable and denote the time to compute $\phi$ with respect to an input $U$ as $\tau_\phi(|U|)$. Color update functions are used to iteratively update an initial coloring $c^0$ such that $c^{t+1} = \phi(X, c^t)$ for every $t \geq 0$. We denote $t$ applications of $\phi$ to compute $c^t$ as $\phi^t(X, c^0)$. We also say that $\phi^0(X, c^0) = c^0$. The initial coloring is often given by the features $c^0 = F|_U$, meaning the color $c^0$ is given by the domain restriction of $F$ to the set of objects $U$. Then, the color of each object $u \in U$ is $c_u^0 = F_u$. If we are not given features, then we use a constant coloring, i.e., a coloring that assigns the same (arbitrary) color to each object. Color update functions can sometimes

be used to determine whether two graphs are isomorphic. For this, a coloring is computed for both graphs. If at some iteration the colorings of the two graphs are different this implies that the graphs are not isomorphic. If the colorings are never different, then the result of this method is inconclusive. A common color update function is the Weisfeiler-Leman algorithm (WL). In this work we define WL to use edge features.

**Definition 2.3.** (WL) The Weisfeiler-Leman algorithm (WL) is a color update function operating on a graph $G = (V, E, F)$ defined as $\text{WL}(v, G, c) = \text{HASH}\left(c_v, \{\!\{(c_u, F((u, v)) \mid u \in \mathcal{N}_G(v)\}\!\}\right)$ for $v \in V$ where HASH is an injective mapping to colors.

Message passing graph neural networks (MPNNs) can be seen as a generalization of WL where the colors correspond to learned node embeddings and each layer of the MPNN corresponds to a color update function. The color update function in layer $t \geq 1$ of an MPNN is defined as $\phi(v, G, c) = \text{COMB}^t\left(c_v, \text{AGG}^t\left(\{\!\{(c_x, F((x, v)) \mid x \in \mathcal{N}_G(v)\}\!\}\right)\right)$ where $\text{COMB}^t$ and $\text{AGG}^t$ are learnable functions, e.g., multi-layer perceptrons (MLPs).[1] We refer to WL and MPNNs as *standard message passing*. It has been shown that MPNNs are at most as *expressive* as WL, i.e., it can only distinguish non-isomorphic graphs that WL can distinguish. As there exist pairs of graphs that WL cannot distinguish this means that there exists graphs that MPNNs cannot distinguish either [Xu et al., 2019, Morris et al., 2019]. A common approach to improve the expressivity of standard message passing is to apply a function $T$ that transforms graphs to other relational structures. We call this mapping $T$ from a graph to a relational structure a *graph-to-structure encoding*. An example of an graph-to-structure encoding can be seen in Figure 1. This mapping $T$ is combined with a color update function $\phi$ tailored to this new structure. We use *non-standard message passing* to refer to color update functions that operate on relational structures that are not graphs. Some examples of non-standard message passing are $k$-WL [Immerman and Lander, 1990] which operates on $k$-tuples of nodes and CW Networks [Bodnar et al., 2021a] which operate on regular cell complexes.

# 3 Strong Simulation

We show that standard message passing together with graph transformations can achieve the same expressivity as many algorithms with non-standard message passing. As standard message passing operates on graphs we need to map relational structures to graphs. Note that merely guaranteeing at least the same expressivity as a color update function $\phi$ in each iteration can easily be achieved by, e.g., using the final coloring of $\phi$ as node features. To avoid such a trivial solution, we enforce straightforward restrictions on the generated graphs.

**Definition 3.1.** (Structure-to-graph encoding) Let $R$ be a mapping $R : X \mapsto G$ that maps relational structures $X = (U, \mathcal{N}_1, \ldots, \mathcal{N}_k, F)$ to graphs $G = (V, E, F')$. We call $R$ a structure-to-graph encoding if it can be written as $R(X) = R_{\text{feat}}(R_{\text{graph}}(U, \mathcal{N}_1, \ldots, \mathcal{N}_k), F)$ such that

1. $R_{\text{graph}}$ maps a relational structure *without* features to a graph $G = (V, E, F_{\text{graph}})$.

2. $R_{\text{feat}}((V, E, F_{\text{graph}}), F) = (V, E, F')$ creates the features $F'$ by concatenating each (node or edge) feature from $F_{\text{graph}}$ with the corresponding feature from $F$ if it exists.

As a graph-to-structure encoding $T$ maps graphs to structures and a structure-to-graph encoding $R$ maps structures to graphs, this means that $R \circ T$ maps graphs to graphs. We call such functions $R \circ T$ *graph transformations*, an example can be seen in Figure 1. As we define relational structures over *sets* of objects, this implies that $R$ is *permutation equivariant* with respect to a permutation of the objects. Furthermore, since $R$ is a function it is also *deterministic*. Next, we define strong simulation of color update functions.

**Definition 3.2.** (Strong Simulation) Let $\phi$ be a color update function. Let $R$ be a structure-to-graph encoding that runs in $\mathcal{O}(\tau_\phi(|U|))$ for every relational structure with object set $U$ and creates a graph with vertex set $V \supseteq U$. We consider two arbitrary relational structures from the domain of $\phi$, say $X = (U, \mathcal{N}_1, \ldots, \mathcal{N}_k, F)$ and $X' = (U', \mathcal{N}'_1, \ldots, \mathcal{N}'_k, F')$. Let $(V_1, E_1, F_1) = R(X)$ and $(V_2, E_2, F_2) = R(X')$. We say $\phi$ can be *strongly simulated* under $R$ if for every $t \geq 0$ it holds that $\text{WL}^t(R(X), F_1|_{V_1}) \cup \text{WL}^t(R(X'), F_2|_{V_2})$ refines $\phi^t(X, F|_U) \cup \phi^t(X', F'|_{U'})$.

---

[1]Here, we define MPNNs to use edge features which practically is almost always the case.

Intuitively, a color update function $\phi$ can be strongly simulated if instead of running $t$ iterations of $\phi$ on relational structure $X$ we can instead run WL on the graph $R(X)$ for $t$ iterations and achieve at least the same expressivity in every iteration. Strong simulation is a stronger property than expressivity: strong simulation implies being at least as expressive *in every iteration* whereas expressivity usually refers only to the final coloring. This guarantees that WL on $R(X)$ is *always* at least as expressive as $\phi$ and not just in *some* iteration. There exist MPNNs that are as expressive as WL in every iteration [Xu et al., 2019]. Thus, every layer of an MPNN can be as expressive as the corresponding iteration of WL. Note that as long as we restrict the size of the graphs (the number of vertices), strong simulation allows to learn a color update function $\phi$, e.g., a GNN, exactly. Indeed, if we combine each layer of an MPNN with a large enough MLP (with appropriate activation), we can learn to map the MPNN's embeddings to the output of $\phi$. This problem of fitting a finite number of points exactly is well studied and known as *memorization*, see e.g. Yun et al. [2019] for an overview.

To strongly simulate a large number of GNNs we introduce a class of color update functions we call *augmented message passing* (AMP). AMP is designed to capture many GNNs and we prove that AMP can be strongly simulated. AMP makes use of a building block we call *atoms*. Atoms model operations of color update functions such as colors, constants, or features that are focused on single vertices or pairs of vertices. AMP extends atoms with more general operations that are not necessarily focused on a single vertex such as function applications or aggregations over neighborhoods.

**Definition 3.3.** (Atom). Let $X = (U, \mathcal{N}_1, \ldots, \mathcal{N}_k, F)$ be a relational structure. Let $v, w \in U$ and $c_v, c_w$ be the colors of $v$ and $w$. Then, an atom $\rho(v, c_v, w, c_w)$ is a function that has exactly one of the following four forms:

(A1) $\rho(v, c_v, w, c_w) = k(v, w)$ for a *constant* $k(v, w)$ that only depends on $v$ and $w$ but not on the colors,

(A2) $\rho(v, c_v, w, c_w) \in \{F(v), F(w), F((w, v))\}$ where $F((w, v))$ is only part of this set if the edge $(w, v)$ is part of at least one neighborhood of $X$,

(A3) $\rho(v, c_v, w, c_w) \in \{c_v, c_w\}$, or

(A4) $\rho(v, c_v, w, c_w) = (\rho_1(v, c_v, w, c_w), \ldots, \rho_m(v, c_v, w, c_w))$ with atom $\rho_i$ for all $i \in [\![m]\!]$.

**Definition 3.4.** (Augmented Message Passing). Let $X = (U, \mathcal{N}_1, \ldots, \mathcal{N}_k, F)$ be a relational structure with $\ell(\mathcal{N}_i) = 1$ for all $i \in [\![k]\!]$. We call a color update function $\phi$ *augmented message passing* (AMP) if for all $u \in U$ and coloring $c$ of $X$, it can recursively be defined as exactly one of the following four forms:

(S1) $\phi(u, X, c) = \rho(u, c_u, u, c_u)$ where $\rho$ is an atom,

(S2) $\phi(u, X, c) = \{\!\{\rho(u, c_u, w, c_w) \mid w \in \mathcal{N}_i(u)\}\!\}$ where $\rho$ is any atom and any $i \in [\![k]\!]$,

(S3) $\phi(u, X, c) = (\phi_1(u, X, c), \ldots, \phi_m(u, X, c))$ where all $\phi_1, \ldots, \phi_m$ are AMP, or

(S4) $\phi(u, X, c) = f(\phi'(u, X, c))$ where $f$ maps colors to colors and $\phi'$ is AMP.

Having defined AMP, we next prove that AMP can be strongly simulated by providing a structure-to-graph encoding for every color update function from AMP. Full proofs are in Appendix C.

**Theorem 3.5.** *Augmented message passing can be strongly simulated.*

*Proof sketch.* We define *augmented message encoding* (AME) which returns a structure-to-graph encoding for every given AMP algorithm. Let $\phi$ be AMP that operates on a relational structure $X = (U, \mathcal{N}_1, \ldots, \mathcal{N}_k, F)$. $\text{AME}_\phi(X)$ returns a graph $G$ with vertex set $V_G = U$. Any constant value atom (A1) in $\phi$ and any feature in $F$ is stored in the vertex or edge features of $G$. If messages are passed between two objects then the graph will have an edge between the corresponding vertices with an edge feature encoding the neighborhood. We prove that every AMP $\phi$ can be strongly simulated under the structure-to-graph encoding $\text{AME}_\phi$. $\square$

AMP does not only contain variants of WL, but also GNN layers such as the layer of GSN [Bouritsas et al., 2022]. We say a GNN can be strongly simulated, if all of its layers correspond to color update functions that can be strongly simulated by the same structure-to-graph encoding $R$. This means

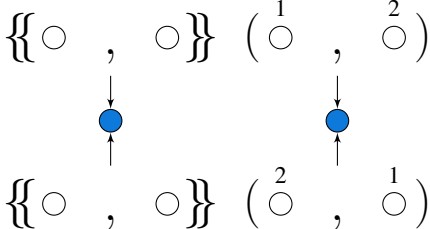

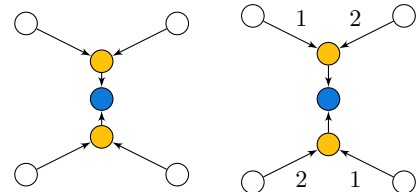

Figure 2: Left: nested aggregations. Right: non-pairwise message passing of 2-tuples. In both examples the state of two vertices gets combined and sent to the blue vertex. Note that for non-pairwise aggregations we have an order on the nodes in a tuple whereas for nested aggregations the aggregated vertices are unordered.

Figure 3: WL on the graph on the left (right) side weakly simulates the nested aggregations (non-pairwise message passing) from Figure 2 by adding additional vertices ( yellow ). This shows how weak simulation of both constructions is done in a similar way with the difference being that for non-pairwise aggregation we encode the order in the tuples on the edges.

that for every relational structure $X$ and every $t \geq 0$, WL achieves at least the same expressivity in iteration $t$ on $R(X)$ as layer $t$ of the GNN on $X$. We prove that this is always the case for color update functions that only differ in their use of function applications (S4) (see Appendix C.3). Different layers of a GNN commonly only differ by function applications (S4) as these correspond to learned functions. As many GNN layers use AMP, we can state one of the main results of this paper.

**Corollary 3.6.** *The following GNNs and variants of WL can be strongly simulated:*

1. $VV_C$-GNN [Sato et al., 2019]
2. $k$-WL / GNN [Morris et al., 2019]
3. $\delta$-$k$-(L)WL / (L)GNN [Morris et al., 2020]
5. GSN-e and GSN-v [Bouritsas et al., 2022]
4. $(k, s)$-LWL / SpeqNet [Morris et al., 2022]
6. DS-WL / GNN [Bevilacqua et al., 2021]

7. $k$-OSWL / OSAN [Qian et al., 2022]
8. $M_k$ GNN [Papp and Wattenhofer, 2022]
9. GMP [Wijesinghe and Wang, 2022]
10. Shortest Path Networks [Abboud et al., 2022]
11. Generalized Distance WL [Zhang et al., 2023]

*Proof sketch.* The proof consists of showing that the color update function underlying the non-standard message passing is AMP. By Theorem 3.5 this implies that the color update function can be simulated. □

**Runtime Complexity.** AME creates a vertex for every object and edges for every pair of objects that exchange messages. This implies that, creating the encoding requires the same amount of time as one iteration of AMP. Furthermore, classical message passing on the transformed graphs passes the same number of messages as AMP in every iteration. Thus, the overall runtime complexity of strongly simulating AMP is equivalent to that of running AMP itself.

**Example.** We illustrate AMP on the example of WL with triangle counts and without edge features. For every $v \in V_G$, we denote by $\triangle_v$ the number of triangles in the graph containing $v$. We define the color update function of WL with triangle counts: $\phi(v, G, c) = (c_v, \{\!\{(c_u, \triangle_u) \mid u \in \mathcal{N}_G(v)\}\!\})$. We show that WL with triangle counts is AMP. Consider the tuple $(c_u, \triangle_u)$, this consists of a color $c_u$ atom (A3) with a constant $\triangle_u$ atom (A1) combined into a tuple atom (A4). Thus, $(c_u, \triangle_u)$ is an atom which means that $\{\!\{(c_u, \triangle_u) \mid u \in \mathcal{N}_G(v)\}\!\}$ is AMP that aggregates this atom over the neighborhood $\mathcal{N}_G$ (S2). Finally, this AMP is combined with the color atom $c_v$ (A1) and combined into a tuple AMP (S3). Thus, the color update function of WL with triangle counts performs AMP which implies that it can be strongly simulated. For this it suffices to attach $\triangle_v$ to the features of every vertex $v \in V$.

## 4 Limits of Strong Simulation

To show the limits of strong simulation we identify three constructions that either cannot be strongly simulated or cannot be strongly simulated efficiently: time dependent neighborhoods, nested ag-

gregations, and non-pairwise message passing. These constructions have previously been used by Deac et al. [2022], Bevilacqua et al. [2021], Feng et al. [2022], Bodnar et al. [2021b], and Bodnar et al. [2021a]. We begin with time dependent neighborhoods which are used in the alternating message passing of Deac et al. [2022]. We illustrate them via the following example. Consider a color update function that updates the color of a vertex $a$ based on the color of $b$ in iteration $t$ but not in iteration $t'$. As it depends on the iteration whether $a$ and $b$ are neighbors, we call this *time dependent neighborhoods*. To model this in a graph, we require an edge between $a$ and $b$ in iteration $t$ or we would lose expressivity whenever the color of $b$ is important for the coloring of $a$. However, this means that the color of $a$ will also be updated with the color of $b$ in iteration $t'$. Thus, we can strongly simulate time dependent neighborhoods but we cannot do so efficiently (i.e., with the same number of sent messages), as we might need to send many more messages.

*Nested aggregations* are a form of color update functions used by Bevilacqua et al. [2021] and Feng et al. [2022]. Consider the following color update function from DSS-WL [Bevilacqua et al., 2021] $\phi(v, X, c) = (c_v, \{\!\{ \{\!\{ c_y \mid y \in \mathcal{N}(x) \}\!\} \mid x \in \mathcal{N}'(v) \}\!\})$ (see Figure 2 left).[2] We call this nesting of two aggregations ($\{\!\{ c. \mid \cdot \in \mathcal{N}(\cdot) \}\!\}$) a nested aggregation. To strongly simulate this we need an edge from every vertex of $\bigcup_{x \in \mathcal{N}'(v)} \mathcal{N}(x)$ to $v$. Furthermore, we need a way to group vertices $\mathcal{N}(x)$ together for every $x \in \mathcal{N}'(v)$. We prove that this is impossible with a structure-to-graph encoding and thus that nested-aggregations cannot be strongly simulated.

The third construction that we cannot strongly simulate is *non-pairwise message passing* which has been used by Bodnar et al. [2021b,a]. Non-pairwise message passing refers to aggregating colors as ordered tuples over a neighborhood $\mathcal{N}$ with $\ell(\mathcal{N}) > 1$. As an example, suppose we want to update the color of vertex $v$ based on the *ordered tuple* of colors $(c_a, c_b)$ of vertices $a$ and $b$ (see Figure 2 right). This is different from the aggregation by WL or AMP as those aggregations are unordered, i.e., $\{\!\{ c_a, c_b \}\!\}$. An issue arises when multiple such ordered tuples have to be used to update a color, as there is no way this can be done within the concept of strong simulation. Similar to nested aggregations, non-pairwise message passing cannot be strongly simulated. Note that non-pairwise message passing is a special case of nested-aggregations where an additional order on the aggregated objects is given as is demonstrated in Figure 2.

**Theorem 4.1.** *Nested aggregations and non-pairwise message passing cannot be strongly simulated.*

*Proof sketch.* To prove this (in Appendix D), we construct two relational structures that can be distinguished with nested aggregations in one iteration. These two structures have the same adjacencies but different features. We prove that no structure-to-graph encoding can create two graphs that one iteration of WL can distinguish. By the definition of graph-to-structure encoding, it is not possible for the edges of the resulting graph to depend on the original features. Thus, the two relational structure will lead to two graphs that have the same edges. We prove that WL cannot distinguish these two graphs in one iteration. The proof for non-pairwise message passing works similarly. □

## 5   Weak Simulation

We have shown in the previous section that not all color update functions can be strongly simulated. In this section, we introduce the more general concept of weak simulation and show that color update functions can be weakly simulated that are impossible to strongly simulate. Weak simulation differs from strong simulation by requiring only the same expressivity after certain number of iterations instead of every iteration.

**Definition 5.1.** (Weak Simulation) Let $\phi$ be a color update function. Let $R$ be a structure-to-graph encoding that runs in $\mathcal{O}(\tau_\phi(|U|))$ for every relational structure with object set $U$ and creates a graph with vertex set $V \supseteq U$. We consider two arbitrary relational structures from the domain of $\phi$, say $X = (U, \mathcal{N}_1, \ldots, \mathcal{N}_k, F)$ and $X' = (U', \mathcal{N}'_1, \ldots, \mathcal{N}'_k, F')$. Let $(V_1, E_1, F_1) = R(X)$ and $(V_2, E_2, F_2) = R(X')$. We say $\phi$ can be *weakly simulated* under $R$ with simulation factor $\zeta \geq 1$ if for every $t \geq 0$ it holds that $\mathrm{WL}^{\zeta \cdot t}(R(X), F_1|_{V_1}) \cup \mathrm{WL}^{\zeta \cdot t}(R(X'), F_2|_{V_2})$ refines $\phi^t(X, F|_U) \cup \phi^t(X', F'|_{U'})$.

---

[2]We have altered the notation of DSS-WL from the original paper for the sake of simplicity.

The simulation factor $\zeta$ measures the relative slowdown of weak simulation with respect to the original GNN. It follows that strong simulation is a special case of weak simulation as every strongly simulatable algorithm is weakly simulatable with a simulation factor of 1. Next, we prove that it is possible to weakly simulate non-pairwise aggregations and nested aggregations. This shows that weak simulation contains a larger set of color update functions than strong simulation. The full proof can be found in Appendix E.

**Theorem 5.2.** *Nested aggregations and non-pairwise message passing can be weakly simulated with simulation factor 2.*

*Proof sketch.* For weak simulation of nested aggregations consider the following construction. Suppose a nested aggregation $\{\!\{\{\!\{c_y^t \mid y \in \mathcal{N}(x)\}\!\} \mid x \in \mathcal{N}(v)\}\!\}$. For every $x \in \mathcal{N}(v)$ we create incoming edges from $\mathcal{N}(x)$ and an outgoing edge $(x, v)$ (see Figure 3 left). This allows for weak simulation with a simulation factor of 2. Non-pairwise message passing can be weakly simulated in a similar way. Suppose we intend to update the color of vertex $v$ based on the ordered tuple of colors $(c_a, c_b)$ of vertices $a$ and $b$. Then, we create a vertex $x$ and an edge $(x, v)$ for this message. Furthermore, we create an edge $(a, x)$ labeled with 1 to indicate that it is the first element in the tuple and an edge $(b, x)$ labeled with 2 (see Figure 3 right). This construction weakly simulates the non-pairwise message passing with simulation factor 2. □

As a consequence, we can weakly simulate the following algorithms.

**Corollary 5.3.** *The following GNNs and variants of WL can be weakly simulated:*

1. Message Passing Simplicial Networks [Bodnar et al., 2021b]
2. CW Networks [Bodnar et al., 2021a]
3. DSS [Bevilacqua et al., 2021]
4. K-hop message passing and KP-GNNs [Feng et al., 2022]

**Discussion.** It is possible to design graph transformations that result in smaller graphs, enabling more efficient weak simulation than the transformations automatically following from Theorem 5.2. We demonstrate that this is possible for all algorithms from Corollary 5.3. For Message Passing Simplicial Networks and CW Networks it is possible to construct graphs that require no additional vertices for weak simulation (see Appendix F.1). Similarly, it is possible to weakly simulate DSS without creating a large number of additional edges by adding a second copy of the original graph if we accept increasing the simulation factor to 3 as a trade-off (see Appendix F.2). Interestingly, $K$-hop message passing and KP-GNNs induce an additional order on the message passing which even allows for efficient *strong* simulation (see Appendix F.3). We did not find any architectures where weak simulation led to large increases in time and space complexity. However, in general, we cannot rule out the possibility of architectures that exhibit such an increase. This indicates the possibility for more powerful non-standard message passing, opening a promising direction for future work.

## 6 Experiments

We have shown many cases in which graph transformation based methods achieve the same expressivity as non-standard message passing. In this section, we empirically investigate whether MPNNs together with graph transformations can also achieve similar predictive performance as the simulated GNNs with non-standard message passing. We strongly simulate DS [Bevilacqua et al., 2021], weakly simulate DSS [Bevilacqua et al., 2021] and weakly simulate CW Networks (CWN) [Bodnar et al., 2021a], as representative approaches. For a non-standard message passing algorithm, we denote the corresponding graph transformation with a bar, e.g., $\overline{\text{DSS}}$. The code for our experiments can be found at https://github.com/ocatias/GNN-Simulation.

**Models.** With GIN [Xu et al., 2019] and GCN [Kipf and Welling, 2017] we use two of the most common MPNNs as baselines. We combine these MPNNs with the graph transformations $\overline{\text{DS}}$, $\overline{\text{DSS}}$ and $\overline{\text{CWN}}$. DS, $\overline{\text{DS}}$, DSS, and $\overline{\text{DSS}}$ require a policy that maps a graph to subgraphs, for this we chose the common 3-egonets policy that extracts the induced 3-hop neighborhood for each node [Bevilacqua et al., 2021]. We chose this policy as it creates only small subgraphs and Bevilacqua

Table 1: Predictive Performance of different GNNs on 10 different datasets. BEATS MPNN is the percentage of datasets on which the model beats the corresponding MPNN (either GIN or GCN, as noted in the model name). BEATS NSMP is the percentage of datasets where the graph transformation based model beats the corresponding non-standard message passing model. BEST MODEL is the percentage of datasets where this model achieves the best results among all models.

| | BEATS MPNN | BEATS NSMP | BEST MODEL |
|---|---|---|---|
| GIN | - | - | 0% |
| GCN | - | - | 0% |
| DS (GIN) | 60% | - | 10% |
| $\overline{\text{DS}}$ + GIN | 60% | 60% | 0% |
| $\overline{\text{DS}}$ + GCN | 70% | 30% | 0% |
| DSS (GIN) | 70% | - | 20% |
| $\overline{\text{DSS}}$ + GIN | 70% | 60% | 30% |
| $\overline{\text{DSS}}$ + GCN | 70% | 30% | 10% |
| CWN | 60% | - | 20% |
| $\overline{\text{CWN}}$ + GIN | 60% | 50% | 0% |
| $\overline{\text{CWN}}$ + GCN | 50% | 30% | 10% |

et al. [2021] reported good results for it. More details on $\overline{\text{DSS}}$ can be found in Appendix F.2. For CWN and $\overline{\text{CWN}}$ we construct cells as described by Bodnar et al. [2021a]. More details on $\overline{\text{CWN}}$ can be found in Appendix G and F.1. We apply the same training and evaluation procedure to all GNNs.

**Datasets.** Due to the large number of combination of models and graph transformations, we focus on medium size datasets. To experimentally investigate how the graph transformations increase expressivity, we perform an initial investigation with GIN on the synthetic CSL dataset [Murphy et al., 2019, Dwivedi et al., 2023]. For real world prediction tasks, we use all real world datasets with less than $10^5$ graphs that provide a train, validation, and test split used in Bodnar et al. [2021a], Bevilacqua et al. [2021]: ZINC [Gómez-Bombarelli et al., 2018, Sterling and Irwin, 2015], ogbg-molhiv and ogbg-moltox21 [Hu et al., 2020]. Additionally, we add seven small molecule datasests from OGB [Hu et al., 2020]. In total, we evaluate on 10 real-life datasets.[3]

**Setup.** For real-life datasets we combine all baseline models (GCN, GIN) with all graph transformations ($\overline{\text{DS}}$, $\overline{\text{DSS}}$, $\overline{\text{CWN}}$) and tune hyperparameters individually (including CWN, DS and DSS; see Appendix H). We also measure the preprocessing and training speeds for different models (details and speed results are in Appendix H.3).

**Results.** Table 2 shows the results on CSL. GIN achieves 10% accuracy whereas all other methods achieve a perfect accuracy. This confirms that just like non-standard message passing, MPNNs with graph transformations have superior expressivity compared to GIN. Table 1 summarizes the results on all real life datasets. Indeed, MPNNs with graph transformations achieve the best result on half of them. Just like non-standard message passing, MPNNs with graph transformations outperform MPNNs in the majority of experiments. Notably, graph transformations combined with GIN outperform non-standard message passing in more than half of the experiments. This shows that simulated networks not only have provably at least the same expressivity but also perform surprisingly well in practice.

Table 2: Accuracy on CSL. **Bold** results outperform GIN.

| MODEL | CSL ACCURACY |
|---|---|
| GIN | $0.1 \pm 0.0$ |
| $\overline{\text{CWN}}$ | $\mathbf{1.0 \pm 0.0}$ |
| $\overline{\text{CWN}}$ + GIN | $\mathbf{1.0 \pm 0.0}$ |
| $\overline{\text{DSS}}$ | $\mathbf{1.0 \pm 0.0}$ |
| $\overline{\text{DSS}}$ + GIN | $\mathbf{1.0 \pm 0.0}$ |
| $\overline{\text{DS}}$ | $\mathbf{1.0 \pm 0.0}$ |
| $\overline{\text{DS}}$ + GIN | $\mathbf{1.0 \pm 0.0}$ |

---

[3]We use the following datasets: ZINC with 12k nodes, ogbg-molhiv, ogbg-moltox21, ogbg-molesol, ogbg-molbace, ogbg-molclintox, ogbg-molbbbp, ogbg-molsider, ogbg-moltoxcast, and ogbg-mollipo.

# 7 Discussion and Conclusion

**Discussion.** The purpose of simulating a GNN is to allow an MPNN to learn similar embeddings of graphs as the GNN. We expect such embeddings to model the similarity of graphs in a more fine-grained way than just distinguishing graphs. Thus, we have to avoid trivial solutions: any MPNN can be made as expressive as any other GNN when the output of the GNN is attached to the input of the MPNN. In fact, it is even possible to make an MPNN maximally expressive by precomputing the isomorphism class of the graph, e.g., a number uniquely encoding whether two graphs are isomorphic, and inputting it into the MPNN. However, such techniques defeat the purpose of learning a graph representation in the first place and potentially require a super-polynomial runtime [Babai, 2016]. To avoid such scenarios, we introduce two requirements for weak and strong simulation: (1) the features of the resulting graph depend on the original features solely by concatenation and (2) the transformation must operate in linear time relative to the simulated color update function. These two constraints render it impossible to precompute the colors generated by the simulated method for a non-constant number of iterations and prohibit the computation of graph isomorphism classes.

**Implications.** Our work has implications on (1) the theory of GNNs, (2) the design of new GNNs, and (3) the implementation GNNs with graph transformations. For (1): simulation allows to investigate the expressivity of different GNNs through a unified lens by analyzing the corresponding graph transformation. It should be possible to obtain VC bounds for any weakly simulatable GNN by using the results from Morris et al. [2023]. Similarly, we believe that it is possible to use Geerts and Reutter [2022] to get expressivity upper-bounds in term of the $k$-WL test for any weakly simulatable GNN. For (2): our theorems indicate that nested aggregations and non-pairwise message passing cannot be strongly simulated and are thus fundamentally different from the message passing paradigm. Thus, to build GNNs that go beyond MPNNs in expressivity it seems promising to investigate such constructions. For (3): instead of implementing a GNN with non-standard message passing, our theorems imply it can be implemented as a graph transformation together with an MPNN. This makes it easier to implement GNNs as of-the-shelf MPNNs can be used and makes the resulting method agnostic from the deep learning framework as the graph transformation does most of the heavy lifting.

**Conclusion.** We introduced weak and strong simulation of non-standard message passing. Simulation allows an MPNN together with a graph transformation to obtain the same expressivity every $\zeta \geq 1$ layers (weak simulation) or in every layer (strong simulation) as GNNs with non-standard message passing. We have proposed a straightforward way of proving that an algorithm can be simulated and to generate the corresponding graph transformations. This generalizes previous ideas [Otto, 1997, Morris et al., 2019, 2020, 2022, Qian et al., 2022] and makes simulating GNNs accessible to anyone who intends to build novel GNNs. In total, we have shown that 15 GNNs can be simulated. We chose these 15 models to showcase the variety of GNNs that fall under augmented message passing and expect many more GNNs to be simulatable as well. In future research, we will investigate the use of simulation to analyze aspects of GNNs such as generalization, oversmoothing, and oversquashing.

**Acknowledgements.** Part of this work has been supported by the Vienna Science and Technology Fund (WWTF) project ICT22-059. The authors would like to thank the reviewers for the feedback. We are especially thankful to our colleagues Pascal Welke and Tamara Drucks for giving extensive feedback on a draft of this paper.

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

# A   Related Work: Expressivity of GNNs

In this section, we talk about research on the expressivity of GNNs and how it relates to our paper. Xu et al. [2019] and Morris et al. [2019] proved that MPNNs have limited expressivity. This lead to the development of new GNNs that have higher expressivity than MPNNs. We distinguish between GNNs that perform non-standard message passing and GNNs that use an MPNN on a transformed graph. We begin with approaches that correspond to MPNNs on a transformed graph. A common approach for this adds additional features to the graph. It has been proven that adding random features to vertices improves the expressivity of GNNs [Abboud et al., 2021, Dasoulas et al., 2020, Sato et al., 2021] which allows them to learn every function defined on a graph and thus gives them maximum expressivity. However, to do this they sacrifice determinism, and permutation invariance / equivariance. Some permutation equivariant approaches extend the graph features by counting patterns such as Barceló et al. [2021] which extend the MPNN with rooted homomorphism counts of a set of patterns. All approaches that consist of deterministically adding additional features to vertices and edges in a permutation equivariant way can be trivially strongly simulated and belong to augmented message passing. Graph Structural Networks (GSN-e and GSN-v; Bouritsas et al. [2022]) introduce a new graph convolution layer which extends messages with subgraph isomorphism counts. This cannot be directly seen as transforming graphs. However, it is very similar to adding these subgraph isomorphism counts to the vertex features and thus to an MPNN operating on a transformed graph.

Next, we consider approaches that use non-standard message passing. Such approaches often operate on different structures such as (1) $k$-tuples of nodes, (2) topological objects such as simplical complexes or regular cell complexes, and (3) subgraphs. Type (1) passes messages between $k$-tuples of nodes. Higher dimensional WL ($k$-WL) is a generalization of WL and forms a sequence of algorithms that are stronger than WL [Immerman and Lander, 1990]. Increasing $k$ increases expressivity at the cost of exponentially increasing the runtime. Morris et al. [2019] introduced $k$-dimensional GNNs which extend the concept of $k$-WL to GNNs. For every $k$ there exist $k$-GNNs that have equal expressivity as $k$-WL. However, as $k$ increases, the runtime of $k$-GNNs grows exponentially. To combat this, Morris et al. [2020] introduced (local) $\delta$-$k$ dimensional WL and GNNs; Morris et al. [2022] introduce $(k, s)$-LWL and $(k, s)$-SpeqNets. These methods make use of the sparsity of the graph to reduce computation time. Type (2) operates on topological structures such as simplical complexes or regular cell complexes. Three prominent examples of this idea are Simplical Networks [Bodnar et al., 2021b], CW Networks [Bodnar et al., 2021a] and message-passing combinatorial complex neural networks [Hajij et al., 2023]. Other algorithms that work on topological structures are Simplical Neural Networks [Ebli et al., 2020], Dist2Cycle [Keros et al., 2022], and Cell Complex Neural Networks [Hajij et al., 2020]. Type (3) decomposes the graph into subgraphs and then performs message passing on these subgraphs. Examples of these algorithms are Automorphism-based Neural Networks [Thiede et al., 2021], Subgraph GNNs [Frasca et al., 2022], and Equivariant Subgraph Aggregation Networks (ESAN; Bevilacqua et al. [2021]). ESAN uses a policy to transform a graph into a set of subgraphs and then processes them either separately (DS-WL / GNN) or in a way that allows information transfer between the subgraphs (DSS-WL / GNN). The GNN $M_k$ [Papp and Wattenhofer, 2022] separately processes subgraphs consisting of the entire original graph with $k$ nodes that are marked. Ordered subgraph aggregation networks ($k$-OSAN) [Qian et al., 2022] operate on $k$-ordered vertex subgraphs, $k$-OSAN is upper bounded in expressivity by $(k+1)$-WL and incomparable to $k$-WL. Frasca et al. [2022] proves that GNNs based on subgraphs are upper bounded by 3-WL. Finally, we list other GNNs with non-standard message passing that we can simulate: $VV_c$-GNNs [Sato et al., 2019] have been shown to be stronger than standard message passing GNNs in approximating algorithmic problems such as minimum dominating set. Wijesinghe and Wang [2022] introduce a generalized message passing framework (GMP) which allows them to inject local structure into the aggregation scheme. $K$-hop message passing [Feng et al., 2022] extend MPNNs from 1-hop neighborhoods to $K$-hop neighborhoods. KP-GNNs [Feng et al., 2022] extends $K$-hop message passing by including additional information about the subgraphs induced by the $K$-hop neighborhood. In summary, there exist many GNNs that are more expressive than MPNNs. Despite the conceptual differences between these GNNs we show that many of them are some form of augmented message passing and thus can be simulated.

# B   Related Work: Simulating Message Passing

We investigate previous approaches to simulating GNNs with MPNNs on transformed graphs by analyzing papers and source code of different architectures. We begin with algorithms from Corollary 3.6. GSN-e / GSN-v [Bouritsas et al., 2022] and DS-WL / DS-GN [Bevilacqua et al., 2021] were not implemented via graph transformations. In practice however, there is little difference between their implementation of DS-WL / DS-GN and our implementation with graph transformations: Bevilacqua et al. [2021] implement subgraphs as separate graphs while we implement them as separate subgraphs of the same graphs. For $VV_C$-GNNs [Sato et al., 2019], we could find no indication that they were implemented with graph transformations. $M_k$-GNNs [Papp and Wattenhofer, 2022] is a purely theoretic work and has not been implemented by the corresponding authors. For GMP [Wijesinghe and Wang, 2022], we are not sure if their code can count as being implemented as a graph transformation as they adapt message passing layers to use their structural information. However, for this they change the definition of the layers whereas we only change the graphs. Morris et al. [2019] introduced $k$-GNNs and implemented them with MPNNs on a transformed graph. Morris et al. [2019] even call this *simulation* and give a short proof that WL can simulate $k$-WL on transformed graphs. This corresponds to our concept of strong simulation, and this property of $k$-WL was already noticed by Otto [1997] [Grohe et al., 2021].

For $\delta$-$k$-(L)GNN, Morris et al. [2020] write that they implemented $\delta$-$k$-(L)GNNs

> [ . . . ] using PYTORCH GEOMETRIC, using a Python-wrapped C++11 preprocessing routine to compute the computational graphs for the higher-order GNN. We used the GIN-$\epsilon$ layer to express $f_{\mathrm{mrg}}^{W_1}$ and $f_{\mathrm{aggr}}^{W_2}$ of [ . . . ].

An inspection of their code reveals that they create a similar graph structure as we propose in the proof of Corollary 3.6. For $\delta$-$k$-GNN they create a graph where the vertices represent the $k$-tuples. They add an edge between two nodes if the tuples are $j$-neighbors, with $j \in [\![k]\!]$. Just like in our graph transformation the edge type encodes whether the tuples are *local* or *global* $j$-neighbors and the neighborhood $j$. For $(k, s)$-SpeqNet (or $(k, s)$-LWL), Morris et al. [2022] write that $(k, s)$-SpeqNets were implemented

> [ . . . ] using PyTorch Geometric, using a Python-wrapped C++11 preprocessing routine to compute the computational graphs for the higher-order GNNs. We used the GIN-$\epsilon$ layer to express $f_{\mathrm{mrg}}^{W_1}$ and $f_{\mathrm{aggr}}^{W_2}$ of [ . . . ].

Similar to our graph transformation from Corollary 3.6, they create a new graph where each node corresponds to an ordered subgraph $\mathbf{v} \in V_G^k$ of size $k$ with at most $s$ connected components in this subgraph. Just like in our graph transformation, edges are added between nodes if their tuples only differ at position $j \in [\![k]\!]$ and the nodes at position $j$ are neighbors. Additionally, the edge features in the transformed graph encode the edge features from the original graph in a way that allows them to encode $j$. Thus, their graph transformation follows ours. Both Morris et al. [2020] and Morris et al. [2022] reference Morris et al. [2019] to argue that there exist instantiations powerful enough for $f_{\mathrm{mrg}}^{W_1}$ and $f_{\mathrm{aggr}}^{W_2}$. For $k$-OSWL / OSAN, Qian et al. [2022] state in their paper

> [ . . . ] the computation of $k$-OSWL's coloring relies on the simple and easy-to-implement 1-WL.

It follows that the authors of these four papers were aware of *simulation*. However, to the best of our knowledge they have never formalized or fully explained their idea. In contrast, we fully formalize the idea of simulation and provide proofs that show that these graph transformations provably simulate the corresponding GNNs. Additionally, we provide recipes for obtaining graph transformations for augmented message passing algorithms.

To the best of our knowledge, so far Simplical Message Passing Networks [Bodnar et al., 2021b], CWN [Bodnar et al., 2021a], DSS-GNN / WL [Bevilacqua et al., 2021], $K$-hop message passing and KP-GNNs [Feng et al., 2022] have not been implemented using weak simulation. However, another topological deep learning network the *combinatorial complex neural network* [Hajij et al., 2023] has been implemented based on GNNs. For this, the authors use a transformation to an augmented Hasse graph by transforming cells to vertices and adding edges between vertices corresponding to adjacent cells. This is similar to $\overline{\mathrm{CWN}}$, our proposed transformation from regular cell complexes to graph. Specifically, Hajij et al. [2023] state the following about their implementation TopoEmbedX:

TopoEmbedX [...] supports higher-order representation learning on cell complexes, simplicial complexes, and CCs. [...] TopoEmbedX converts a given higher-order domain into a subgraph of the corresponding augmented Hasse graph, and then utilizes existing graph representation learning algorithms to compute the embedding of elements of this subgraph.

Finally, Veličković [2022] proposes that all functions over graphs that are of interest can be computed with message passing over a transformed graph. They call this *augmented message passing* which inspired our use of this term. They discuss how to perform some particular variants of non-standard message passing through standard message passing. This is helpful and demonstrates the wide applicability of this approach. However, their work is intended as a positional paper and thus mostly stays high level and does not provide sufficient formal proofs to back up their claims. Providing the necessary proofs is non-trivial: once one understands simulation it is straightforward to apply it to a large number of architectures and it is simple to prove simulatability for any one given architecture. However, difficulty arises when one tries to prove this for a large number of architectures. Thus, the difficulty is not proving simulatability but defining a framework that allows us to prove this for many different architectures. We provide such a framework based on color update functions. Furthermore, Veličković [2022] does not provide a formal definition of simulation (or in their case augmented message passing). Indeed, it is trivially possible to build maximally expressive MPNNs by precomputing the isomorphism class of the input graph. Thus, without a restriction on the transformation, the general statement that a GNN can be implemented as an MPNN plus a transformation is only meaningful if the transformation is in some sense close to what the GNN does. We solve this issue through our careful definition of structure-to-graph encoding which restrict how the transformation can access features of the structure and through the definition of weak / strong simulation that pose a runtime restriction on the transformation.

## C  Proofs for Strong Simulation (Section 3)

### C.1  Rewriting AMPs (Lemma C.1)

We use the following Lemma that allows us to rewrite AMPs without losing the strong simulation property. For a function $\phi$ we use IM$[\phi]$ to denote the image of $\phi$.

**Lemma C.1.** *Let $\phi$ be a color update function. We are given a set of objects with neighborhood functions from the domain of $\phi$ and colors. Let $\Gamma(\phi)$ be a color update functions such that there exists an injective (bijective) mapping $A$ between IM$[\phi]$ and IM$[\Gamma[\phi]]$. Then, a coloring obtained from $\Gamma(\phi)$ refines a coloring obtained from $\phi$. Furthermore, if the mapping $A$ is bijective the coloring from $\phi$ refines the coloring from $\Gamma(\phi)$.*

This lemma allows us to simplify AMPs by replacing functions (S4) with the identity function i.e. $f(\phi)$ is replaced by $\phi$. This transformation is bijective if $f$ is injective. Furthermore, we can use this to flatten tuples (A4) (S3) i.e. $(\ldots, (\phi_1, \ldots, \phi_n), \ldots)$ is replaced by $(\ldots, \phi_1, \ldots, \phi_n, \ldots)$. Flattening tuples with the same structure is bijective. Thus, we can apply these transformations without losing the simulatable property. We prove the correctness of the above lemma.

*Proof.* Let $A$ be the injective mapping from the color assigned to objects from $\phi$ to $\Gamma(\phi)$. For an arbitrary iteration, let $\pi_w, \pi_p$ be the colors assigned to $w \in W, p \in P$ by $\Gamma(\phi)$ and $c_w, c_p$ the colors assigned to them by $\phi$. We start by showing that if there exists the required injective mapping, then the coloring $\pi$ from $\Gamma(\phi)$ refines the coloring $c$ from $\phi$. For this, we assume that $\phi$ can distinguish $w, p$ and show that this implies that $\Gamma(\phi)$ can also distinguish them. Formally, we show $c_w \neq c_p \Rightarrow \pi_w \neq \pi_p$, this is the contraposition of the definition of refinement. Suppose $c_w \neq c_p$, then from $\pi_w = A(c_w)$ and $\pi_p = A(c_p)$ and the fact that $A$ is injective follows that $\pi_w \neq \pi_p$.

Next, we show that if $A$ is bijective then the colorings from $\phi$ and $\Gamma(\phi)$ refine each other. We assume that $A$ is bijective. From the above argument, it then follows that the coloring from $\Gamma(\phi)$ refines the coloring from $\phi$. It remains to show that the coloring from $\phi$ refines that from $\Gamma(\phi)$. Similar to above, we show that if $\Gamma(\phi)$ can distinguish $w$ and $p$, then so can $\phi$. Thus, we assume $\pi_w \neq \pi_p$ and show $c_w \neq c_p$. Using the bijectiveness of $A$ it follows from our assumption that $A^{-1}(\pi_w) \neq A^{-1}(\pi_p)$ and thus that $c_w \neq c_p$. This concludes the proof. $\qquad\square$

---

**Algorithm 1** Helper algorithm for atoms

---

**Input:** atom $\rho$ of type (A1), (A2),(A3), graph $(V, E, F_{\text{graph}})$, object $w$, object $q$, indices $i, j, l$.
**Output:** updated graph $G = (V, E, F_{\text{graph}})$
**if** $\rho$ is of type (A2) or (A3) **then**
    **return** $(V, E, F_{\text{graph}})$
**else** $\rho(w, c_w, q, c_q)$ is a constant $k(w, q)$
    **if** $w$ is $q$ **then**
        $F_{\text{graph}}(w) := \text{ENC}(F_{\text{graph}}(w), i, j, l, k(w, w))$
    **else**
        $F_{\text{graph}}((q, w)) := \text{ENC}(F_{\text{graph}}((q, w)), i, j, l, k(w, q))$
    **end if**
**end if**
**return** $(V, E, F_{\text{graph}})$

---

**Algorithm 2** Helper algorithm for (S2)

---

**Input:** AMP (S2) of shape $\{\{\rho(w, c_w^t, q, c_q^t) \mid q \in \mathcal{N}_s(w)\}\}$ where $\rho$ is an atom and $\ell(\mathcal{N}_s) = 1$;
graph $(V, E, F_{\text{graph}})$, relational structure $(U, \mathcal{N}_s)$ without features, indices $i, j$.
**Output:** updated graph $G = (V, E, F_{\text{graph}})$
**for** $u \in U$ **do**
    **for** $q \in \mathcal{N}_s(u)$ **do**
        $e := (q, u)$
        **if** $e \notin E$ **then**
            $E := E \cup \{e\}$
            $F_{\text{graph}}(e) = $ empty vector
        **end if**
        $F_{\text{graph}}(e) := \text{ENC}(F_{\text{graph}}(e), i, j, 0, s)$
    **end for**
**end for**
/* $\rho$ has the shape $(\rho_1(w, c_w, \Omega_1, c_q), \ldots, \rho_m(w, c_w, \Omega_m, c_q))$ */
**for** $l \in [\![m]\!]$ **do**
    /* $\rho_l$ is either an (A1), (A2) or an (A3) */
    **for** $u \in U$ **do**
        **for** $q \in \mathcal{N}_s(u)$ **do**
            $(V, E, F_{\text{graph}}) := \text{Algorithm } 1(\rho_l, (V, E, F_{\text{graph}}), u, q, i, j, l)$
        **end for**
    **end for**
**end for**
**return** $(V, E, F_{\text{graph}})$

---

### C.2 Augmented Message Passing can be Simulated (Theorem 3.5)

We are interested in showing the following theorem.

**Theorem 3.5.** *Augmented message passing can be strongly simulated.*

To prove the above theorem we define augmented message encoding (AME) in Algorithm 4. For this we assume a function ENC: $(x, i, j, l, c) \mapsto x'$, that updates a feature $x$ to $x'$ by attaching the color $c$ at the indices $i, j, l$. We assume that applying ENC multiple times never overwrites information if different indices are used. For example, we can think of the feature $x$ as a 3d-tensor such that $\text{ENC}(x, i, j, l, c)$ updates this vector by writing the entry $c$ at position $i, j, l$. Observe, that in Algorithm 4 it never happens that the same indices are used, thus whenever AME encodes information in the graph, this information does not get overwritten by latter operations. We prove the theorem by showing that AME+WL can simulate any given AMP.

*Proof.* (Theorem 3.5) Let $\phi$ be an arbitrary augmented message passing algorithm. We consider two arbitrary relational structures from the domain of $\phi$, say $X = (U, \mathcal{N}_1, \ldots, \mathcal{N}_k, F)$ and $X' = (U', \mathcal{N}_1', \ldots, \mathcal{N}_k', F')$. Let $(U, E_1, F_1) = R(X)$ and $(U', E_2, F_2) = R(X')$. We prove (1) that

---

**Algorithm 3** Helper algorithm for AMP

---

**Input:** AMP color update function $\phi = (\phi_1, \ldots, \phi_n)$ where each individual AMP is (A1), (A2), (A3) or (S2), graph $(V, E, F_{\text{graph}})$, relational structure $(U, \mathcal{N}_1, \ldots, \mathcal{N}_k)$ without features where $\ell(\mathcal{N}_s) = 1$ for all $s \in [\![k]\!]$, index $i$.
**Output:** updated graph $G = (V, E, F_{\text{graph}})$
**for** $j \in [\![n]\!]$ **do**
   **if** $\phi_j$ is (A1), (A2) or (A3) **then**
      **for** $u \in U$ **do**
         $(V, E, F_{\text{graph}}) := $ Algorithm $1(\phi_j, (V, E, F_{\text{graph}}), u, u, i, j, 0)$
      **end for**
   **else**
      /* $\phi_i$ has the shape $\{\!\{\rho(w, c_w, q, c_q) \mid q \in \mathcal{N}_s(w)\}\!\}$ where $\rho$ is an atom and $s \in [\![k]\!]$ */
      $(V, E, F_{\text{graph}}) := $ Algorithm $2(\phi_i, (V, E, F_{\text{graph}}), (U, \mathcal{N}_s), i, j)$
   **end if**
**end for**
**return** $(V, E, F_{\text{graph}})$

---

**Algorithm 4** Augmented message encoding (AME)

---

**Input:** relational structure $X = (U, \mathcal{N}_1, \ldots, \mathcal{N}_k, F)$ where $\ell(N_i) = 1$ for every $i \in [\![k]\!]$, AMP color update function $\phi$.
**Output:** graph $G = (U, E, F')$

/* $R_{\text{graph}}$ */
$E := \emptyset$
$F_{\text{graph}} := $ empty features
Replace function applications (S4) in $\phi$ by the identity function, i.e., $f(\phi)$ becomes $\phi$.
Flatten all (A4), (S3) constructions in $\phi$ except the outermost, i.e., $(\ldots, (\phi_1, \ldots, \phi_n), \ldots)$ becomes $(\ldots, \phi_1, \ldots, \phi_n, \ldots)$.
/* $\phi$ has the shape $(\phi_1, \ldots, \phi_n)$ where each $\phi_i$ with $i \in [\![n]\!]$ is one of (A1), (A2), (A3) or (S2) */
$(U, E, F_{\text{graph}}) := $ Algorithm $3(\phi, (U, E, F_{\text{graph}}), (U, \mathcal{N}_1, \ldots, \mathcal{N}_k), 0)$

/* $R_{\text{feat}}$ */
$F' := $ empty features
**for** $u \in U$ **do**
   $F'(u) := $ concat $(F(u), F_{\text{graph}}(u))$
**end for**
**for** $e \in E$ **do**
   $F'(e) := $ concat $(F(e), F_{\text{graph}}(e))$
**end for**
**return** $(U, E, F')$

---

$\text{AME}_\phi$ is a structure-to-graph encoding such that $R(X)$ runs in $\mathcal{O}(\tau_\phi(|U|))$; (2) for every $t \geq 0$ the coloring $\text{WL}^t(R(X), F_1|_U) \cup \text{WL}^t(R(X'), F_2|_{U'})$ on $\text{AME}_\phi(X)$ refines the coloring $\phi^t(X, F|_U) \cup \phi^t(X', F'|_{U'})$.

We begin by proving (1). Observe that Algorithm 4 can be split into $R_{\text{graph}}$ and $R_{\text{feat}}$ as required by the definition of structure-to-graph encoding. Furthermore, Algorithm 4 first iterates through all AMPs in $\phi$ and all possible objects that are relevant to these AMPs. Hence, Algorithm 4 runs in the same runtime as one iteration of AMP which is in $\mathcal{O}(\tau_\phi(|U|))$.

Next, we prove (2). We use $\pi_u^t, \pi_r^t$ and $c_u^t, c_r^t$ to denote the colors assigned to vertex / object $u, r \in (U \cup U')$ in iteration $t$ by WL on $R(X), R(X')$ or $\phi$ on $X, X'$, respectively. We show by induction that if WL cannot distinguish $u$ and $r$ in a given iteration $t$ then neither can $\phi$. Formally, we show $\pi_u^t = \pi_r^t \Rightarrow c_u^t = c_r^t$ which implies that $\text{WL}^t(R(X), F_1|_U) \cup \text{WL}^t(R(X'), F_2|_{U'})$ refines $\phi^t(X, F|_U) \cup \phi^t(X', F'|_{U'})$ as the domain of the coloring computed by $\phi$ is the same as the domain of the coloring computed by WL $(U \cup U')$.

Algorithm 4 starts by replacing all function applications (S4) by the identity function and flattens all tuples (A4) (S3). Recall that this transformation is either injective or bijective. By applying Lemma C.1 they do not influence the strong simulation property. From now on we assume that $\phi = (\phi_1, \ldots, \phi_n)$ such that all $\phi_j$ with $j \in [\![n]\!]$ are either (A1), (A2) (A3), or (S2) and if they are an atom, then the first and third function argument are identical. We use $w$ to denote the object the AMP is defined to update. Algorithm 3 iterates through all $\phi_1, \ldots, \phi_n$ and encodes them in the graph.

**Base case:** We show that $\pi_u^0 = \pi_r^0 \Rightarrow c_u^0 = c_r^0$. For this, we assume that $\pi_u^0 = \pi_r^0$. Since the initial colors of WL correspond to the vertex features which encode the object features of $X$ and $X'$ that are used as the initial coloring of AMP, this implies that $c_u^0 = c_r^0$.

**Induction hypothesis:** We assume that $\pi_u^t = \pi_r^t \Rightarrow c_u^t = c_r^t$ holds for all $t \leq T$.

**Induction step:** We show that $\pi_u^{T+1} = \pi_r^{T+1} \Rightarrow c_u^{T+1} = c_r^{T+1}$. We assume $\pi_u^{T+1} = \pi_r^{T+1}$. We iterate through all AMPs from $\phi_1, \ldots, \phi_n$ and show that they must all return the same result for $u$ and $r$ which implies $c_u^{T+1} = c_r^{T+1}$. We do a case distinction on the type of this AMP. Note that in AMP the object / edge features are referred to us $F(\cdot)$ whereas in the relational structures $X$ and $X'$ they are denoted as $F(\cdot)$ and $F'(\cdot)$. Below, when we argue that two AMP constructions are equal and use the $F(\cdot)$ notation and neglect $F'(\cdot)$ for the sake of simplicity.

(A1) $\left( k(u, u) \overset{!}{=} k(r, r) \right)$. This atom was encoded into the graph by Algorithm 1. These constants are part of the vertex features of the graphs $R(X), R(X')$ and thus are part of the initial coloring $\pi_u^0, \pi_r^0$. Since WL refines colorings, the assumption that $\pi_u^{T+1} = \pi_r^{T+1}$ implies $\pi_u^0 = \pi_r^0$. Thus, $k(u, u) = k(r, r)$.

(A2) $\left( F(u) \overset{!}{=} F(r) \right)$. As the object features are encoded into the vertex features of $R(X), R(X')$ (see $R_{\text{feat}}$ in Algorithm 4), they are part of the initial coloring $\pi_u^0, \pi_r^0$. Since WL refines colorings, the assumption that $\pi_u^{T+1} = \pi_r^{T+1}$ implies $\pi_u^0 = \pi_r^0$. Thus, $F(u) = F(r)$.

(A3) $\left( c_u^T \overset{!}{=} c_r^T \right)$. This atom was not encoded into the graph. From our assumption we know that $\pi_u^{T+1} = \pi_r^{T+1}$. Since WL refines colorings this implies that $\pi_u^T = \pi_r^T$. By applying the induction hypothesis we obtain that $c_u^T = c_r^T$.

(S2) $\left( \{\!\{ \rho(u, c_u^T, x, c_x^T) \mid x \in \mathcal{N}_i(u) \}\!\} \overset{!}{=} \{\!\{ \rho(r, c_r^T, y, c_y^T) \mid y \in \mathcal{N}_i(r) \}\!\} \right)$. This AMP was encoded into the graph by Algorithm 2. Note that the neighborhood relation $\mathcal{N}_i$ was encoded by edges with edge features labeled $i$, this means that if two nodes have the same color $\pi_u^{T+1} = \pi_r^{T+1}$ then for every neighborhood relation $\mathcal{N}_i$ there exists a bijective $\alpha : \mathcal{N}_i(u) \to \mathcal{N}_i(r)$ such that for all $x \in \mathcal{N}_i(u)$ it holds that $\pi_x^T = \pi_{\alpha(x)}^T$ and that the edges $(x, u), (\alpha(x), r)$ have the same edge features. From the induction hypothesis it then follows that $c_x^T = c_{\alpha(x)}^T$. Let $\rho(\cdot, c_\cdot^T, \Omega, c_\Omega^T) = \left( \rho_1(\cdot, c_\cdot^T, \Omega_1, c_{\Omega_1}^T), \ldots, \rho_m(\cdot, c_\cdot^T, \Omega_m, c_{\Omega_m}^T) \right)$ for some $m \geq 1$. We show for an arbitrary $j \in [\![m]\!]$ that $\rho_j \left( u, c_u^T, \Omega_j, c_{\Omega_j}^T \right) = \rho_j \left( r, c_r^T, \Omega_j', c_{\Omega_j'}^T \right)$ where either $\Omega_j = u, \Omega_j' = r$, or $\Omega_j = x, \Omega_j' = \alpha(x)$. We do a case distinction on the type of $\rho_j$.

(A1) $\rho_j = k(a, b)$. We do a case distinction whether $a = b$:
* Case $a = b$. By assumption we need to show $k(u, u) \overset{!}{=} k(r, r)$. Recall that the vertex features $F_\mu(u), F_\nu(r)$ encode $k(u, u)$ and $k(r, r)$ where $\mu, \nu \in \{1, 2\}$ (see Algorithm 1), respectively. Hence, the initial colors $\pi_u^0$ and $\pi_r^0$ also encode $k(u, u)$ and $k(r, r)$. As WL refines colorings and $\pi_u^T = \pi_r^T$ it holds that $\pi_u^0 = \pi_r^0$ which implies that $k(u, u) = k(r, r)$.
* Case $a \neq b$. Then $(b, a)$ forms an edge meaning we need to show that $k(u, x) \overset{!}{=} k(r, \alpha(x))$. Recall that the feature $F_\mu((x, u))$ encodes $k(u, x)$ and that feature $F_\nu((\alpha(x), r))$ encodes $k(r, \alpha(x))$ where, $\mu, \nu \in \{1, 2\}$ (Algorithm 1). From the definition of $\alpha$ it follows that the edges $(x, u)$ and $(\alpha(x), r)$ have the same features and thus that $k(u, x) = k(r, \alpha(x))$.

(A2) We do a case distinction:

* Case $F(u) \overset{!}{=} F(r)$. See proof of (A2) above.
* Case $F(x) \overset{!}{=} F(\alpha(x))$. We know that $\pi_x^T = \pi_{\alpha(x)}^T$. Recall that the object features are encoded in the vertex features and thus in the initial coloring $\pi_.^0$. Since WL refines colorings, it follows from $\pi_x^T = \pi_{\alpha(x)}^T$ that $\pi_x^0 = \pi_{\alpha(x)}^0$ which implies $F(x) = F(\alpha(x))$.
* Case $F((x,u))) \overset{!}{=} F((\alpha(x), r))$. As argued above we know that the edges $(x, u)$ and $(\alpha(x), r)$ have the same features which implies $F((x,u))) = F((\alpha(x), r))$.

(A3) $\rho_j \overset{!}{=} c_\sigma^T$. We do a case distinction on $\sigma$:

* Case $\sigma$ is the vertex to update i.e., $u$ and $r$. Then, we need to show that $c_u^T \overset{!}{=} c_r^T$. We know that $\pi_u^T = \pi_r^T$ and by the induction hypothesis it follows that $c_u^T = c_r^T$.
* Case $\sigma$ is the neighboring vertex i.e., $x$ and $\alpha(x)$. Then, we need to show that $c_x^T \overset{!}{=} c_{\alpha(x)}^T$. We combine $\pi_x^T = \pi_{\alpha(x)}^T$ with the induction hypothesis to obtain $c_x^T = c_{\alpha(x)}^T$.

Thus we conclude that $\rho_j(u, c_u^T, \Omega_j, c_x^T) = \rho_j \left( r, c_r^T, \Omega_j', c_{\alpha(x)}^T \right)$ which means that $\rho(u, c_u^T, x, c_x^T) = \rho \left( r, c_r^T, \alpha(x), c_{\alpha(x)}^T \right)$. This concludes the proof of the induction step.

$\square$

## C.3 Strongly Simulating GNNs

We prove a theorem that allows us to simulate a combination of multiple AMPs that differ only by function applications (S4). This allows us to simulate GNNs whose layers correspond to AMPs.

**Theorem C.2.** *Let $\phi_1, \ldots, \phi_l$ be $l \geq 1$ AMPs that only differ in function application (S4). Let $R = AME_\psi$ be a structure-to-graph encoding where $\psi$ is the AMP obtained by removing function applications from $\phi_1$. Then, it holds for every pair of relational structures $X, X'$ from the domain of $\phi_1, \ldots, \phi_l$ and every $l \geq t \geq 1$ that $t$ iterations of WL on $R(X)$ and $R(X')$ refines the coloring produced by $(\phi_1 \circ \ldots \circ \phi_t)$ on $X$ and $X'$.*

Note that it does not matter whether we construct $\psi$ by removing function applications from $\phi_1$ or any other AMP from $\phi_1, \ldots, \phi_l$ as these AMP only differ by function application.

*Proof.* Let $X = (U, \mathcal{N}_1, \ldots, \mathcal{N}_k, F)$ and $X' = (U', \mathcal{N}_1', \ldots, \mathcal{N}_k', F')$ be two arbitrary relational structure from the domain of $\phi_1 \circ \ldots \circ \phi_l$ and let $l \geq t \geq 1$. By Lemma C.1 any coloring produced by $\psi$ refines the corresponding coloring from every AMP from $\phi_1, \ldots, \phi_l$. Thus, $\psi^l(X, F|_U)$ refines $(\phi_1 \circ \ldots \circ \phi_l)(X, F|_U)$ and $\psi^l(X', F'|_{U'})$ refines $(\phi_1 \circ \ldots \circ \phi_l)(X', F'|_{U'})$. By Theorem 3.5 we can strongly simulate $\psi$ under $R$. Thus, the theorem follows. $\square$

## C.4 List of Strongly Simulatable Algorithms (Corollary 3.6)

We prove Corollary 3.6.

**Corollary C.3.** *The following GNNs and variants of WL can be strongly simulated:*

1. $VV_C$-GNN [Sato et al., 2019]
2. $k$-WL / GNN [Morris et al., 2019]
3. $\delta$-$k$-(L)WL / (L)GNN [Morris et al., 2020]
5. GSN-e and GSN-v [Bouritsas et al., 2022]
4. $(k, s)$-LWL / SpeqNet [Morris et al., 2022]
6. DS-WL / GNN [Bevilacqua et al., 2021]
7. $k$-OSWL / OSAN [Qian et al., 2022]
8. $M_k$ GNN [Papp and Wattenhofer, 2022]
9. GMP [Wijesinghe and Wang, 2022]
10. Shortest Path Networks [Abboud et al., 2022]
11. Generalized Distance WL [Zhang et al., 2023]

*Proof.* The proofs consist of showing that the update rule underlying the corresponding algorithms are AMP. By invoking Theorem 3.5 this proves that the algorithm is strongly simulatable. For

algorithms that have both a GNN and a WL variant we only prove this for the WL variant as for GNN the proof can be done analogously. Note that most variants of WL and GNNs contain the color of the vertex they want to update in the updated color. An example of this the color $c_s^{t-1}$ in the definition of $k$-WL: $c_s^t = \text{HASH}\left(c_s^{t-1}, (C_1^t(s), \ldots, C_k^t(s))\right)$ where $C_j^t(s) = \text{HASH}\left(\{\!\{c_{s'}^{t-1} \mid s' \in \mathcal{N}_j(s)\}\!\}\right)$. This color can be modeled by a color atom (A3). We will argue this again for the first two algorithms and will assume that this is obvious for the latter algorithms.

1. $VV_C$-**GNNs [Sato et al., 2019]:** Let $\Delta$ be the maximum degree of the vertices. Suppose we are given a port numbering with functions $p_{\text{tail}}(v, i) : V \times [\![\Delta]\!] \to V \cup \{-\}$ and $p_{\mathbf{n}} : V \times [\![\Delta]\!] \to [\![\Delta]\!] \cup \{-\}$.[4]

   The color update function for $VV_C$-GNNs is

   $$z_v^{l+1} = f_\theta^l\left(z_v^l, z_{p_{\text{tail}}(v,1)}^l, p_{\mathbf{n}}(v,1), z_{p_{\text{tail}}(v,2)}^l, p_{\mathbf{n}}(v,2), \ldots z_{p_{\text{tail}}(v,\Delta)}^l, p_{\mathbf{n}}(v,\Delta),\right).$$

   Let $i \in [\![\Delta]\!]$. The color update function applies a function $f_\theta^l$ (S4) to a tuple. We need to show that this tuple corresponds to tuple AMP construction (S3) by showing that each element of the tuple is AMP. The first element of the tuple is $z_v^l$ which corresponds to an AMP (S1) atom. Each $p_{\mathbf{n}}(v,i)$ can be seen as a constant atom (A1). Each $z_{p_{\text{tail}}(v,i)}^l$ can be seen as $\left\{z_w^l \mid w \in \{p_{\text{tail}}(v,i)\}\right\}$ when $p_{\text{tail}}(v,i) \neq -$ and the empty set $\emptyset$ otherwise which defines a neighborhood function. Thus, $\left(z_v^l, z_{p_{\text{tail}}(v,1)}^l, p_{\mathbf{n}}(v,1), z_{p_{\text{tail}}(v,2)}^l, p_{\mathbf{n}}(v,2), \ldots z_{p_{\text{tail}}(v,\Delta)}^l, p_{\mathbf{n}}(v,\Delta),\right)$ is an AMP tuple (S3) of a color atom (A3) $z_v^l$, constant atoms (A1) and aggregation AMPs (S2). Hence, the color update function is strongly simulatable.

2. $k$-**WL [Morris et al., 2019]:** Let $W = V(G)^k$ be the set of $k$-tuples of vertices for a given graph $G$. Then, $k$-WL can be written as $c_s^t = \text{HASH}\left(c_s^{t-1}, (C_1^t(s), \ldots, C_k^t(s))\right)$ where $C_j^t(s) = \text{HASH}\left(\{\!\{c_{s'}^{t-1} \mid s' \in \mathcal{N}_j(s)\}\!\}\right)$. Both hash functions are function application AMPs (S4) and and $c_s^{t-1}$ is a color atom AMP (S1). Here, $\mathcal{N}_j(s)$ encodes the $j$-neighborhood of the $k$-tuple $s$. This means that each $C_1^t(s)$ corresponds to an aggregation AMP (S2). Then, the combination into $(C_1^t(s), \ldots, C_k^t(s))$ is a tuple of AMPs (S3). Hence, the color update functions is a AMP implying that $k$-WL is strongly simulatable.

3. $\delta$-$k$-**(L)WL [Morris et al., 2020]:** We only show this for $\delta$-$k$-WL as $\delta$-$k$-LWL is a special case obtained by removing elements from the color update function of $\delta$-$k$-WL. Let $W = V(G)^k$ be the set of $k$-tuples of vertices for a given graph $G$. We use bold $\mathbf{v}$ to denote a $k$-tuple and $v$ to denote a vertex. For $j \in [\![k]\!]$, the function $\phi_j(\mathbf{v}, w)$ returns the $k$-tuple obtained by replacing the $j$-th vertex of $\mathbf{v}$ by $w$. The function $\text{adj}(\mathbf{v}, \mathbf{w})$ returns 1 if the two vertices that distinguish the $j$-neighbors $\mathbf{v}$ and $\mathbf{w}$ are neighbors in $G$ and 0 otherwise. Then, $\delta$-$k$-WL can be written as $c_{\mathbf{v}}^{t+1} = \left(c_{\mathbf{v}}^i, M_i^{\delta,\bar{\delta}}(\mathbf{v})\right)$ with

   $$M_i^{\delta,\bar{\delta}}(\mathbf{v}) = \left(\left\{\!\!\left\{\left(c_{\phi_1(\mathbf{v},w)}^i, \text{adj}(\mathbf{v}, \phi_1(\mathbf{v},w))\right) \mid w \in V(G)\right\}\!\!\right\}, \ldots,\right.$$
   $$\left.\left\{\!\!\left\{\left(c_{\phi_k(\mathbf{v},w)}^i, \text{adj}(\mathbf{v}, \phi_k(\mathbf{v},w))\right) \mid w \in V(G)\right\}\!\!\right\}\right).$$

   Let $j \in [\![k]\!]$ be arbitrary. Note, that $\text{adj}(\mathbf{v}, \phi_j(\mathbf{v}, w))$ is an edge constant atom $k(\mathbf{v}, \phi_j(\mathbf{v}, w))$ (A1) and that $c_{\phi_j(\mathbf{v},w)}^i$ is a color atom (A3) for object $\phi_j(\mathbf{v}, w)$. Thus, $\left(c_{\phi_j(\mathbf{v},w)}^i, \text{adj}(\mathbf{v}, \phi_j(\mathbf{v}, w))\right)$ is a tuple atom (A4). Furthermore, we can rewrite the aggregation by replacing $\phi_j(\mathbf{v}, w)$ with $\mathbf{w}$ as $\{\!\{\left(c_{\mathbf{w}}^i, k(\mathbf{v}, \mathbf{w})\right) \mid \mathbf{w} \in \mathcal{N}_j(\mathbf{v})\}\!\}$ where $\mathcal{N}_j(\mathbf{v}) = \{\phi_j(\mathbf{v}, w) \mid w \in V(G)\}$. Thus, each $\left\{\!\!\left\{\left(c_{\phi_j(\mathbf{v},w)}^i, \text{adj}(\mathbf{v}, \phi_j(\mathbf{v}, w))\right) \mid w \in V(G)\right\}\!\!\right\}$ is an aggregation AMP (S2) and $M_i^{\delta,\bar{\delta}}(\mathbf{v})$ is a tuple AMP (S3). Thus, the update rule of $\delta$-$k$-WL is strongly simulatable.

---

[4]We think that there is a small typographical error in these functions signatures as defined by Sato et al. [2019]. There, they define them as $p_{\text{tail}}(v, i) : V \times \Delta \to V \cup \{-\}$ and $p_{\mathbf{n}} : V \times \Delta \to \Delta \cup \{-\}$.

4. $(k, s)$**-LWL and** $(k, s)$**-SpeqNets [Morris et al., 2022]:** This algorithm is very similar to $\delta$-$k$-LWL and we use the same definition of $\phi$ as above. We prove this only for $(k, s)$-LWL as $(k, s)$-SpeqNets can be argued analogously. In what follows we use $\#\text{com}(G[\mathbf{v}])$ to denote the number of connected components in the graph induced by the $k$-tuple of nodes $\mathbf{v}$ of $G$ i.e., the subgraph of $G$ that contains only vertices from $\mathbf{v}$ and all edges from $E(G)$ incident to these vertices. For $k \geq 1$ and $1 \leq s \leq k$, let

$$V(G)_s^k = \left\{ \mathbf{v} \in V^k \mid \#\text{com}(G[\mathbf{v}]) \leq s \right\}.$$

Similarly to $\delta$-$k$-LWL, $(k, s)$-LWL assigns a color to $k$-tuples. However, $(k, s)$-LWL works on elements from $V(G)_s^k$. Hence, $W = V(G)_s^k$. The update function can the be defined as:

$$c_{\mathbf{v}}^{i+1} = \left( c_{\mathbf{v}}^i, M_i^{\delta, k, s}(\mathbf{v}) \right)$$

with

$$M_i^{\delta, k, s}(\mathbf{v}) = \left( \left\{\!\!\left\{ c_{\phi_1(\mathbf{v}, w)}^i \mid w \in \mathcal{N}(v_1) \text{ and } \phi_1(\mathbf{v}, w) \in V(G)_s^k \right\}\!\!\right\}, \dots, \right.$$

$$\left. \left\{\!\!\left\{ c_{\phi_k(\mathbf{v}, w)}^i \mid w \in \mathcal{N}(v_k) \text{ and } \phi_k(\mathbf{v}, w) \in V(G)_s^k \right\}\!\!\right\} \right)$$

where $v_i$ is the $i$-th element of $k$-tuple $\mathbf{v}$. Similar to above, we can rewrite the aggregation in all multisets. Let $i \in [\![k]\!]$, then we define $\mathcal{N}_i(v) = \left\{ \phi_i(\mathbf{v}, w)) \mid w \in \mathcal{N}(v_i) \text{ and } \phi_i(\mathbf{v}, w)) \in V(G)_s^k \right\}$, which allows us to rewrite the multisets as $\left\{\!\!\left\{ c_{\mathbf{x}}^i \mid \mathbf{x} \in \mathcal{N}_i(v) \right\}\!\!\right\}$. This implies that each of the multisets is an aggregation AMP (S2) and that $M_i^{\delta, k, s}(\mathbf{v})$ is a tuple AMP (S3). With this it follows that the color update function is simulatable.

5. **GSN-e and GSN-v [Bouritsas et al., 2022]:** We only prove this for GSN-v as GSN-e can be argued similarly. Let $W = V(G)$. The update function behind GNS-v is defined as

$$h_v^{t+1} = \text{UP}^{t+1} \left( h_v^t, m_v^{t+1} \right)$$

with

$$m_v^{t+1} = M^{t+1} \left( \left\{\!\!\left\{ \left( h_v^t, h_u^t, x_v^V, x_u^V, e_{u,v} \right) \mid u \in \mathcal{N}(v) \right\}\!\!\right\} \right).$$

In this definition $\text{UP}^{t+1}$ is an arbitrary function approximator such as an MLP and $M^{t+1}$ is a neighborhood aggregation function meaning meaning an arbitrary function that operates on multisets. We can see $M^{t+1}, \text{UP}^{t+1}$ as function applications (S4). Additionally, $x_v^V, x_u^V$ and $e_{u,v}$ are vectors encoding subgraph isomorphism counts with respect to some set of patterns. These isomorphism counts correspond to atom constants (A1): $k(v, v), k(u, u)$ and $k(u, v)$. As $h_v^t$ and $h_u^t$ are color atoms (A3) and $e_{u,v}$ is a constant atom (A1) it follows that $\left( h_v^t, h_u^t, x_v^V, x_u^V, e_{u,v} \right)$ is a tuple atom (A4) and $\left\{\!\!\left\{ \left( h_v^t, h_u^t, x_v^V, x_u^V, e_{u,v} \right) \mid u \in \mathcal{N}(v) \right\}\!\!\right\}$ is an aggregation AMP (S2). Applying the function $M^{t+1}$ (S4) to this AMP yields $m_v^{t+1}$. Thus, the update rule is strongly simulatable.

6. **DS-WL [Bevilacqua et al., 2021]:** For a given graph $G$ and a policy $\pi$ that compute subgraphs we define the set of objects $W$ as the disjoint union of the vertices in each subgraph $W = \dot{\bigcup}_{S \in \pi(G)} V(S)$. We identify each object from $W$ by its original vertex $v \in V$ and subgraph $S \in \pi(G)$ that created it. This means that $c_{v,S}$ is the color of vertex $v$ from subgraph $S$. Then, the update rule of DS-WL can be written as

$$c_{v,S}^{t+1} = \text{HASH} \left( c_{v,S}^t, \left\{\!\!\left\{ c_{x,S}^t \mid (x, S) \in \mathcal{N}(v, S) \right\}\!\!\right\} \right)$$

where $\mathcal{N}(v, S) = \{(x, S) \mid x \in \mathcal{N}_S(v)\}$ is the set of all neighbors of vertex $v$ in subgraph $S$. Hence, $\left\{\!\!\left\{ c_{x,S}^t \mid (x, S) \in \mathcal{N}(v, S) \right\}\!\!\right\}$ is an aggregation AMP (S2) which implies that the update rule is strongly simulatable.

7. $k$**-OSWL and** $k$**-OSANs [Qian et al., 2022]:** We only argue the case of $k$-OSWL as the proof works similarly for $k$-OSANs. In what follows we use $\mathbf{g} \in G_k$ to denote a $k$ tuple of vertices from $V(G)$ corresponding to $k$-vertex induced subgraph of $G$. We use $G_k$ to denote the set of all $k$-vertex induced subgraphs of $G$. The algorithm computes colors for each

combination of a vertex with $k$-vertex induced subgraph combination i.e. $W = V(G) \times G_k$. It uses the color update function:

$$C_{v,\mathbf{g}}^{i+1} = \text{RELABEL}\left(C_{v,\mathbf{g}}^i, \left\{\!\left\{C_{u,\mathbf{g}}^i \mid u \in \square\right\}\!\right\}\right).$$

Here, $\square$ is either $\mathcal{N}_G(v)$ the neighborhood relation in the graph $G$, or $V(G)$ the set of all vertices. Since both neighborhood allows us to define corresponding neighborhood functions it follows that $\left\{\!\left\{C_{u,\mathbf{g}}^i \mid u \in \square\right\}\!\right\}$ is an aggregation AMP (S2) and thus that the color update function is strongly simulatable.

8. $M_k$-**GNNs [Papp and Wattenhofer, 2022]:** As Papp and Wattenhofer [2022] note, their approach is related to ESAN [Bevilacqua et al., 2021]. For a graph $G$ with some set of marked nodes the following update rule that computes the colorings:

$$h_u^{t+1} = \text{UPDATE}\left(h_u^t, a_u^{t+1}\right)$$

with

$$a_u^{t+1} = \text{AGGR}_{\text{marked}}\left(\left\{\!\left\{h_v^t \mid v \in \mathcal{N}_M(u)\right\}\!\right\}\right) + \text{AGGR}_{\text{marked}}\left(\left\{\!\left\{h_v^t \mid v \in \mathcal{N}_U(u)\right\}\!\right\}\right).$$

Where $\mathcal{N}_M$ and $\mathcal{N}_U$ are the neighborhood relations referring to marked (unmarked) neighbors in the subgraphs, respectively. Thus, $\left\{\!\left\{h_v^t \mid v \in \mathcal{N}_M(u)\right\}\!\right\}$ and $\left\{\!\left\{h_v^t \mid v \in \mathcal{N}_U(u)\right\}\!\right\}$ are aggregation AMPs (S2). We can model $a_u^t$ as a function $\phi((x,y)) = \text{AGGR}_{\text{marked}}(x) + \text{AGGR}_{\text{marked}}(y)$ with $x = \left\{\!\left\{h_v^t \mid v \in \mathcal{N}_M(u)\right\}\!\right\}$ and $y = \left\{\!\left\{h_v^t \mid v \in \mathcal{N}_U(u)\right\}\!\right\}$. Hence, $a_u^t$ is a function AMP (S4) applied to a tuple AMP (S3). Thus, the color update function is strongly simulatable.

9. **GMP [Wijesinghe and Wang, 2022]:** The color update function underlying GMP has the shape

$$m_a^t = \text{AGGREGATE}^N\left(\left\{\!\left\{\left(\bar{A}_{vu}, h_u^t\right) \mid u \in \mathcal{N}(v)\right\}\!\right\}\right),$$
$$m_v^t = \text{AGGREGATE}^N\left(\left\{\!\left\{\bar{A}_{vu} \mid u \in \mathcal{N}(v)\right\}\!\right\}\right) h_v^t$$
$$c_v^{t+1} = \text{COMBINE}\left(m_v^t, m_a^t\right).$$

Here, $\bar{A}_{vu} \in \mathbb{R}$ are *structural coefficients* encoding local structures in the graph around $v, u$. Note that we can model $\text{AGGREGATE}^N\left(\left\{\!\left\{\bar{A}_{vu} \mid u \in \mathcal{N}(v)\right\}\!\right\}\right)$ as a constant atom (A1) $k(v,v)$. Thus, we can then write $m_v^t = k(v,v) \cdot h_v^t = f_{\text{mul}}(k(v,v), h_v^t)$ where $f_{\text{mul}}(x,y) = x \cdot y$. Hence, $m_v^t$ is a function application (S4) to a tuple (S3) of a constant node specific value (A1) together and a color (A3). Furthermore, $m_a^t$ is an aggregation (S2) of tuples (A4) of a constant value (A1) combined with a color (A3). Thus, the color update function is strongly simulatable.

10. **Shortest Path Networks [Abboud et al., 2022]:** Let $G = (V, E, F)$ be a graph, $v \in V$ a vertex and $i \geq 1$ an integer. We denote by $\mathcal{N}_i(v)$ the $i$-hop shortest path neighborhood i.e., the set of all vertices that can be reached from $v$ in a shortest path of length $i$. Then, the color update function of shortest path message passing graph neural networks (SP-MPNNs) are defined as

$$c_v^{t+1} = \text{COM}\left(c_v^t, \left(c_v^t, \text{AGG}_1\left(\left\{\!\left\{c_u^t \mid u \in \mathcal{N}_1(v)\right\}\!\right\}\right)\right), \ldots, \left(\text{AGG}_k\left(\left\{\!\left\{c_u^t \mid u \in \mathcal{N}_k(v)\right\}\!\right\}\right)\right)\right)$$

where $k \geq 1$ is an integer, COM is a combination function and $\text{AGG}_{\ldots}$ are functions that map a multisets of colors to a single color. Let $W = V$. Obviously, COM and all AGG functions are function applications (S4). It is clear that $c_v^t$ corresponds to a color atom (A3). Next we investigate the term $\left(c_v^t, \text{AGG}_j\left(\left\{\!\left\{c_u^t \mid u \in \mathcal{N}_j(v)\right\}\!\right\}\right)\right)$ where $j \in [\![k]\!]$. In this term, $\left\{\!\left\{c_u^t \mid u \in \mathcal{N}_j(v)\right\}\!\right\}$ is the aggregation AMP (S2) of color atoms (A3). Thus, $\left(c_v^t, \text{AGG}_j\left(\left\{\!\left\{c_u^t \mid u \in \mathcal{N}_j(v)\right\}\!\right\}\right)\right)$ is a tuple AMP (S3). It follows that the color update functions of SP-MPNNs corresponds to AMP which implies that SP-MPNN can be strongly simulated.

11. **Generalized Distance WL [Zhang et al., 2023]:** Note that this generalizes shortest path networks. Let $G = (V, E, F)$ be a graph. Let $d_G(\cdot, \cdot)$ be an arbitrary distance metric i.e., a function $d_G : V \times V \to \mathbb{R}$. The color update function of generalized distance Weisfeiler-Leman (GD-WL) is defined as $c_v^{t+1} = \text{HASH}\left(\left\{\!\left\{(d_G(u,v), c_u^t) \mid u \in V\right\}\!\right\}\right)$. Let $W = V$. It

follows that $\{\!\{(d_G(u,v), c_u^t) \mid u \in V\}\!\}$ is an aggregation AMP (S2) that aggregates a tuple atom (A4) $(d_G(u,v), c_u^t)$ built from a constant atom (A1) $d_G(u,v)$ and a color atom (A3) $c_u^t$. As HASH corresponds to a function application (S4) we conclude that GD-WL is AMP and can be strongly simulated.

$\square$

# D    Proofs for Limits of Strong Simulation (Section 4)

## D.1    Cannot Simulate Nested Aggregation (Theorem 4.1 Part 1)

We prove the correctness of Theorem 4.1 for the nested aggregations.

**Theorem 4.1.** *Nested aggregations and non-pairwise message passing cannot be strongly simulated.*

*Proof.* We define two relational structures $X_1, X_2$ that can be distinguished in the first round of message passing with nested aggregations, but cannot be distinguished in the first round of WL when representing the relational structures as graphs. Consider the set of objects $U = \{x, a, b, c, d, h, h'\}$ with neighborhood functions:

- $\mathcal{N}_1(a) = \mathcal{N}_1(b) = \mathcal{N}_1(c) = \mathcal{N}_1(d) = \{x\}$,

- $\mathcal{N}_2(x) = \{h, h'\}$,

- $\mathcal{N}_3(h) = \{a, b\}$,

- $\mathcal{N}_3(h') = \{c, d\}$.

In the above definition, the neighborhood function returns the empty set for all other objects. Both relational structures will have the same objects and neighborhood functions but different initial features:

- $F_1(a) = F_1(b) = \square$,
  $F_1(c) = F_1(d) = \bigcirc$,
  $F_1(x) = F_1(h) = F_1(h') = \triangle$,

- $F_2(a) = F_2(d) = \square$,
  $F_2(b) = F_2(c) = \bigcirc$, and
  $F_2(x) = F_2(h) = F_2(h') = \triangle$ .

We define the two relational structures as $X_1 = (U, \mathcal{N}_1, \mathcal{N}_2, \mathcal{N}_3, F_1)$ and $X_2 = (U, \mathcal{N}_1, \mathcal{N}_2, \mathcal{N}_3, F_2)$. We define a color update function with nested-aggregations over these relations structures:

$$c_v^{t+1} = \text{HASH}\left(c_v^t, \{\!\{c_w^t \mid w \in \mathcal{N}_1(v)\}\!\}, \{\!\{\{\!\{c_y^t \mid y \in \mathcal{N}_3(x)\}\!\} \mid x \in \mathcal{N}_2(v)\}\!\}\right),$$

where $c_v^t$ is the color of object $v$ in iteration $t$. We use $\pi_1, \pi_2$ to denote the color $c_x^1$ for $X_1$ and $X_2$, respectively. By definition we know that

$$\pi_1 = \text{HASH}\left(\triangle, \emptyset, \{\!\{\{\!\{\bigcirc, \bigcirc\}\!\}, \{\!\{\square, \square\}\!\}\}\!\}\right),$$

$$\pi_2 = \text{HASH}\left(\triangle, \emptyset, \{\!\{\{\!\{\square, \bigcirc\}\!\}, \{\!\{\bigcirc, \square\}\!\}\}\!\}\right).$$

Obviously $\pi_1 \neq \pi_2$. Suppose there exists a structure-to-graph encoding $R$ such that WL can distinguish the color of the vertex $x$ corresponding to object $x$ for the two graphs $R(X_1), R(X_2)$. We use $\tau_1, \tau_2$ to denote the color of vertex $x$ after one iteration of WL on $R(X_1), R(X_2)$, respectively. Since the color of $x$ depends on $a, b, c, d$, we can assume that there are edges from these vertices to $x$ in both graphs. Since $R$ is a structure-to-graph encoding it makes use of the mappings $R_{\text{graph}}$ and $R_{\text{feat}}$. By definition of $R_{\text{graph}}$ it has no access to the features thus the graph created for $X_1, X_2$ by $R_{\text{graph}}$ are identical. Hence, the graphs $R(X_1)$ and $R(X_2)$ only differ by the features assigned to $a, b, c, d$. By definition of WL, the colors $\tau_1, \tau_2$ only depend on the features of the neighbors of $x$ and are independent from any other vertices (or adjacencies between other vertices). Let $Y = y_1, \ldots, y_m$ be all neighbors of $x$ that are not $a, b, c, d$. Note that the features of the vertices of $Y$ is the same for

both graphs as all vertices from $Y$ are either not from $U$ or they are assigned the same color by $F_1$ and $F_2$. We add the relational structure $X_i$ (for $i \in \{1, 2\}$) as a subscript to the color notation to indicate where the color is originally coming from, e.g., $c_{x, X_1}^0$ is the initial color of vertex $x$ in graph $R(X_1)$. By definition of WL it holds that:

$$\tau_1 = \text{HASH}\left(c_{x, X_1}^0, \left\{\!\left\{c_{a, X_1}^0, c_{b, X_1}^0, c_{c, X_1}^0, c_{d, X_1}^0\right\}\!\right\} \cup \left\{\!\left\{c_{y, X_1}^0 \mid y \in Y\right\}\!\right\}\right),$$

$$\tau_2 = \text{HASH}\left(c_{x, X_2}^0, \left\{\!\left\{c_{a, X_2}^0, c_{b, X_2}^0, c_{c, X_2}^0, c_{d, X_2}^0\right\}\!\right\} \cup \left\{\!\left\{c_{y, X_2}^0 \mid y \in Y\right\}\!\right\}\right).$$

As $R_{\text{graph}}$ assigns the same features to $x$ and all vertices of $Y$ for the two graphs and $R_{\text{feat}}$ concatenates the vertex features with the features from the structures, it follows that $c_{x, X_1}^0 = c_{x, X_2}^0$ and $\left\{\!\left\{c_{y, X_1}^0 \mid y \in Y\right\}\!\right\} = \left\{\!\left\{c_{y, X_2}^0 \mid y \in Y\right\}\!\right\}$. As we intend to argue that there exists no $R$ such that $\tau_1 \neq \tau_2$ we intend to argue that

$$\left\{\!\left\{c_{a, X_1}^0, c_{b, X_1}^0, c_{c, X_1}^0, c_{d, X_1}^0\right\}\!\right\} \overset{!}{=} \left\{\!\left\{c_{a, X_2}^0, c_{b, X_2}^0, c_{c, X_2}^0, c_{d, X_2}^0\right\}\!\right\}.$$

By definition the color of a vertex $v \in \{a, b, c, d\}$ in graph $R(X_i)$ (with $i \in \{1, 2\}$) is the color assigned to $v$ by $R_{\text{graph}}(X_i)$:

$$c_{v, X_i}^0 = \text{concat}(F_{\text{graph}, i}(v), F_i(v)).$$

We argue that all $v \in \{a, b, c, d\}$ have the same feature $F_{\text{graph}, i}(v)$ for all $i \in \{1, 2\}$. First, any two $v, w \in \{a, b, c, d\}$ that are part of the same set in $\mathcal{N}_3$ ($\{a, b\}, \{c, d\}$) are necessarily assigned the same feature as they are indistinguishable for $R_{\text{graph}}$ based on the input $(U, \mathcal{N}_1, \mathcal{N}_2, \mathcal{N}_3)$. Second, for two $v, w \in \{a, b, c, d\}$ that are not part of the same set in $\mathcal{N}_3$ (e.g. $v = a, w = c$), $R_{\text{graph}}$ can detect that they are part of different sets but cannot encode this into the vertex features. Any such encoding would treat the sets $\{a, b\}$ and $\{c, d\}$ differently which is not possible since the function operates on (multi)sets and is thus permutation equivariant. Together with the fact that $\left\{\!\left\{F_1(a), F_1(b), F_1(c), F_1(d)\right\}\!\right\} = \left\{\!\left\{F_2(a), F_2(b), F_2(c), F_2(d)\right\}\!\right\}$ it follows that $\tau_1 = \tau_2$. This contradicts the initial assumption and proves that no structure-to-graph encoding $R$ exists that allows for strong simulation on $X_1$ and $X_2$. This proves the theorem. $\qquad\square$

### D.2 Non-pairwise message passing cannot be simulated (Theorem 4.1 Part 2)

We prove the correctness of Theorem 4.1 for non-pairwise message passing.

**Theorem 4.1.** *Nested aggregations and non-pairwise message passing cannot be strongly simulated.*

*Proof.* The proof is similar to the case of nested aggregation (Appendix D.1). We define two relational structures $X_1, X_2$ that can be distinguished in the first round of message passing with non-pairwise aggregations, but cannot be distinguished in the first round of WL when representing the relational structures as graphs. Consider the set of objects $U = \{x, a, b, c, d\}$ with neighborhood functions:

- $\mathcal{N}_1(a) = \mathcal{N}_1(b) = \mathcal{N}_1(c) = \mathcal{N}_1(d) = \{x\}$,

- $\mathcal{N}_2(x) = \{(a, b), (c, d)\}$.

In the above definition, the neighborhood function returns the empty set for all other objects. Both relational structures will have the same objects and neighborhood functions but different initial features:

- $F_1(a) = F_1(b) = \square$,
  $F_1(c) = F_1(d) = \bigcirc$,
  $F_1(x) = \triangle$,

- $F_2(a) = F_2(d) = \square$,
  $F_2(b) = F_2(c) = \bigcirc$, and
  $F_2(x) = \triangle$.

We define the two relational structures as $X_1 = (U, \mathcal{N}_1, \mathcal{N}_2, F_1)$ and $X_2 = (U, \mathcal{N}_1, \mathcal{N}_2, F_2)$. We define a color update function with non-pairwise aggregations over these relational structures:

$$c_v^{t+1} = \text{HASH}\left(c_v^t, \{\!\{c_w^t \mid w \in \mathcal{N}_1(v)\}\!\}, \{\!\{(c_x^t, c_y^t) \mid (x,y) \in \mathcal{N}_2(v)\}\!\}\right),$$

where $c_v^t$ is the color of object $v$ in iteration $t$. We use $\pi_1, \pi_2$ to denote the color $c_x^1$ for $X_1$ and $X_2$, respectively. By definition we know that

$$\pi_1 = \text{HASH}\left(\triangle, \emptyset, \{\!\{(\square, \square)(\bigcirc, \bigcirc)\}\!\}\right),$$

$$\pi_2 = \text{HASH}\left(\triangle, \emptyset, \{\!\{(\square, \bigcirc), (\bigcirc, \square)\}\!\}\right).$$

Obviously $\pi_1 \neq \pi_2$. Suppose there exists a structure-to-graph encoding $R$ such that WL can distinguish the color of the vertex $x$ corresponding to object $x$ for the two graphs $R(X_1), R(X_2)$. We use $\tau_1, \tau_2$ to denote the color of vertex $x$ after one iteration of WL on $R(X_1), R(X_2)$, respectively. Since the color of $x$ depends on $a, b, c, d$, we can assume that there are edges from these vertices to $x$ in both graphs. Since $R$ is a structure-to-graph encoding it makes use of the mappings $R_{\text{graph}}$ and $R_{\text{feat}}$. By definition of $R_{\text{graph}}$ it has no access to the features thus the graph created for $X_1, X_2$ by $R_{\text{graph}}$ are identical. Hence, the graphs $R(X_1)$ and $R(X_2)$ only differ by the features assigned to $a, b, c, d$. By definition of WL, the colors $\tau_1, \tau_2$ only depend on the features of the neighbors of $x$ and are independent from any other vertices (or adjacencies between other vertices). Let $Y = y_1, \ldots, y_m$ be all neighbors of $x$ that are not $a, b, c, d$. Note that the features of the vertices of $Y$ is the same for both graphs as all vertices from $Y$ are either not from $U$ or they are assigned the same color by $F_1$ and $F_2$. We add the relational structure $X_i$ (for $i \in \{1, 2\}$) as a subscript to the color notation to indicate where the color is originally coming from, e.g., $c_{x,X_1}^0$ is the initial color of vertex $x$ in graph $R(X_1)$. By definition of WL it holds that:

$$\tau_1 = \text{HASH}\left(c_{x,X_1}^0, \{\!\{c_{a,X_1}^0, c_{b,X_1}^0, c_{c,X_1}^0, c_{d,X_1}^0\}\!\} \cup \{\!\{c_{y,X_1}^0 \mid y \in Y\}\!\}\right),$$

$$\tau_2 = \text{HASH}\left(c_{x,X_2}^0, \{\!\{c_{a,X_2}^0, c_{b,X_2}^0, c_{c,X_2}^0, c_{d,X_2}^0\}\!\} \cup \{\!\{c_{y,X_2}^0 \mid y \in Y\}\!\}\right).$$

As $R_{\text{graph}}$ assigns the same features to $x$ and all vertices of $Y$ for the two graphs and $R_{\text{feat}}$ concatenate the vertex features with the features from the structures, it follows that $c_{x,X_1}^0 = c_{x,X_2}^0$ and $\{\!\{c_{y,X_1}^0 \mid y \in Y\}\!\} = \{\!\{c_{y,X_2}^0 \mid y \in Y\}\!\}$. As we intend to argue that there exists no $R$ such that $\tau_1 \neq \tau_2$ we intend to argue that

$$\{\!\{c_{a,X_1}^0, c_{b,X_1}^0, c_{c,X_1}^0, c_{d,X_1}^0\}\!\} \stackrel{!}{=} \{\!\{c_{a,X_2}^0, c_{b,X_2}^0, c_{c,X_2}^0, c_{d,X_2}^0\}\!\}.$$

By definition the color of a vertex $v$ from $a, b, c, d$ in graph $X_i$ from $X_1, X_2$ where $F_{\text{graph},i}(v)$ is the color assigned to $v$ by $R_{\text{graph}}(X_i)$:

$$c_{v,X_i}^0 = \text{concat}(F_{\text{graph},i}(v), F_i(v)).$$

We prove the the two multisets of color are equal by showing $c_{a,X_1}^0 = c_{a,X_2}^0$, $c_{b,X_1}^0 = c_{d,X_2}^0$, $c_{c,X_1}^0 = c_{c,X_2}^0$ and $c_{d,X_1}^0 = c_{b,X_2}^0$. This follows from the observation that two vertices $x \in V_{R(X_1)}$ and $y \in R_{R(X_2)}$ are indistinguishable by $R_{\text{graph}}$ if they are in the same position of a tuple in $\mathcal{N}_2$. For example, both $b$ and $d$ are in position 2 of their respective tuple. This means that $R_{\text{graph}}$ has to assign the same features to both $b$ and $d$ implying $F_{\text{graph},1}(b) = F_{\text{graph},2}(d)$. As $F_1(b) = F_2(d)$ it follows that $c_{b,X_1}^0 = c_{d,X_2}^0$. An analogous argument allows us to show the rest of the equalities which implies $\tau_1 = \tau_2$. This contradicts the initial assumption and shows that no structure-to-graph encoding $R$ exists that allows for strong simulation of non-pairwise message passing on $X_1$ and $X_2$. This proves the theorem. $\qquad\square$

# E   Proofs for Weak Simulation (Section 5)

## E.1   Weak simulation of nested aggregations and non-pairwise message passing (Theorem 5.2)

We prove the correctness of Theorem 5.2.

**Theorem 5.2.** *Nested aggregations and non-pairwise message passing can be weakly simulated with simulation factor 2.*

For this we define the class of *generalized augmented message passing* (gAMP) color update functions. This class contains augmented message passing (AMP) but also non-pairwise message passing and nested aggregations.

**Definition E.1.** (Generalized Augmented Message Passing. Let $X = (U, \mathcal{N}_1, \ldots, \mathcal{N}_k, F)$ be a relational structure. We call a color update function $\phi$ *generalized augmented message passing* (gAMP) if for all $u \in U$ and coloring $c$ of $X$, it can recursively be defined as exactly one of the following five forms:

(W1) $\phi(u, X, c) = \psi(u, (U, \mathcal{M}_1, \ldots, \mathcal{M}_r, F), c)$ where $\psi$ is AMP and $\{\mathcal{M}_1, \ldots, \mathcal{M}_r\}$ is a subset of $\{\mathcal{N}_1, \ldots, \mathcal{N}_k\}$ only containing neighborhood functions $\mathcal{N}$ where $\ell(\mathcal{N}) = 1$.

(W2) $\phi(u, X, c) = \{\!\{(c_{u_1}, \ldots, c_{u_m}) \mid (u_1, \ldots, u_m) \in \mathcal{N}_i(u)\}\!\}$ where $i \in [\![k]\!]$. This is called *non-pairwise message passing*.

(W3) $\phi(u, X, c) = \{\!\{\psi(x, (U, \mathcal{M}_1, \ldots \mathcal{M}_r, F), c) \mid x \in \mathcal{N}_i(u)\}\!\}$ where $\psi$ is AMP that contains a neighborhood aggregation (S2), $i \in [\![k]\!]$ and $\{\mathcal{M}_1, \ldots, \mathcal{M}_r\}$ is a subset of $\{\mathcal{N}_1, \ldots, \mathcal{N}_k\}$ only containing neighborhood functions $\mathcal{N}$ where $\ell(\mathcal{N}) = 1$. This construction is called a *nested aggregation*.

(W4) $\phi(u, X, c) = (\phi_1(u, X, c), \ldots, \phi_n(u, X, c)$ where $\phi_1, \ldots, \phi_n$ are gAMP.

(W5) $\phi(u, X, c) = f(\psi(u, X, c))$ where $f$ maps colors to colors and $\psi$ is gAMP.

We want to prove that we can weakly simulate all coloring algorithms in this class. For this we define a structure-to-graph encoding in Algorithm 6 for every color update function in this class. We use the same encoding function ENC as defined in Appendix C. We assume that if a feature function is called with an vertex, object or edge that is assigned no features, it returns a fixed default color. We prove that combining Algorithm 6 with WL is at least as expressive as the corresponding non-standard message passing algorithm.

*Proof.* (Theorem 5.2) This proof works similarly to the proof of Theorem 3.5. We prove weak simulatability for $\zeta = 2$. Let $\phi$ be an arbitrary generalized augmented message passing algorithm. We consider two arbitrary relational structures from the domain of $\phi$, say $X = (U, \mathcal{N}_1, \ldots, \mathcal{N}_k, F)$ and $X' = (U', \mathcal{N}_1', \ldots, \mathcal{N}_k', F')$. Let $(V_1, E_1, F_1) = R(X)$ and $(V_2, E_2, F_2) = R(X')$. We prove (1) that $\text{gAME}_\phi$ is a structure-to-graph encoding that runs in $\mathcal{O}(\tau_\phi(|U|))$; (2) for every $t \geq 0$ the coloring $\text{WL}^{2t}(R(X), F_1|_{V_1}) \cup \text{WL}^{2t}(R(X'), F_2|_{V_2})$ on $\text{gAME}_\phi(X)$ refines the coloring $\phi^t(X, F|_U) \cup \phi^t(X', F'|_{U'})$.

We begin by proving (1). Observe that Algorithm 6 can be split into $R_{\text{graph}}$ and $R_{\text{feat}}$ as required by the definition of structure-to-graph encoding. Furthermore, Algorithm 6 first iterates through all gAMPs in $\phi$ and all possible objects that are relevant to these gAMPS. Hence, Algorithm 6 runs in the same runtime as one iteration of gAMP which is in $\mathcal{O}(\tau_\phi(|U|))$.

Next, we prove (2). We use $\pi_u^t, \pi_r^t$ and $c_u^t, c_r^t$ to denote the colors assigned to vertex / object $u, r \in (U \cup U')$ in iteration $t$ by WL on $R(X), R(X')$ or $\phi$ on $X, X'$, respectively. Let $u, r \in (U \cup U')$ be arbitrary objects / vertices. We show by induction that if WL cannot distinguish $u$ and $r$ in iteration $2t$ then $\phi$ cannot distinguish them in iteration $t$. Formally, we show that $\pi_u^{2t} = \pi_r^{2t} \Rightarrow c_u^t = c_r^t$ which implies that $\text{WL}^{2t}(R(X), F_1|_{V_1}) \cup \text{WL}^{2t}(R(X'), F_2|_{V_2})$ refines $\phi^t(X, F|_U) \cup \phi^t(X', F'|_{U'})$ as $(U \cup U') \subseteq (V_1 \cup V_2)$ (by Algorithm 5).

Algorithm 6 starts by replacing all function applications (S4), (W5) by the identity function and flattens all tuples (A4), (S3), (W4). Recall that this transformation is either injective or bijective. By applying Lemma C.1 they do not influence the strong simulation property and thus also do not influence the weak simulation property. From now on we assume that $\phi = (\phi_1, \ldots, \phi_n)$ such that all $\phi_j$ with $j \in [\![n]\!]$ are either (W1) (which means they are one of (A1), (A2) (A3), (S2)), (W2), or (W3) and if they are an atom, then the first and third function argument are identical. We use $w$ to denote the object the gAMP is defined to update. Algorithm 3 iterates through all $\phi_1, \ldots, \phi_n$ and encodes them in the graph.

**Base case:** We show that $\pi_u^0 = \pi_r^0 \Rightarrow c_u^0 = c_r^0$. For this, we assume that $\pi_u^0 = \pi_r^0$. As the initial colors correspond to vertex or object features and Algorithm 6 encodes the object features in the vertex features it follows that $c_u^0 = c_r^0$.

---

**Algorithm 5** $R_{\text{graph}}$ for generalized augmented message encoding

---

**Input:** relational structure $X = (U, \mathcal{N}_1, \ldots, \mathcal{N}_k)$ without features, gAMP color update function $\phi$ that has the shape $(\phi_1, \ldots, \phi_n)$ where each $\phi_i$ with $i \in [\![n]\!]$ is one of (A1), (A2), (A3), (S2), (W2) or (W3).
**Output:** graph $G = (V, E, F_{\text{graph}})$

$V := W$
$E := \emptyset$
$F_{\text{graph}} :=$ empty features
**for** $i \in [\![n]\!]$ **do**
  **if** $\phi_i$ is (A1), (A2) or (A3) **then**
    **for** $u \in U$ **do**
      $(V, E, F_{\text{graph}}) :=$ Algorithm $1(\phi_i, (V, E, F_{\text{graph}}), u, u, i, 0, 0)$
    **end for**
  **else if** $\phi_i$ is (S2) which aggregates on neighborhood $\mathcal{N}$ **then**
    $(V, E, F_{\text{graph}}) :=$ Algorithm $2(\phi_i, (V, E, F_{\text{graph}}, (U, \mathcal{N}), i, 0)$
  **else if** $\phi_i$ is (W2) **then**
    /* $\phi_i$ has the shape $\{\!\{(c_{u_1}^t, \ldots, c_{u_m}^t) \mid (u_1, \ldots, u_m) \in \mathcal{N}_j(u)\}\!\}$ where $j \in [\![k]\!]$ */
    **for** $u \in U$ **do**
      **for** $(u_1, \ldots, u_m) \in \mathcal{N}_j(u)$ **do**
        $V := V \cup \{u_{(u_1,\ldots,u_m)}\}$
        $E := E \cup \{(u_{(u_1,\ldots,u_m)}, u)\}$
        $F(u_{(u_1,\ldots,u_m)}) :=$ ENC(empty features, $i, 0, 0, 1)$
        $F(u_{u_1,\ldots,u_m}, u) :=$ ENC(empty features, $i, 0, 0, 1)$
        **for** $r \in [\![m]\!]$ **do**
          $E := E \cup \{(u_r, u_{u_1,\ldots,u_m})\}$
          $F(u_r, u_{u_1,\ldots,u_m}) :=$ ENC(empty features, $i, r, 0, 1)$
        **end for**
      **end for**
    **end for**
  **else if** $\phi_i$ is (W3) **then**
    /* $\phi_i$ has shape $\{\!\{\psi(x, (U, \mathcal{M}_1, \ldots \mathcal{M}_r, F), c) \mid x \in \mathcal{N}_j(u)\}\!\}$ where $\psi$ is AMP */
    Replace function applications (S4) in $\psi$ by the identity function, i.e., $f(\varphi)$ becomes $\varphi$.
    Flatten tuples (A4), (S3) in $\psi$ except outermost, i.e., $(\ldots, (\varphi_1, \ldots, \varphi_r), \ldots)$ becomes
$(\ldots, \varphi_1, \ldots, \varphi_r, \ldots)$.
    /* $\psi$ has shape $(\psi_1, \ldots, \psi_n)$ where each $\psi_i$ with $i \in [\![n]\!]$ is one of (A1), (A2), (A3) or (S2) */
    $(V, E, F_{\text{graph}}) :=$ Algorithm $3(\psi, (V, E, F_{\text{graph}}), (U, \mathcal{M}_1, \ldots \mathcal{M}_r), i)$
    **for** $u \in U$ **do**
      **for** $x \in \mathcal{N}_j(u)$ **do**
        $E := E \cup \{(x, u)\}$
        $F(x, u) :=$ ENC(empty features, $i, 0, 0, 1)$
      **end for**
    **end for**
  **end if**
**end for**
**return** $(V, E, F_{\text{graph}})$

---

**Induction hypothesis:** We assume that $\pi_u^{2t} = \pi_r^{2t} \Rightarrow c_u^t = c_r^t$ holds for all $t \leq T$.

**Induction step:** We show that $\pi_u^{2(T+1)} = \pi_r^{2(T+1)} \Rightarrow c_u^{T+1} = c_r^{T+1}$. We assume $\pi_u^{2(T+1)} = \pi_r^{2(T+1)}$ which is equivalent to $\pi_u^{2T+2} = \pi_r^{2T+2}$. We prove for an arbitrary gAMP from $\phi_1, \ldots, \phi_n$ that it returns the same result for $u$ and $r$ which implies $c_u^{T+1} = c_r^{T+1}$. We do a case distinction on the type of this gAMP. Note that in gAMP and AMP the object / edge features are referred to us $F(\cdot)$ whereas in the relational structures $X$ and $X'$ they are denoted as $F(\cdot)$ and $F'(\cdot)$. Below, when we argue that two AMP constructions are equal and use the $F(\cdot)$ notation and neglect $F'(\cdot)$ for the sake of simplicity.

---

**Algorithm 6** Generalized augmented message encoding (gAME)

---

**Input:** relational structure $X = (U, \mathcal{N}_1, \ldots, \mathcal{N}_k, F)$, gAMP color update function $\phi$.
**Output:** graph $G = (V, E, F')$

Replace function applications (W5), (S4) in $\phi$ by the identity function, i.e., $f(\phi)$ becomes $\phi$.
Flatten all (W4), (A4), (S3) constructions in $\phi$ except the outermost, i.e., $(\ldots, (\phi_1, \ldots, \phi_n), \ldots)$
becomes $(\ldots, \phi_1, \ldots, \phi_n, \ldots)$.
/* $\phi$ has the shape $(\phi_1, \ldots, \phi_n)$ where each $\phi_i$ with $i \in [\![n]\!]$ is one of (A1), (A2), (A3), (S2), (W2)
or (W3) */

/* $R_{\text{graph}}$ */
$(V, E, F_{\text{graph}}) := $ Algorithm 5$((U, \mathcal{N}_1, \ldots, \mathcal{N}_k), \phi)$

/* $R_{\text{feat}}$ */
$F' :=$ empty features
**for** $v \in V$ **do**
    $F'(v) :=$ concat $(F(v), F_{\text{graph}}(v))$
**end for**
**for** $e \in E$ **do**
    $F'(e) :=$ concat $(F(e), F_{\text{graph}}(e))$
**end for**
**return** $(V, E, F')$

---

(W1) $\left( \phi_s(u, X, c) \stackrel{!}{=} \phi_s(r, X, c) \right)$. Then, $\phi_s$ is one of (A1), (A2), (A3), (S2). Observe that, then
Algorithm 6 performs the same operations as Algorithm 4: for atoms (A1), (A2), (A3) it
instantiates Algorithm 1 and for (S2) it instantiates Algorithm 2. We apply the induction step
of the proof of Theorem 3.5 in a slightly altered form: instead of proving $\pi_u^{T+1} = \pi_r^{T+1} \Rightarrow$
$c_u^{T+1} = c_r^{T+1}$ we instead prove $\pi_u^{2T+1} = \pi_r^{2T+1} \Rightarrow c_u^T = c_r^T$. Note, that by the induction
hypothesis we already *know* the stronger fact $\pi_u^{2T+2} = \pi_r^{2T+2}$. However, we *only use* the
fact $\pi_u^{2T+1} = \pi_r^{2T+1}$ so that we can later apply this argument in a context where only the
weaker $\pi_u^{2T+1} = \pi_r^{2T+1}$ holds. We obtain a proof for $\pi_u^{2T+1} = \pi_r^{2T+1} \Rightarrow c_u^T = c_r^T$ from
the induction step in the proof of Theorem 3.5 by replacing $\pi^T$ and $\pi^{T+1}$ by $\pi^{2T}$ and $\pi^{2T+1}$,
respectively. Thus, we know that for AMP $\phi_s$ it holds that $\phi_s(u, X, c) = \phi_s(r, X, c)$.

(W2) $(\{\!\{(c_{u_1}^T, \ldots, c_{u_m}^T) \mid (u_1, \ldots, u_m) \in \mathcal{N}_j(u)\}\!\} \stackrel{!}{=} \{\!\{(c_{r_1}^T, \ldots, c_{r_m}^T) \mid (r_1, \ldots, r_m) \in \mathcal{N}_j(r)\}\!\})$.
From $\pi_u^{2T+2} = \pi_r^{2T+2}$ it follows that there exists a bijective function $\alpha : \mathcal{N}_G(u) \to \mathcal{N}_G(r)$
that maps vertices assigned the same color by $\pi^{2T+1}$ to another. Note, that $\mathcal{N}_G(u)$
is the set of all neighboring vertices of vertex $u$ in the graph generated from $X$ or
$X'$. By the construction of the edge features, it follows that $\alpha$ maps vertices from
$\{u_{u_1, \ldots, u_m} \mid (u_1, \ldots, u_m) \in \mathcal{N}_j(u)\}$ to $\{r_{r_1, \ldots, r_m} \mid (r_1, \ldots, r_m) \in \mathcal{N}_j(r)\}$. Let $\beta$ be the
restriction of the function $\alpha$ to these two sets as its domain and image, respectively. This
implies that

$$\left\{\!\!\left\{ \pi_{u_{u_1, \ldots, u_m}}^{2T+1} \mid (u_1, \ldots, u_m) \in \mathcal{N}_j(u) \right\}\!\!\right\} = \left\{\!\!\left\{ \pi_{r_{r_1, \ldots, r_m}}^{2T+1} \mid (r_1, \ldots, r_m) \in \mathcal{N}_j(r) \right\}\!\!\right\}.$$

Let $x \in \{u_{u_1, \ldots, u_m} \mid (u_1, \ldots, u_m) \in \mathcal{N}_j(u)\}$ be arbitrary with $x = u_{u_1, \ldots, u_m}$. Then, we
call $\beta(x) = r_{r_1, \ldots, r_m}$. Note that all neighbors that send messages to $x$ are $u_1, \ldots, u_m$.
Similarly, all neighbors of $\beta(x)$ that send messages to $\beta(x)$ are $r_1, \ldots, r_m$. Since $\pi_x^{2T+1} = $
$\pi_{\beta(x)}^{2T+1}$ it holds that there exists a bijective function $\gamma(x) : \{u_1, \ldots, u_m\} \to \{r_1, \ldots, r_m\}$
mapping vertices assigned the same color by $\pi^{2T}$ to each other. Note for every $i \in [\![m]\!]$ the
edge connecting $u_i$ to $x$ is labeled with $i$ and the same holds for $r_i$ and $\beta(x)$. Thus, $\gamma(u_i) = $
$r_i$ which implies $(u_1^{2T}, \ldots, u_m^{2T}) = (r_1^{2T}, \ldots, r_m^{2T})$. From the induction hypothesis it holds
that $(c_{u_1}^T, \ldots c_{u_m}^T) = (c_{r_1}^T, \ldots c_{r_m}^T)$. From the fact that $\beta$ is a bijection and by iterating over
all possible vertices for $x$ be we obtain

$$\{\!\{(c_{u_1}^T, \ldots, c_{u_m}^T) \mid (u_1, \ldots, u_m) \in \mathcal{N}_j(u)\}\!\} = \{\!\{(c_{r_1}^T, \ldots, c_{r_m}^T) \mid (r_1, \ldots, r_m) \in \mathcal{N}_j(r)\}\!\}.$$

 $(\{\!\{\theta(x, X', c^T) \mid x \in \mathcal{N}_j(u)\}\!\} \overset{!}{=} \{\!\{\theta(x, X'c^T) \mid x \in \mathcal{N}_j(r)\}\!\})$ where $\theta$ is AMP, $X' = (U, \mathcal{M}_1, \ldots, \mathcal{M}_r, F)$ and all neighborhoods of $X'$ have $\ell(\cdot) = 1$. We use the function $\alpha$ defined above. By definition all vertices from $\mathcal{N}_j(u)$ and $\mathcal{N}_j(r)$ are connected to $u$ and $r$ by an edge with a special edge label encoding $j$. Thus, there exists a restriction from a bijective $\alpha$ to a bijective $\beta : \mathcal{N}_j(u) \to \mathcal{N}_j(r)$ such that for every $x \in \mathcal{N}_j(u)$ it holds that $\pi_x^{2T+1} = \pi_{\beta(x)}^{2T+1}$. Next, we can apply the above induction step proof for a gAMP that correspond to AMP (W1). From this proof it follows that for every $x \in \mathcal{N}_j(u)$ it holds that $\theta(x, X', c^T) = \theta(\beta(x), X', c^T)$. Together with the fact that $\beta$ is bijective we obtain that

$$(\{\!\{\theta(x, X', c^T) \mid x \in \mathcal{N}_j(u)\}\!\} = \{\!\{\theta(x, X', c^T) \mid x \in \mathcal{N}_j(r)\}\!\})$$

which contradicts the initial assumption.

This concludes the proof. $\qquad\square$

## E.2 List of Weakly Simulatable Algorithms (Corollary 5.3)

Next, we leverage Theorem 5.2 to prove Corollary 5.3.

**Corollary E.2.** *The following GNNs and variants of WL can be weakly simulated:*

1. Message Passing Simplicial Networks [Bodnar et al., 2021b]
2. CW Networks [Bodnar et al., 2021a]
3. DSS [Bevilacqua et al., 2021]
4. K-hop message passing and KP-GNNs [Feng et al., 2022]

*Proof.* Note that this has already been proven for Message Passing Simplicial Networks and CW Networks in Theorem F.1. We start by proving the corollary for DSS [Bevilacqua et al., 2021] by showing that its color update rule performs gAMP. We reuse the notation from the proof of Corollary 3.6. The update rule for DSS-WL can be written as

$$c_{v,S}^{t+1} = \text{HASH}\bigg( c_{v,S}^t, \{\!\{ c_{x,S}^t \mid x \in \mathcal{N}_S(v) \}\!\}, C_v^t, \{\!\{ C_x^t \mid x \in \mathcal{N}_G(v) \}\!\} \bigg)$$

where $\mathcal{N}_S(v)$ is the set of all neighbors of vertex $v$ in subgraph $S$, $\mathcal{N}_G(v)$ is the set of all neighbors of $v$ in the original graph, and $C_v^t$ contains the colors of $v$ across different subgraphs

$$C_v^t = \{\!\{ c_{v,S'}^t \mid S' \in \pi(G) \text{ and } v \in V(S') \}\!\} .$$

We define the set of objects $W$ as the disjoint union of all vertices in subgraphs $W = \dot{\bigcup}_{S \in \pi(G)} V(S)$ and identify each object from $W$ by its original vertex and the subgraph it created. The update-rule applies a HASH function (W5) to a tuple (W4) of other gAMPs. The first element of the tuple is $c_{v,S}^t$ which is a color atom (A3). The second element is an aggregation over neighbors (S2).[5] The third element can also be thought as an aggregation over neighbors (S2) $C_v^t = \{\!\{ c_{v,S'}^t \mid S' \in \pi(G) \text{ and } v \in V(S') \}\!\} = \{\!\{ c_{v,S'}^t \mid S' \in \mathcal{N}_S'(v, S) \}\!\}$ where $\mathcal{N}_S'(v, S) = \{ S' \mid S' \in \pi(G) \text{ and } v \in V(S') \}$. Finally, the fourth element of the tuple is a nested aggregation (W3) that aggregates over the AMP defined in the previous sentence (with $x$ instead of $v$) for all $x \in \mathcal{N}_G(v)$. Thus, the color update function performs gAMP and DSS can be weakly simulated.

---

[5]There is some more subtlety to this. The notation $\mathcal{N}_S$ implies a separate neighborhood function $\mathcal{N}_S$ for each subgraph $S$ which would give an unbounded number of different neighborhood functions which does not directly work with our definitions. Instead, we need to create a new neighborhood function $\mathcal{N}_1((v, s)) = \{ x \mid x \in \mathcal{N}_S(v) \}$. The same holds for the following definition of $\mathcal{N}_S'$.

Next, we prove the corollary for KP-GNNs [Feng et al., 2022], $K$-hop message passing [Feng et al., 2022] will follow as a special case of this. The color update function of KP-GNNs is defined as

$$m_v^{l+1,k} = \text{MES}_k^{l,\text{normal}}\bigg( \Big\{\!\!\Big\{ (h_u^l, e_{uv}) \mid u \in Q_{v,G}^{k,t} \Big\}\!\!\Big\} \bigg) + f(G_{v,G}^{k,t}),$$

$$f(G_{v,G}^{k,t}) = \text{EMB}((E(Q_{v,G}^{k,t}), C_k^{k'})),$$

$$h_v^{l+1,k} = \text{UPD}_k^l\bigg( m_v^{l+1,k}, h_v^l \bigg),$$

$$h_v^{l+1} = \text{COMBINE}^l \left( \Big\{\!\!\Big\{ h_v^{l+1,k} \mid k = 1, 2, \ldots, K \Big\}\!\!\Big\} \right).$$

Here, $t \in \{\text{gd}, \text{spd}\}$ with gd = graph diffusion and spd = shortest path distance. The set $Q_{v,G}^{k,\text{gd}}$ contains all nodes from $V(G)$ that can be reached within $k$-steps of random walk diffusion steps from $v$. The set $Q_{v,G}^{k,\text{spd}}$ contains all nodes from $V(G)$ whose shortest-path to node $v$ is exactly $k$.

Let $W = V(G) \cup \{v_k \mid v \in V(G), k \in [\![K]\!]\}$ be a set of objects with associated features $F$. We define three neighborhood functions $\mathcal{N}_1, \mathcal{N}_2, \mathcal{N}_3$:

1. $\mathcal{N}_1(v) = \{v_k \mid k \in [\![K]\!]\}$,

2. $\mathcal{N}_2(v_k) = \{v\}$,

3. $\mathcal{N}_3(v_k) = Q_{v,G}^{k,t}$.

For all other vertices the neighborhood functions return the empty set. We prove that $h_v^{l+1}$ performs gAMP it inside out, starting with $m_v^{l+1,k}$. We represent $m_v^{l+1,k}$ as AMP $\phi(v_k, (W, \mathcal{N}_1, \mathcal{N}_2, \mathcal{N}_3, F), h^l)$.[6] Let $\rho(v_k, h_{v_k}^l, u, h_u^l) = (h_u^l, e_{uv_k})$ where $e_{uv_k}$ is an edge constant depending on $u, v_k$. By definition $\rho$ is an atom that is constructed by combining a color atom (A3) with a constant atom (A1) into a tuple atom (A4). Note that $\text{MES}_k^{l,\text{normal}}\bigg( \Big\{\!\!\Big\{ (h_u^l, e_{uv}) \mid$

$u \in Q_{v,G}^{k,t} \Big\}\!\!\Big\} \bigg)$ is AMP that applies the function $\text{MES}_k^{l,\text{normal}}$ (S4) to an AMP that aggregates (S2) the atom $\rho(v, h_v^l, u, h_u^l)$ over all $u \in \mathcal{N}_3(v_k)$. Next, $f(G_{v,G}^{k,t})$ can be seen as a constant atom (A1) $k(v_k, v_k) = f(G_{v,G}^{k,t})$. Let $\varphi, \phi$ be two AMPs, then we define the addition AMP as $f_+((\varphi, \phi)) = \varphi + \phi$. Hence, we can represent $m_v^{l+1,k}$ as an AMP

$$\phi(v_k, (W, \mathcal{N}_1, \mathcal{N}_2, \mathcal{N}_3, F), h^l) = f_+\left( \text{MES}_k^{l,\text{normal}}\bigg( \Big\{\!\!\Big\{ (h_u^l, e_{uv}) \mid u \in Q_{v,G}^{k,t} \Big\}\!\!\Big\} \bigg), f(G_{v,G}^{k,t}) \right).$$

Next, we argue that $h_v^{l+1,k}$ is AMP represented as $\varphi(v_k, (W, \mathcal{N}_1, \mathcal{N}_2, \mathcal{N}_3, F), h^l)$. Observe, that we can write $\varphi$ as a function application of $\text{UPD}_k^l$ to a tuple

$$\big( \phi(v_k, (W, \mathcal{N}_1, \mathcal{N}_2, \mathcal{N}_3, F), \{ h_x^l \mid x \in \mathcal{N}_2(v_k) \} \big)$$

where $\{ h_x^l \mid x \in \mathcal{N}_2(v_k) \}$ is AMP (S2) that aggregates over the only $\mathcal{N}_2$ neighbor. As $\mathcal{N}_2(v_k) = \{v\}$ it holds that $\varphi(v_k, (W, \mathcal{N}_1, \mathcal{N}_2, \mathcal{N}_3, F)$ corresponds to $h_v^{l+1,k}$. Finally, we argue that $h_v^{l+1}$ is gAMP $\psi(v, (W, \mathcal{N}_1, \mathcal{N}_2, \mathcal{N}_3, F)$. We write $\psi$ as a function application (W5) of $\text{COMBINE}^l$ to the nested aggregation (W3)

$$\big\{\!\!\big\{ \varphi(v_k, h^l, W, \mathcal{N}_1, \mathcal{N}_2, \mathcal{N}_3) \mid v_k \in \mathcal{N}_1(v) \big\}\!\!\big\}.$$

Thus, $\psi(v, (W, \mathcal{N}_1, \mathcal{N}_2, \mathcal{N}_3, F)$ corresponds to $h_v^{l+1}$. As $h_v^{l+1}$ is a gAMP it follows by Theorem 5.2 that KP-GNN can be weakly simulated.

We obtain a proof that $K$-hop message passing [Feng et al., 2022] can be weakly simulated as special case of the above proof by removing the term $f(G_{v,G}^{k,t})$ from the definition of $m_v^{l+1,k}$. $\qquad\square$

---

[6]Note that atoms, AMP and gAMP are generally defined over all objects. However, by construction this AMP will only be used to aggregate colors of objects $v_k$ with $v \in V(G), k \in [\![K]\!]$. Thus, we denote it only for these objects despite it being defined over all objects. We use the similar simplifications throughout this proof.

# F Handcrafted Efficient Graph Transformations

## F.1 $\overline{\text{CWN}}$: Efficient Weak Simulation of CW Networks

We define a transformation $\overline{\text{CWN}}$ that allows for more efficient weak simulation of CW Networks. We prove the following theorem:

**Theorem F.1.** *Message Passing Simplicial Networks [Bodnar et al., 2021b] and CW Networks [Bodnar et al., 2021a] can be weakly simulated by $\overline{\text{CWN}}$ with a simulation factor of 2 such that the number of vertices is equal to the number of cells.*

*Proof.* We prove this theorem for CW Networks as they a generalization of Message Passing Simplicial Networks. We begin with definitions required for the proof. A regular cell complexes consists of different cells each with a fixed dimension. We have a boundary relation $\prec$ given on the cells. For a cell $\sigma$ we call $\mathcal{B}(\sigma) = \{\tau \mid \tau \prec \sigma\}$ its boundary adjacent cells and $\mathcal{C}(\sigma) = \{\tau \mid \sigma \prec \tau\}$ its co-boundary adjacent cells. We define $\mathcal{C}(\sigma, \mu) = \mathcal{C}_\sigma \cap \mathcal{C}_\mu$. For a cell $\sigma$ we call $\mathcal{N}_\uparrow(\sigma) = \{\tau \mid \exists \delta \text{ such that } \sigma \prec \delta \text{ and } \tau \prec \delta\}$ its upper adjacent cells. For a color $c^t$ at iteration $t$ we use the notation $c_\mathcal{F}^t(\sigma)$ to denote the color of all objects in $\mathcal{F}(\sigma)$ at iteration $t$ meaning $c_\mathcal{F}^t(\sigma) = \{\!\{c_x^t \mid x \in \mathcal{F}(\sigma)\}\!\}$. For a given cell $\sigma$ in a regular cell complex $X$ a CW Network computes its color in iteration $t+1$ with the color update function

$$c_\sigma^{t+1} = \text{UPD}_1\left(\sigma, X, c^t\right) = \text{HASH}\left(c_\sigma^t, c_\mathcal{B}^t(\sigma), c_\uparrow^t(\sigma)\right).$$

where $c_\uparrow^t(\sigma)$ performs the non pairwise message passing

$$c_\uparrow^t(\sigma) = \{\!\{\left(c_\mu^t, c_\delta^t\right) \mid \mu \in \mathcal{N}_\uparrow(\sigma) \text{ and } \delta \in \mathcal{C}(\sigma, \mu)\}\!\}.$$

It is straightforward to show that the color update function $\text{UPD}_1$ performs generalized augmented message passing and that CWN can thus be weakly simulated with a simulation factor of 2. However, for this Algorithm 6 would generate more vertices than cells. Instead, we present a hand-crafted graph transformation that requires no additional vertices. We define a color update function that replaces the non pairwise message passing into separate pairwise message aggregations:

$$c_\sigma^{t+1} = \text{UPD}_2\left(\sigma, X, c^t\right) = \text{HASH}\left(\pi_\sigma^t, \pi_\mathcal{B}^t(\sigma), \pi_{\mathcal{N}_\uparrow}^t(\sigma), \pi_{\mathcal{N}_\Uparrow}^t(\sigma)\right),$$

where

$$\mathcal{N}_\Uparrow(\sigma) = \{\delta \mid \mu \in \mathcal{N}_\uparrow(\sigma) \text{ and } \delta \in \mathcal{C}(\sigma, \mu)\}.$$

We require the dimension of cells to be encoded in the initial coloring. Note that $\text{UPD}_2$ performs augmented message passing: $\pi_\sigma^t$ is a color atom (A3) and $\pi_\mathcal{B}^t(\sigma), \pi_{\mathcal{N}_\uparrow}^t(\sigma), \pi_{\mathcal{N}_\Uparrow}^t(\sigma)$ correspond to a AMPs (S2) that aggregates a color over a neighborhood from $\mathcal{B}(\sigma), \mathcal{N}_\uparrow(\sigma), \mathcal{N}_\Uparrow(\sigma)$. Hence, we can *strongly simulate* $\text{UPD}_2$ by Theorem 3.5. Next, we show that for every $t \geq 0$ the coloring produced in iteration $2t$ of $\text{UPD}_2$ refines the coloring produced by $t$ iterations of $\text{UPD}_1$ (CWN). Combining this with the fact that we can strongly simulated $\text{UPD}_2$ means that the coloring produced by $2t$ iterations of strongly simulating $\text{UPD}_2$ refines the coloring from $t$ iterations of CWN. This proves that we can weakly simulated CWN with a simulation factor of 2. We use $\overline{\text{CWN}}$ to denote the transformation $\text{AME}_{\text{UPD}_2}$.

Let $t \geq 0$ be an integer. Let $X, X'$ be two arbitrary regular cell complexes. Let $\sigma, \nu$ be two arbitrary cells from these cell complexes. We show by induction on $t$ that if $\text{UPD}_2$ is unable to distinguish $\sigma$ from $\nu$ in iteration $2t$, then $\text{UPD}_1$ is unable to distinguish cell $\sigma$ from cell $\nu$ in iteration $t$. We use $c_\sigma^t$ to denote the color assigned to cell $\sigma$ by the coloring from iteration $t$ of $\text{UPD}_1$. We use $\pi_\sigma^t$ to denote the color assigned to vertex $\sigma$ in from iteration $t$ of $\text{UPD}_2$. To prove the theorem, we show that $\pi_\sigma^{2t} = \pi_\nu^{2t} \Rightarrow c_\sigma^t = c_\nu^t$. The base case ($t = 0$) holds trivially, as the features of the cells are encoded in the vertex features. Next, we assume the statement holds for $t = T$ and prove that it holds for $t = T + 1$. This means we assume $\pi_\sigma^{2(T+1)} = \pi_\nu^{2(T+1)}$ and have to show:

$$c_\sigma^{T+1} = \left(c_\sigma^T, c_\mathcal{B}^T(\sigma), c_\uparrow^T(\sigma)\right) \stackrel{!}{=} \left(c_\nu^T, c_\mathcal{B}^T(\nu), c_\uparrow^T(\nu)\right) = c_\nu^{T+1}.$$

From $\pi_\sigma^{2(T+1)} = \pi_\nu^{2(T+1)}$ it follows that $\pi_\sigma^{2T+2} = \pi_\nu^{2T+2}$ and $\pi_\sigma^{2T} = \pi_\nu^{2T}$. By the induction hypothesis it then follows that $c_\sigma^T = c_\nu^T$. From $\pi_\sigma^{2T+2} = \pi_\nu^{2T+2}$ it follows that $\pi_\mathcal{B}(\sigma)^{2T+1} = $

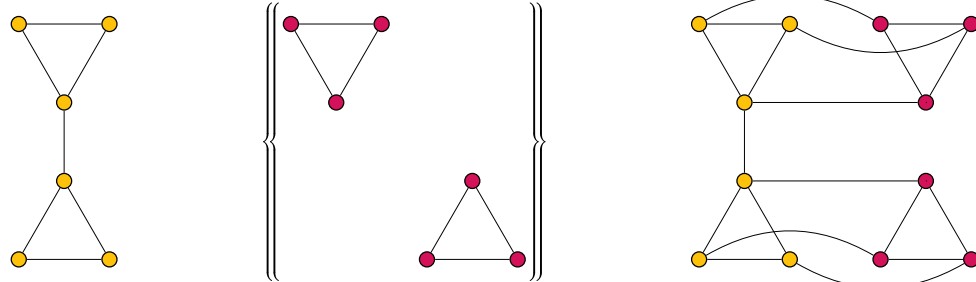

Figure 4: A graph (left) transformed into bags of subgraphs (center) by a policy and the result of applying $\overline{\text{DSS}}$ to the graph and bags of subgraphs (right). Yellow vertices represent the original graph and red vertices the subgraphs.

$\pi_{\mathcal{B}}(\nu)^{2T+1}$. Hence, there exists a bijective function $\alpha : \mathcal{B}(\sigma) \to \mathcal{B}(\nu)$ that maps vertices with the same color assigned by $\pi^{2T+1}$ to another, meaning for all $x \in \mathcal{B}(\sigma)$ it holds that $\pi_x^{2T+1} = \pi_{\alpha(x)}^{2T+1}$ and $\pi_x^{2T} = \pi_{\alpha(x)}^{2T}$. The existence of $\alpha$ and the induction hypothesis imply that $c_{\mathcal{B}}^T(\sigma) = c_{\mathcal{B}}^T(\nu)$. Similarly, there exist a function $\beta : \mathcal{N}_{\Uparrow}(\sigma) \to \mathcal{N}_{\Uparrow}(\nu)$ that bijectively map vertices assigned the same color by $\pi^{2T+1}$ to another. Thus, for every $x \in \mathcal{N}_{\Uparrow}(\sigma)$ it holds that $\pi_x^{2T+1} = \pi_{\beta(x)}^{2T+1}$ and $\pi_x^{2T} = \pi_{\beta(x)}^{2T}$. Furthermore, it holds that for every $x \in \mathcal{N}_{\Uparrow}(\sigma)$ there exists a bijective function $\gamma_x : \mathcal{B}(x) \to \mathcal{B}(\alpha(x))$ that maps cells to another that get assigned the same color by $\pi^{2T}$. Combining this yields

$$\{\!\{ \{\!\{ \pi_x^{2T}, \pi_y^{2T} \}\!\} \mid y \in \mathcal{B}(x), x \in N_{\Uparrow}(\sigma) \}\!\} = \{\!\{ \{\!\{ \pi_x^{2T}, \pi_y^{2T} \}\!\} \mid y \in \mathcal{B}(x), x \in N_{\Uparrow}(\nu) \}\!\} .$$

By construction each $y$ has a higher dimension than $x$. This is encoded in the features, thus it follows that

$$\{\!\{ (\pi_x^{2T}, \pi_y^{2T}) \mid y \in \mathcal{B}(x), x \in \mathcal{B}(\sigma) \}\!\} = \{\!\{ (\pi_x^{2T}, \pi_y^{2T}) \mid y \in \mathcal{B}(x), x \in N_{\Uparrow}(\nu) \}\!\} .$$

From the induction hypothesis it follows that

$$\{\!\{ (c_x^T, c_y^T) \mid y \in \mathcal{B}(x), x \in N_{\Uparrow}(\sigma) \}\!\} = \{\!\{ (c_x^T, c_y^T) \mid y \in \mathcal{B}(x), x \in N_{\Uparrow}(\nu) \}\!\} . \tag{1}$$

We prove that this is equivalent to showing that $c_{\uparrow}^T(\sigma) = c_{\uparrow}^T(\nu)$. For this, we show that for every cell $\rho$ it holds that

$$\{ (\mu, \delta) \mid \mu \in \mathcal{B}(\delta), \delta \in \mathcal{N}_{\Uparrow}(\rho) \} \overset{!}{=} \{ (\mu, \delta) \mid \mu \in \mathcal{N}_{\uparrow}(\rho) \text{ and } \delta \in \mathcal{C}(\rho, \mu) \}.$$

First, we show that the set on the left is a subset of the set on the right. Suppose an arbitrary element $(\mu, \delta)$ with $\mu \in \mathcal{B}(\delta)$ and $\delta \in \mathcal{N}_{\Uparrow}(\rho)$ from the left set. Note that $\mu \in \mathcal{B}(\delta)$ implies $\mu \prec \delta$. Furthermore, $\delta \in \mathcal{N}_{\Uparrow}(\rho)$ means $\rho \prec \delta$. Thus, $\delta \in \mathcal{C}(\rho, \mu)$. Finally, this also implies that $\mu \in \mathcal{N}_{\uparrow}(\rho)$. Hence, $(\mu, \delta)$ is an element of the right set. Next, we show that the set on the right is a subset of the left set. Let $(\mu, \delta)$ with $\mu \in \mathcal{N}_{\uparrow}(\rho)$ and $\delta \in \mathcal{C}(\rho, \mu)$ be an arbitrary element from the set on the right. From $\delta \in \mathcal{C}(\rho, \mu)$ it follows that $\rho \prec \delta$ and $\mu \prec \delta$. By the definition of $\mathcal{B}$, it follows that $\rho \in \mathcal{B}(\delta)$ and $\mu \in \mathcal{B}(\delta)$. It remains to show that $\delta \in \mathcal{N}_{\Uparrow}(\rho)$ meaning we need to show that $\delta \in \{ y \mid x \in \mathcal{N}_{\uparrow}(\delta) \text{ and } y \in \mathcal{C}(\delta, x) \}$. By definition we know that $\mu \in \mathcal{N}_{\uparrow}(\delta)$ and that $\delta \in \mathcal{C}(\delta, \mu)$. Hence, $\delta \in \mathcal{N}_{\Uparrow}(\rho)$. This implies that $(\mu, \delta)$ is an element of the set on the left, meaning that the set on the right is a subset of the set on the left. Thus, the two sets are equal. We use this to rewrite Equation 1 into

$$\{\!\{ (c_x^T, c_y^T) \mid x \in \mathcal{N}_{\uparrow}(\sigma) \text{ and } y \in \mathcal{C}(\sigma, x) \}\!\} = \{\!\{ (c_x^T, c_y^T) \mid x \in \mathcal{N}_{\uparrow}(\nu) \text{ and } y \in \mathcal{C}(\nu, x) \}\!\} . \tag{2}$$

Hence, $c_{\uparrow}^T(\sigma) = c_{\uparrow}^T(\nu)$. This concludes the proof. $\qquad \square$

## F.2 $\overline{\text{DSS}}$: Weakly Simulating DSS Efficiently

Recall that the color update function of DSS WL is defined as

$$c_{v,S}^{t+1} = \text{HASH}\left( c_{v,S}^t, \{\!\{ c_{x,S}^t \mid x \in \mathcal{N}_S(v) \}\!\}, C_v^t, \{\!\{ C_x^t \mid x \in \mathcal{N}_G(v) \}\!\} \right)$$

where $\mathcal{N}_S(v)$ is the set of all neighbors of vertex $v$ in subgraph $S$, and $C_v^t$ contains the colors of $v$ across different subgraphs

$$C_v^t = \{\!\{ c_{v,S'}^t \mid S' \in \pi(G) \text{ and } v \in V(S') \}\!\} \,.$$

Let $v \in V(G)$. According to the graph transformation from Algorithm 6 with simulation factor 2 we would create a separate helper vertex for each vertex from $X = \{x \mid x \in \mathcal{N}_G(v)\}$ where each $x \in X$ is connected to all instances of $x$ across the different subgraphs. A straightforward way to do this, is to add an additional copy of the original graph. A drawback of the graph transformation from Algorithm 6 is that because of $C_v^t = \{\!\{ c_{v,S'}^t \mid S' \in \pi(G) \text{ and } v \in V(S') \}\!\}$ we need to create edges between every copy of $v$ among all subgraphs, that means we create up to $|V| \cdot |\pi(G)|^2$ extra edges. We can remedy this by adding edges between $v$ in the original graph and all instances of $v$ in the other subgraphs. Thus, it suffices to create at most $|E(G)| + |V(G)| \cdot |\pi(G)|$ edges. Similarly, for every $v \in V(G)$ and all subgraphs $S \in \pi(G)$ containing $v$ we need to add edges to the corresponding helper node for all neighbors $\mathcal{N}_G(v)$. Interestingly, it is not necessary to add these edges and instead it is possible to rely on the edges generated by adding an additional copy of the original graph. However, this means that for a vertex $v$ it takes three steps to obtain the information $\{\!\{ C_x^t \mid x \in \mathcal{N}_G(v) \}\!\}$. Thus, this comes at the cost of increasing the simulation factor to 3. We call the resulting graph transformation $\overline{\text{DSS}}$. Figure 4 shows an example of $\overline{\text{DSS}}$.

**Definition F.2 ($\overline{\text{DSS}}$).** Given a graph $G$ and policy $\pi$ we define $G_\pi = (V_\pi, E_\pi, X_\pi, Y_\pi)$ to be the graph obtained by $\overline{\text{DSS}}$. Where

$$V_\pi = \dot{\bigcup_{S \in \pi(G) \cup G}} V(S),$$

$$E_\pi = \bigcup_{S \in \pi(G) \cup G} E(S) \cup \{\{v_G, v_S\} \mid S \in \pi(G) \text{ and } v \in V(S)\}.$$

For unlabeled graphs we introduce a vertex feature that encodes whether it was created from a subgraph $S \in \pi(G)$ or was created from $G$. For labeled graphs we extend the features correspondingly. We add edge features that allow us to distinguish whether an edge was created due from $E(S)$ or from $\{\{v_G, v_S\} \mid S \in \pi(G) \text{ and } v \in V(S)\}$.

We show that for arbitrary policies and graphs WL on the transformed graphs is at least as expressive as DSS-WL on the original graphs.

**Theorem F.3.** $\overline{\text{DSS}}$ *weakly simulates DSS-WL with a simulation factor of 3.*

*Proof.* Let $G$ and $H$ be two graphs and $\pi$ a policy. We use $\tau_{v,S}^t$ to denote the coloring of vertex $v_S$ obtained by iteration $t$ of WL on $G_\pi$ or $H_\pi$, and $\tau_{v,G}^t$ to denote the color in iteration $t$ of WL of vertex $v$ that was created from the original graph $G$ (respectively $\tau_{v,H}^t$). We use $c_{v,S}^t$ to denote the color of vertex $v$ in subgraph $S$ obtained in the $t$-th iteration of DSS-WL. We prove that for every $t \geq 0$ it holds that

$$\forall v, w \in V(G), S, T \in \pi(G) : \tau_{v,S}^{3t} = \tau_{w,T}^{3t} \Rightarrow c_{v,S}^t = c_{w,T}^t$$

by induction on the iteration $t$ of DSS-WL.

**Base case:** All vertices $v_S \in V(G_\pi), w_T \in V(H_\pi)$ are assigned the same initial color (except the indicator that they were created from a subgraph) as their counterpart from DSS-WL. Thus from $\tau_{v,S} = \tau_{w,T}$ it follows that $c_{v,S}^0 = c_{w,T}^0$.

**Induction hypothesis:** We assume the statement holds for all $t < n$ where $n$ is an arbitrary non-negative integer.

**Induction step:** Let $v \in V(G), w \in V(H), S \in \pi(G), T \in \pi(H)$. We assume that $\tau_{v,S}^{3(n+1)} = \tau_{w,T}^{3(n+1)}$ holds and want to show $c_{v,S}^{n+1} = c_{w,T}^{n+1}$. It is equivalent to the assumption that $\tau_{v,S}^{3n+3} = \tau_{w,T}^{3n+3}$ which also implies $\tau_{v,S}^{3n} = \tau_{w,T}^{3n}$. We need to show that

1. $c_{v,S}^n = c_{w,T}^n$,

2. $\{\{c_{x,S}^n \mid x \in \mathcal{N}_S(v)\}\} = \{\{c_{y,T}^n \mid y \in \mathcal{N}_T(w)\}\}$, and

3. $C_v^n = C_w^n$, and $\{\{C_x^n \mid x \in \mathcal{N}_G(v)\}\} = \{\{C_y^n \mid y \in \mathcal{N}_H(w)\}\}$.

From the induction hypothesis and $\tau_{v,S}^{3n} = \tau_{w,T}^{3n}$ it immediately follows that $c_{v,S}^n = c_{w,T}^n$.

We want to show that $\{\{c_{x,S}^n \mid x \in \mathcal{N}_S(v)\}\} = \{\{c_{y,T}^n \mid y \in \mathcal{N}_T(w)\}\}$. From $\tau_{v,S}^{3n+3} = \tau_{w,T}^{3n+3}$ it follows that there exists a bijection $\beta : \mathcal{N}_{G_\pi}(v_S) \to \mathcal{N}_{H_\pi}(w_T)$ where for every $x \in \mathcal{N}_{G_\pi}(v_S)$ it holds that $\tau_x^{3n+2} = \tau_{\beta(x)}^{3n+2}$ which implies $\tau_x^{3n} = \tau_{\beta(x)}^{3n}$. Additionally, all neighbors of $v_S$ are from the subgraph $S$ or the original graph $G$ and the vertex features allow us to distinguish between those two types. Thus, $\beta$ maps vertices from $V(S)$ to vertices from $V(T)$. Note that these are exactly the vertices whose colors are in $\{\{c_{x,S}^n \mid x \in \mathcal{N}_S(v)\}\}$, $\{\{c_{y,T}^n \mid y \in \mathcal{N}_T(w)\}\}$. Thus, by combining this with the induction hypothesis we obtain that $\{\{c_{x,S}^n \mid x \in \mathcal{N}_S(v)\}\} = \{\{c_{y,T}^n \mid y \in \mathcal{N}_T(w)\}\}$.

We want to show that $C_v^n = C_w^n$. We use the function $\beta$ defined in the previous paragraph. Due to the vertex features we know that $\beta$ must map $v_G$ to $w_H$ as there exists only one neighbor with the "created from the original graph" feature among $v_S$' and $w_T$'s neighbors. Hence, $\tau_{v,G}^{3n+2} = \tau_{w,H}^{3n+2}$ which implies $\tau_{v,G}^{3n+1} = \tau_{w,H}^{3n+1}$. This implies that there exists a bijective function $\gamma$ mapping neighbors of $v_G$ to neighbors of $w_G$ such that they are assigned the same color by $\tau^{3n}$. The vertex $v_G$ and $w_G$ in the copy of $H$ have two types of neighbors: neighbors $\mathcal{N}_G(v), \mathcal{N}_H(w)$ from the original graph and copies of $v, w$ over different subgraphs. By the vertex features we can distinguish these two types of neighbors. Thus, $\gamma$ maps copies of $v$ from the different subgraphs to copies of $w$ over different subgraphs. As these vertices correspond to $v, S'$ and $w, T'$ for all $S' \in \pi(G), T' \in \pi(G)$, we can combine this with the induction hypothesis to obtain that $C_v^n = \{\{c_{v,S'}^n \mid S' \in \pi(G) \text{ and } v \in V(S')\}\} = \{\{c_{w,T'}^n \mid T' \in \pi(H) \text{ and } w \in V(T')\}\} = C_w^n$.

Finally, we want to show $\{\{C_x^n \mid \in \mathcal{N}_G(v)\}\} = \{\{C_y^n \mid y \in \mathcal{N}_H(w)\}\}$. Since the copy of $v$ and $w$ created from the original graph can be uniquely identified by its vertex feature from $\mathcal{N}_{G_\pi}(v), \mathcal{N}_{H_\pi}(w)$ we know that $\beta$ maps these two vertices together and thus that $\tau_{v,G}^{3n+2} = \tau_{w,H}^{3n+2}$. Hence, there exists a bijective function $\sigma : \mathcal{N}_G(v) \to \mathcal{N}_H(w)$ such that for every neighbor $x \in \mathcal{N}_G(v)$ it holds that $\tau_{x,G}^{3n+1} = \tau_{\sigma(x),H}^{3n+1}$. We want to show that for every such neighbor $x \in N_G(v)$ it holds that $C_x^n = C_{\sigma(x)}^n$. Let $x$ be one such vertex. As $\tau_{x,G}^{3n+1} = \tau_{\sigma(x),H}^{3n+1}$ we can use the same argument as in the previous paragraph to obtain that $C_x^n = C_{\sigma(x)}^n$. This concludes the proof of the induction hypothesis and shows that the theorem holds. $\qquad \square$

### F.3 Strong Simulation of $K$-hop GNNs and KP-GNNs

We prove the following Theorem:

**Theorem F.4.** *$K$-hop message passing and KP-GNNs [Feng et al., 2022] can be strongly simulated.*

We prove the theorem only for KP-GNNs as $K$-hop message passing is a special case of KP-GNNs. In the proof we define a new message passing algorithm we call *simple KP-GNN* (sKP-GNN). We prove that (1) sKP-GNNs are as expressives as KP-GNNs in every iteration; and (2) that sKP-GNN is strongly simulatable. The theorem then follows from the combination of (1) and (2). We begin with the definition of sKP-GNN.

**Definition F.5.** (sKP-GNN) The color update function of sKP-GNNs is defined as

$$m_v^{l+1,k} = \text{MES}_k^{l,\text{normal}}\left( \{\{(h_u^l, e_{uv}) \mid u \in Q_{v,G}^{k,t}\}\} \right) + f(G_{v,G}^{k,t}),$$

$$f(G_{v,G}^{k,t}) = \text{EMB}((E(Q_{v,G}^{k,t}), C_k^{k'})),$$

$$h_v^{l+1,k} = \text{UPD}_k^l\left( m_v^{l+1,k}, h_v^l \right),$$

$$h_v^{l+1} = \text{COMBINE}^l\left( h_v^{l+1,1}, \ldots, h_v^{l+1,K} \right).$$

Note that sKP-GNN only differs from KP-GNN by the definition of $h_v^{l+1}$:

- sKP-GNN: $h_v^{l+1} = \text{COMBINE}^l\left(h_v^{l+1,1}, \ldots, h_v^{l+1,K}\right)$ and

- KP-GNN: $h_v^{l+1} = \text{COMBINE}^l\left(\{\!\{h_v^{l+1,k} \mid k = 1, 2, \ldots, K\}\!\}\right)$.

It is important to note that this transformation from a nested-aggregation to a tuple is only possible because there is a natural order on the multisets aggregated (the hop size $k$). We prove that sKP-GNN is as expressive as KP-GNN in every iteration.

**Lemma F.6.** *Suppose both sKP-GNN and KP-GNN use an injective COMBINE· function. Then, for every graph $G = (V, E, F)$ and integer $l \geq 0$ it holds that the coloring sKP-GNN$^l(G, F|_V)$ refines the coloring KP-GNN$^l(G, F|_V)$.*

*Proof.* Let $G = (V, E, F)$ and $G' = (V', E', F')$ be two graphs, $v, w \in (V \cup V')$, and $l \geq 0, K \geq 1$ be integers. We furthermore assume that the COMBINE· function is injective. We prove that if

$$\text{COMBINE}^l\left(h_v^{l+1,1}, \ldots, h_v^{l+1,K}\right) = \text{COMBINE}^l\left(h_w^{l+1,1}, \ldots, h_w^{l+1,K}\right) \tag{3}$$

then

$$\text{COMBINE}^l\left(\{\!\{h_v^{l+1,k} \mid k = 1, 2, \ldots, K\}\!\}\right) = \text{COMBINE}^l\left(\{\!\{h_w^{l+1,k} \mid k = 1, 2, \ldots, K\}\!\}\right). \tag{4}$$

From Equation 3 and the assumption that the COMBINE· function is injective it follows that

$$\left(h_v^{l+1,1}, \ldots, h_v^{l+1,K}\right) = \left(h_w^{l+1,1}, \ldots, h_w^{l+1,K}\right).$$

Thus, for every $j \in [\![K]\!]$ it holds that $h_v^{l+1,j} = h_w^{l+1,j}$. This implies Equation 4. $\qquad\square$

**Lemma F.7.** *sKP-GNN can be strongly simulated.*

*Proof.* The proof reuses large parts the proof that KP-GNNs can be weakly simulated (Theorem 5.3). As sKP-GNN only differs from KP-GNN in the definition of $h_v^{t+1}$ it follows from the proof that $h_v^{l+1,k}$ is AMP for every $k \in [\![K]\!]$. Then, $h_v^{t+1}$ is AMP that applies the function COMBINE$^t$ (S4) to a tuple (S3) built from the AMPs $h_v^{l+1,1}, \ldots, h_v^{l+1,K}$. From Theorem 3.5 it follows that sKP-GNN can be strongly simulated. $\qquad\square$

*Proof.* (Theorem F.4) The theorem follows from Lemma F.6 and Lemma F.7. $\qquad\square$

# G  Intuition for CWN and $\overline{\text{CWN}}$

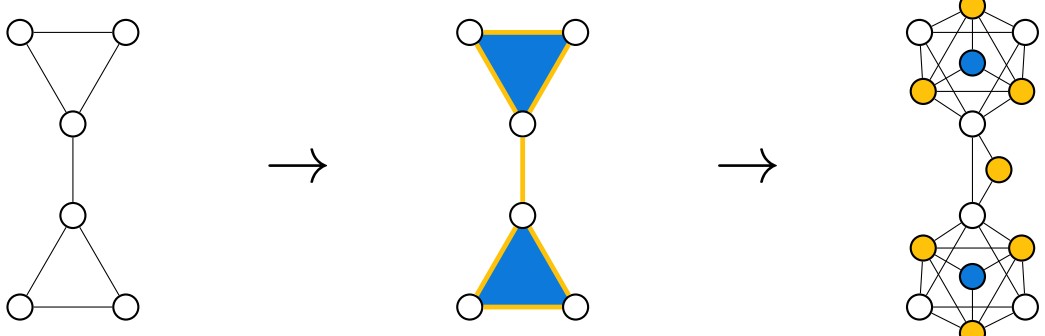

Figure 5: Left: graph. Center: regular cell complex built from the graph through a graph-to-structure encoding [Bodnar et al., 2021a]. Vertices correspond to 0-dimensional cells, edges to 1-dimensional cells ( yellow ) and induced cycles to 2-dimensional cells ( blue ). Right: a graph created by structure-to-graph encoding the regular cell complex to a graph. Vertices corresponds to cells as indicated by color.

CW Networks [Bodnar et al., 2021a, CWN] perform message passing on topological structures called regular cell complexes. A regular cell complex is built out of individual cells of some dimension.

Similar to message passing on graphs, CWN computes a representation (or color) for each cell by aggregating from the neighboring cells. To apply CWNs to graphs, a regular cell complex has to be constructed from a graph (see Figure 5), we call this process graph-to-structure encoding. For this vertices are transformed into 0-dimensional cells and edges into 1-dimensional cells. In this process additional structures such as (induced) cycles or cliques can also be lifted to cells to improve the expressiveness and connectivity in the structure. In the experiments we use a lifting map that lifts induced cycles of length at most 6 to cells. Figure 5 shows how a regular cell complex is transformed to a graph and shows how our structure-to-graph encoding $\overline{\text{CWN}}$ transforms the complex back to a graph. Note that in our implementation of $\overline{\text{CWN}}$ we use undirected instead of directed for the sake of simplicity. When transforming a cell to a vertex, we set the vertex features to the most common feature of the vertices making up the cell for each feature. For example, a cell that corresponds to a cycle $v, w, p$ is transformed to a vertex whose features are the most common of the features of $v, w, p$. Similarly, the features of edges connecting cells are the most common of the edge features between the vertices making up the cells.

## H More Details about the Experiments

We provide further details about our experiments. In Section H.1 we elaborate on our setup and in Section H.2 we present more details on the results from the main paper. In Section H.3 we present an additional evaluation of transformation and training speed. The code for our experiments can be found at https://github.com/ocatias/GNN-Simulation.

### H.1 Setup

All models are implemented in Pytorch Geometric [Fey and Lenssen, 2019]. We use WandB [Biewald, 2020] to track our experiments. Unfortunately, CWN requires an old version of PyTorch [Paszke et al., 2019] meaning that we have to train it on older GPUs. CWN training and all speed evaluations are performed on an an NVIDIA GeForce GTX 1080 GPU. All other experiments are performed on NVIDIA GeForce RTX 3080 GPUs.

**Type Pooling.**   In our transformations $\overline{\text{DSS}}$ and $\overline{\text{CWN}}$ vertices get assigned additional features that encode the type of a vertex. In $\overline{\text{CWN}}$, these features encode whether the cell was built from a vertex, edge or cycle. In $\overline{\text{DSS}}$, these features encode whether a vertex is part of the original graph or in a subgraph. Inspired by a pooling operation by Bodnar et al. [2021a] we propose *type pooling*. The idea is that vertices with different types contain different information and have varying importance. Thus, instead of pooling all vertices in a graph, we instead pool all vertices of a type. To get a single output for the entire graph, we then apply a multilayer perceptron with a ReLU activation function to each pooled vector and then sum the results. Let POOL be a pooling operation such as $\sum$. Let $T$ be the set of types and let $H_t$ with $t \in T$ be the multiset of all representations of vertices of type $t$, and $\text{MLP}_t$ be a multilayer perceptron with activation function. Then, type pooling computes

$$\text{Type-Pool} = \sum_{t \in T} \text{MLP}_t \left( \text{POOL}(H_t) \right).$$

We use type pooling for all combinations of an MPNN plus $\overline{\text{DS}}$, $\overline{\text{DSS}}$ and $\overline{\text{CWN}}$. Note that for $\overline{\text{DS}}$ all nodes have just one type.

**Hyperparameter Tuning.**   Table 3 contains all hyperparameter grids for the real life tasks. During preliminary experiments we found that it is crucial to tune the pooling operation and number of layers to achieve strong results. For DS(S) we do not tune the pooling function as the code provided by Bevilacqua et al. [2021] does not directly support this. Since CWN is very slow, we use a smaller hyperparameter grid, containing a superset of the hyperparameters used by Bodnar et al. [2021a].

**Setup Real-Life Datasets.**   For all real-life datasets we tune the hyperparameters on the validation set and evaluate the best hyperparameters on the test set 10 times. We report the average and standard deviation of the metric that is typically used on the given dataset. For all OGB datasets, we train with a batch size of 32 for 100 epochs with a fixed learning rate to allow for fast training of many models. For ZINC we train as described by Bevilacqua et al. [2021], Bodnar et al. [2021a] and Dwivedi et al.

Table 3: Hyperparameter grid used for the real life datasets. The learning rate is only tuned on `OGB` tasks. On `ZINC` all models used the same initial learning rate that decays over time.

| MODEL → 
 ↓ PARAMETERS | GIN, GCN | DSS, DS | CIN ON OGB | CIN ON ZINC |
|---|---|---|---|---|
| EMBEDDING DIMENSION | 64, 256 | 64, 256 | 64, 256 | 64, 256 |
| NUMBER OF LAYERS | $1, \dots, 5$ | $1, \dots, 5$ | 1,2,3 | 3,4,5 |
| LEARNING RATE | $10^{-3}, 10^{-4}$ | $10^{-3}, 10^{-4}$ | $10^{-3}, 10^{-4}$ | $10^{-3}$ |
| POOLING OPERATION | SUM, MEAN | MEAN | SUM, MEAN | SUM, MEAN |
| DROPOUT RATE | 0.5 | 0.5 | 0.5 | 0.5 |

[2023] i.e., for up to 500 epochs with a batch size of 128 with an initial learning rate of $10^{-3}$ that reduces by a factor of $0.5$ every time the validation result has not improved for 20 epochs. If the learning rate dips below $10^{-5}$ the training stops.

**Setup CSL.** For the synthetic CSL dataset, we only evaluate GIN plus the graph transformations and compare to the results reported by Bevilacqua et al. [2021] and Bodnar et al. [2021a]. We perform 10-fold cross validation on this dataset. For all models with graph transformations we replicate the hyperparameters used by the corresponding non-standard message passing algorithms described in the original papers. For GIN, we tune hyperparameters as described for real-life datasets.

## H.2 Results on Real-Life Datasets

Tables 4 and 5 show the prediction quality of our models on all real-life datasets.

Table 4: **Bold** results are better than the corresponding message passing baseline, red results are graph transformation based methods that are better than the corresponding non-standard message passing variation. ↓ means lower is better. ↑ means higher is better.

| DATA SET → 
 ↓ MODEL | MOLHIV 
 ROC-AUC ↑ | MOLTOX21 
 ROC-AUC ↑ | MOLESOL 
 RSMSE ↓ | ZINC 
 MAE ↓ |
|---|---|---|---|---|
| GIN | $77.7 \pm 1.0$ | $76.1 \pm 0.5$ | $0.946 \pm 0.086$ | $0.306 \pm 0.035$ |
| GCN | $76.6 \pm 1.0$ | $74.9 \pm 0.6$ | $0.928 \pm 0.049$ | $0.456 \pm 0.021$ |
| MLP | $72.2 \pm 0.9$ | $71.2 \pm 0.7$ | $1.128 \pm 0.021$ | $1.44 \pm 0.001$ |
| DS | $75.4 \pm 1.4$ | $75.4 \pm 0.6$ | $0.986 \pm 0.051$ | $\mathbf{0.161 \pm 0.008}$ |
| $\overline{\text{DS}}$ + GIN | $77.2 \pm 1.2$ | $\mathbf{77.0 \pm 0.6}$ | $\mathbf{0.924 \pm 0.03}$ | $\mathbf{0.143 \pm 0.008}$ |
| $\overline{\text{DS}}$ + GCN | $75.4 \pm 0.8$ | $\mathbf{76.6 \pm 0.6}$ | $0.872 \pm 0.037$ | $0.285 \pm 0.044$ |
| $\overline{\text{DS}}$ + MLP | $69.8 \pm 6.1$ | $70.8 \pm 0.8$ | $1.292 \pm 0.115$ | $\mathbf{1.281 \pm 0.002}$ |
| DSS | $76.8 \pm 1.3$ | $\mathbf{77.0 \pm 1.0}$ | $0.864 \pm 0.024$ | $0.105 \pm 0.005$ |
| $\overline{\text{DSS}}$ + GIN | $74.4 \pm 1.3$ | $\mathbf{78.1 \pm 0.7}$ | $\mathbf{0.853 \pm 0.015}$ | $0.149 \pm 0.008$ |
| $\overline{\text{DSS}}$ + GCN | $74.8 \pm 2.1$ | $\mathbf{76.3 \pm 0.7}$ | $0.868 \pm 0.022$ | $0.221 \pm 0.028$ |
| $\overline{\text{DSS}}$ + MLP | $71.9 \pm 0.8$ | $69.7 \pm 0.5$ | $1.272 \pm 0.025$ | $\mathbf{1.283 \pm 0.006}$ |
| CWN | $\mathbf{78.6 \pm 1.5}$ | $75.2 \pm 0.9$ | $1.117 \pm 0.051$ | $\mathbf{0.13 \pm 0.003}$ |
| $\overline{\text{CWN}}$ + GIN | $\mathbf{78.4 \pm 1.5}$ | $74.4 \pm 0.7$ | $0.958 \pm 0.021$ | $\mathbf{0.137 \pm 0.007}$ |
| $\overline{\text{CWN}}$ + GCN | $\mathbf{78.1 \pm 1.0}$ | $74.2 \pm 0.9$ | $0.933 \pm 0.036$ | $\mathbf{0.155 \pm 0.013}$ |
| $\overline{\text{CWN}}$ + MLP | $71.8 \pm 0.8$ | $71.0 \pm 0.5$ | $1.217 \pm 0.032$ | $\mathbf{1.045 \pm 0.006}$ |

## H.3 Speed Evaluation

**Setup.** We evaluate the preprocessing and training speed on `ZINC` for GIN and our graph transformations against DS(S) and CWN. We select the hyperparameters for all GNNs such that they all have roughly $8 \cdot 10^5$ trainable parameters. To measure the transformation speed, we begin the time measurement after the graphs have been loaded into memory and stop before we store the final objects. For our graph transformations, this measures how long it takes to run the graph transformations. For CWN, this measures how long it takes to create the cell complexes from the graphs. Additionally, we measure how long it takes to train the model for 100 epochs on `ZINC` evaluating it after every epoch. We average all time measurements over 10 trials.

Table 5: **Bold** results are better than the corresponding message passing baseline, red results are graph transformation based methods that are better than the corresponding non-standard message passing variation. ↓ means lower is better. ↑ means higher is better.

| DATA SET →  ↓ MODEL | MOL-BACE  ROC-AUC ↑ | MOL-CLINTOX  ROC-AUC ↑ | MOL-BBBP  ROC-AUC↑ | MOL-SIDER  ROC-AUC ↑ | MOL-TOXCAST  ROC-AUC ↑ | MOL-LIPO  RMSE ↓ |
|---|---|---|---|---|---|---|
| GIN | $75.3 \pm 4.9$ | $87.1 \pm 1.5$ | $66.8 \pm 2.3$ | $56.8 \pm 1.0$ | $64.2 \pm 0.7$ | $0.787 \pm 0.026$ |
| GCN | $76.9 \pm 3.8$ | $88.2 \pm 1.2$ | $66.6 \pm 2.1$ | $57.5 \pm 0.9$ | $63.2 \pm 0.5$ | $0.793 \pm 0.028$ |
| MLP | $41.6 \pm 11.2$ | $57.1 \pm 3.9$ | $63.7 \pm 1.8$ | $58.9 \pm 1.4$ | $62.8 \pm 0.3$ | $1.048 \pm 0.009$ |
| DS | $\mathbf{79.2 \pm 1.9}$ | $85.7 \pm 2.9$ | $\mathbf{67.4 \pm 1.3}$ | $\mathbf{60.6 \pm 1.1}$ | $\mathbf{66.3 \pm 0.5}$ | $\mathbf{0.736 \pm 0.022}$ |
| $\overline{\text{DS}}$ + GIN | $75.3 \pm 3.5$ | $81.4 \pm 4.2$ | $66.2 \pm 1.8$ | $\mathbf{\color{red}59.3 \pm 0.9}$ | $\mathbf{\color{red}66.7 \pm 0.5}$ | $\mathbf{\color{red}0.716 \pm 0.014}$ |
| $\overline{\text{DS}}$ + GCN | $\mathbf{77.7 \pm 2.8}$ | $83.8 \pm 2.4$ | $66.0 \pm 1.1$ | $\mathbf{\color{red}61.0 \pm 0.9}$ | $66.1 \pm 0.4$ | $0.754 \pm 0.018$ |
| $\overline{\text{DS}}$ + MLP | $\mathbf{71.8 \pm 5.6}$ | $55.4 \pm 5.0$ | $63.6 \pm 2.1$ | $52.9 \pm 1.6$ | $58.5 \pm 5.5$ | $1.048 \pm 0.011$ |
| DSS | $74.5 \pm 7.4$ | $84.6 \pm 2.6$ | $\mathbf{67.5 \pm 1.5}$ | $57.5 \pm 1.5$ | $66.4 \pm 0.8$ | $0.652 \pm 0.009$ |
| $\overline{\text{DSS}}$ + GIN | $\mathbf{\color{red}77.2 \pm 3.0}$ | $86.4 \pm 1.2$ | $64.3 \pm 2.5$ | $\mathbf{\color{red}59.0 \pm 1.1}$ | $\mathbf{66.8 \pm 0.4}$ | $0.76 \pm 0.018$ |
| $\overline{\text{DSS}}$ + GCN | $\color{red}74.8 \pm 3.9$ | $\color{red}87.3 \pm 1.6$ | $\mathbf{67.3 \pm 2.0}$ | $\mathbf{\color{red}62.0 \pm 1.0}$ | $66.1 \pm 0.6$ | $0.767 \pm 0.027$ |
| $\overline{\text{DSS}}$ + MLP | $\mathbf{72.8 \pm 4.3}$ | $51.9 \pm 7.7$ | $63.5 \pm 2.0$ | $51.6 \pm 2.0$ | $62.0 \pm 0.8$ | $1.054 \pm 0.01$ |
| CWN | $75.0 \pm 2.7$ | $84.7 \pm 2.4$ | $\mathbf{69.5 \pm 1.6}$ | $57.6 \pm 1.6$ | $64.7 \pm 0.9$ | $0.755 \pm 0.018$ |
| $\overline{\text{CWN}}$ + GIN | $\color{red}75.2 \pm 3.3$ | $\color{red}85.4 \pm 2.4$ | $68.1 \pm 2.4$ | $\mathbf{\color{red}59.1 \pm 1.0}$ | $64.5 \pm 1.1$ | $\mathbf{\color{red}0.749 \pm 0.013}$ |
| $\overline{\text{CWN}}$ + GCN | $72.2 \pm 3.8$ | $\color{red}88.0 \pm 1.6$ | $61.5 \pm 3.0$ | $\mathbf{\color{red}62.4 \pm 1.1}$ | $64.4 \pm 0.6$ | $\mathbf{0.787 \pm 0.016}$ |
| $\overline{\text{CWN}}$ + MLP | $38.1 \pm 0.6$ | $55.7 \pm 4.8$ | $62.7 \pm 1.6$ | $\mathbf{\color{red}59.9 \pm 1.1}$ | $\mathbf{63.1 \pm 0.5}$ | $1.055 \pm 0.01$ |

Table 6: Speed comparison of different GNNs. TRANSFORM is the time to transform the graphs to the required structure to perform message passing over. TRAIN is the time to train a model to train for 100 epochs. COMBINED is the combined time of training and transformation. **Bold** indicates the best result for a given non-standard message passing algorithm and corresponding graph transformation.

| MODEL | TRANSFORM  SEC. ↓ | TRAIN.  SEC. ↓ | COMBINED  SEC. ↓ |
|---|---|---|---|
| GIN | NA | $170 \pm 1$ | $170 \pm 1$ |
| CWN | $\mathbf{37 \pm 1}$ | $3259 \pm 16$ | $3296 \pm 17$ |
| $\overline{\text{CWN}}$ + GIN | $67 \pm 1$ | $\mathbf{286 \pm 6}$ | $\mathbf{353 \pm 7}$ |
| DSS | $\mathbf{95 \pm 4}$ | $\mathbf{955 \pm 5}$ | $\mathbf{1050 \pm 9}$ |
| $\overline{\text{DSS}}$ + GIN | $183 \pm 8$ | $1466 \pm 7$ | $1649 \pm 15$ |
| DS | $\mathbf{95 \pm 4}$ | $1235 \pm 5$ | $1330 \pm 9$ |
| $\overline{\text{DS}}$ + GIN | $171 \pm 8$ | $\mathbf{1060 \pm 5}$ | $\mathbf{1231 \pm 13}$ |

**Results.** Table 6 shows the results of the speed evaluation. We can see that the graph transformation are roughly twice as slow as the alternative transformation for the non-standard message passing. However, this runtime is still significantly smaller than the time it takes to train the model a single time. Thus, the runtime of graph transformations is insignificant as the transformation only needs to be performed a single time whereas the model might need to be trained dozens of times to find good hyperparameters. With respect to the training time, $\overline{\text{DSS}}$+GIN is roughly 54% slower than DSS and $\overline{\text{DS}}$+GIN is roughly 17% faster than DS. Interestingly, $\overline{\text{CWN}}$+GIN is 11 times faster than CWN. While it might seem surprising that our graph transformation based variant of CWN is 11x faster than the original implementation, the reasons for that might be: (1) a bug, for example in how the original code of CWN interacts with our environment; (2) CWN requires a large amount of code built on top of PyTorch Geometric, whereas $\overline{\text{CWN}}$+GIN just performs standard message passing; (3) the implementation of CWN by Bodnar et al. [2021a] is not compatible with newer variants of PyTorch (Geometric) meaning that $\overline{\text{CWN}}$+GIN can make use of speed increases due to improvements in PyToch (Geometric). Reasons (2) and (3) highlight the strength of graph transformations in that they are mostly platform and implementation independent as the transformed graphs can be stored in file formats that are independent from the used deep learning framework.

