# OpenReview forum: "Expressivity-Preserving GNN Simulation"
_NeurIPS.cc/2023/Conference — NeurIPS 2023 poster_

### Official Review · Reviewer_sDHK · 2023-06-30

**Soundness:** 3 good
**Presentation:** 3 good
**Contribution:** 2 fair
**Rating:** 4
**Confidence:** 4

**Summary:**

The paper deals with supervised machine learning with graphs, specifically with expressive GNNs, and how to implement them efficiently. Specifically, it investigates the expressive power of graph transformations to transform an input graph such that an ordinary 1-WL-equivalent message-passing GNN can simulate, e.g., k-WL-equivalent GNNs, in a layer-wise fashion (strong simulation). To that, the authors formally define this transformation and show that an MPNN can simulate many popular, more expressive GNNs, see Corollary 3.6.

The authors acknowledge that such a transformation is folklore knowledge.

Moreover, they investigate the regime (weak simulation) where the MPNN on the transformed graph needs more iterations to distinguish the same pairs of non-isomorphic graphs. Restricted to their particular definition of transformation, they show that some GNN layers cannot be strongly simulated but weakly simulated.

Empirically, they show that their transformations often beat standard message-passing GNNs regarding predictive performance on standard benchmarks. Somewhat surprisingly, they show that they sometimes even beat the architectures they are aiming to simulate.




**Strengths:**

The paper is easy to read, and everything is formalized appropriately. The proofs seem to be correct, and the theoretical results are plausible.

To the best of my knowledge, Theorem 5.2 and Corollary 5.3 are new and somewhat interesting.

The experimental study is well conducted, and the presented results are somewhat interesting.

**Weaknesses:**

Section 3 just contains more or less obvious results known within the graph learning community. (This is also clearly acknowledged by the authors in the introduction and the appendix.). However, with this in mind, it is unclear why Section 3 occupies 1/3 of the main paper.

It seems that the proof of Theorem 4.1 strongly exploits specific details of "Structure-to-graph encoding" (Def. 3.1), which makes the results somewhat narrow. The authors should be more clear about this.

The reasoning in lines 191 -- 195 is highly handwavey. You seem to implicitly assume that the function is continuous on a compact domain.




**Questions:**

Q1: Do you, to some extent, understand why the simulated architectures often lead to better predictive performance over their simulated counterpart?

Q2:  Do you, to some extent, know if the simulation factor influences predictive performance? For example, does a high simulation factor lead to better or worse predictive performance?

**Limitations:**

It seems that the proof of Theorem 4.1 strongly exploits specific details of "Structure-to-graph encoding" (Def. 3.1), which makes the results somewhat narrow. This should be clarified.

---

> ### Author Rebuttal · Authors · 2023-08-09
>
> We thank the reviewer for the feedback.
>
> ### Concerning the Weaknesses
> >Section 3 just contains more or less obvious results known within the graph learning community. (This is also clearly acknowledged by the authors in the introduction and the appendix.). However, with this in mind, it is unclear why Section 3 occupies 1/3 of the main paper.
>
> We agree, parts of the graph learning community know that some specific GNN architectures can be simulated (some of these GNNs are even implemented through an MPNN applied on a transformed graph, as discussed in our appendix).
> However, formalizing and proving such informal knowledge is an added value for the GNN community. Additionally, we prove that is it possible to simulate further GNNs for which it was previously not know. In particular, the key results of Section 3 are:
>
> (1) a formal definition of strong simulation (i.e., preserving expressivity in each iteration / layer),
>
> (2) a general algorithm that yields the required graph transformations to strongly simulate GNNs that follow our definition of augmented message passing, and
>
> (3) simulation results for 9 GNNs, 5 of which were not known to be simulatable before (and were originally implemented though non-standard message passing).
>
> As far as we know, the formal definition of strong simulation is novel. Through this definition we generalize individual results into a general theoretical paradigm. Furthermore, while such results might be obvious to the original authors of a GNN, coming up with a corresponding graph transformations is non-trivial and previously has required proving their correctness for every single architecture. With (2) we streamline this approach: our algorithm yields a graph transformation (with guaranteed strong simulation) for any GNN that follows our definition of augmented message passing.
>
> Remark on (3): skimming recent LoG and ICML paper, we found two more strongly simulatable GNNs that have not been known to be simulatable before and were implemented through non-standard message passing (see global response for more details). A thorough literature review would most likely yield many more strongly simulatable GNNs.
>
> >It seems that the proof of Theorem 4.1 strongly exploits specific details of "Structure-to-graph encoding" (Def. 3.1), which makes the results somewhat narrow. The authors should be more clear about this.
>
> To obtain a useful definition of simulation it is **necessary** to put restrictions on the transformations (eg a computationally expensive way to simulate any GNN is to encode the isomorphism class in any graph label).
> Our definition of structure-to-graph encoding details these necessary restrictions.
>
> We enforce two natural restrictions on structure-to-graph encoding and simulation. The first restriction limits the asymptotic runtime of the encoding to the runtime of a single iteration of the simulated algorithm. This avoids the case of precomputing isomorphism classes and running other expensive algorithms. However, this still allows to use the simulated algorithm for a large (but constant) number of iterations to produce features in the structure-to-graph encoding. The second restriction decouples the generation of the graph structure from the features of the original structure which avoids this issue. These two restrictions allow the simulation of many common GNN architectures and simultaneously forbid the previously discussed trivial cases.
>
> We explain this partly in the discussion section of our paper but we will make it clearer in the camera ready version.
>
> >The reasoning in lines 191 -- 195 is highly handwavy. You seem to implicitly assume that the function is continuous on a compact domain.
>
> Thanks for spotting. We will fix this and add the required assumptions to the camera-ready version.
>
>
> ### Concerning the Questions
> > Q1: Do you, to some extent, understand why the simulated architectures often lead to better predictive performance over their simulated counterpart?
>
> As indicated in Table 2, GIN + a graph transformation beats the original GNN in 50-60% of experiments. This is exactly the expected behavior: our simulation performs on par with the original GNN; it's basically a fair coin flip, which of the two is better. We refer the remaining performance differences to randomness / noise in training and hyperparameter search.
>
> > Q2: Do you, to some extent, know if the simulation factor influences predictive performance? For example, does a high simulation factor lead to better or worse predictive performance?
>
> This is an interesting question for follow-up work as we have two opposing ideas about this. A higher simulation factor means that when an MPNN simulates a higher-order GNN it will need more layers to achieve the same expressivity. We hypothesize that this could lead to worse predictive performance as deeper GNNs are harder to train due to issues such as oversmoothing. However, there has been recent work [Bause 2022] [Azabou 2023] that argued that slowing down message passing (hence using more layers to perform one standard $1$-WL aggregation) can lead to better predictive results. As a simulation factor greater than 1 slows down message passing compared to the simulated architecture, this could also improve the predictive performance. Overall, more research into this direction is needed.
>
> #
>
> [Azabou 2023]: Azabou et al.; Half-Hop: A graph upsampling approach for slowing down message passing; ICML, 2023
>
> [Bause 2022]: Bause and Kriege; Gradual Weisfeiler-Leman: Slow and Steady Wins the Race; Learning on Graphs Conference, 2022

---

> > ### Comment · Reviewer_sDHK · 2023-08-11
> > **Answers**
> >
> > Thank you for your answers. If the authors address the above-mentioned shortcomings, I am not against accepting the paper, especially since the other reviewers seem to like the paper. However, I still believe that the results are mostly implicitly known to the community. Hence, I will keep my current score.

---

### Official Review · Reviewer_sQdR · 2023-07-05

**Soundness:** 4 excellent
**Presentation:** 3 good
**Contribution:** 4 excellent
**Rating:** 8
**Confidence:** 3

**Summary:**

The paper investigates the idea of simulating (replacing) non-standard message passing networks (MPNs) using simple standard MPNs by first applying a graph transformation (new nodes and edges). The paper provides a formal construction of graphs and their generalizations in the form of relational structures that allows for describing e.g. non-pair-wise interactions and cellular complexes. The step through relation structures is important to make arrive to the main theorem of the paper, which is that many non-standard MPNs (based on higher-order relational structures) can be simulated with standard MPNs if the relational structure is converted to a graph in an appropriate way. The theoretical results are underpinned with experiments, showing that indeed simple MPNs on transformed graphs perform on-par (or even outperform) advanced non-standard MPNs.

**Strengths:**

* The paper is precise and clear and systematically builds up to the theorems. The various formal definitions provided along the way are helpful in getting a high level picture of the graph NN field. At the same time, I must admit, that the formal approach does require quite some attention from the reader as the paper is dense with information and intuition is sometimes hard to acquire.

* The experiments are effectively presented. Table 1 and 2 are interpretable and convincing.

* The paper provides a novel analysis

* The paper gives useful insights into the expressivity of various forms of graph NNs.

* The paper seems reproducible with a link to clean (anonymous) code in the supplementary materials, and the supplementary materials are otherwise very thorough as well.


**Weaknesses:**

The main paper has few details for practitioners. E.g., how to convert a relational structure to a graph is not explained. Details are however provided in the supplementary materials. The paper could benefit from a running example, perhaps the CWN case (fig 1) in which the struct2graph conversion is intuitively explained, as well as why one can similar CWN on this new graph. An example is given at line 248, but feels a bit too abstract for me.

**Questions:**

Although figure 1 was very helpful in getting the idea across. Figure 2 was less clear, mainly due to the notion of “nested aggregation” not being well explained. E.g. line 273 shows a color update function that depends on a nesting of sets, but how do I relate this to Figure 2? Also I read this as nested sets, but it is written “nested aggregation functions”, is $C_x$ an aggregation function then? I’m confused here as to what aggregation refers to, could you explain this?

Line 94 the notation $2^{(U^l)}$ was unknown to me so I had to google it. It means the space of all mappings/functions $a: U^l\rightarrow\{0,1\}$, right? Perhaps some detail/definition can be added.

Are there any general recommendations for how to derive the graph transform to simulate non-standard MPNNs, or is this a case by case analysis? E.g., it is still unclear to me how one should derive/define R.



**Limitations:**

Limitations are appropriately discussed.

---

> ### Author Rebuttal · Authors · 2023-08-09
>
> We thank the reviewer for their positive feedback and appreciate that the reviewer acknowledges that our paper is precise, clear, has convincing experiments, and provides a novel analysis of graph neural networks.
>
> > At the same time, I must admit, that the formal approach does require quite some attention from the reader as the paper is dense with information and intuition is sometimes hard to acquire.
>
> Thank you for pointing this out, we will add more examples to aid with intuition.
>
> ### Concerning the Weakness
> > The main paper has few details for practitioners. E.g., how to convert a relational structure to a graph is not explained. Details are however provided in the supplementary materials. The paper could benefit from a running example, perhaps the CWN case (fig 1) in which the struct2graph conversion is intuitively explained, as well as why one can similar CWN on this new graph. An example is given at line 248, but feels a bit too abstract for me.
>
> We provide some tips on how to transform structures to graphs in the proof sketch of Theorem 3.5. However, we agree that the main paper does not give enough explanations. We will add CWN as a running example.
>
>
> ### Concerning the Questions
> > Although figure 1 was very helpful in getting the idea across. Figure 2 was less clear, mainly due to the notion of “nested aggregation” not being well explained. E.g. line 273 shows a color update function that depends on a nesting of sets, but how do I relate this to Figure 2? Also I read this as nested sets, but it is written “nested aggregation functions”, is $C_x$ an aggregation function then? I’m confused here as to what aggregation refers to, could you explain this?
>
> In the paper we write _nested aggregation function_ but we think that the word function might be confusing and will thus change it to just _nested aggregation_. Furthermore, we will explain nested aggregations better in the final version of the paper. In the definition $C_x$ is meant to be an aggregation $C_x = \\{\\!\\{ c_y \\mid y \\in \\mathcal{N}(x) \\}\\!\\}$ (see line 273).  A nested aggregation is an aggregation inside another aggregation. For example, $\\{\\!\\{ c_y^t \\mid y \\in \\mathcal{N}_2(x) \\}\\!\\}$  is a single aggregation and $\\{\\!\\{ \\{\\!\\{ c_y^t \\mid y \\in \\mathcal{N}_2(x) \\}\\!\\} \\mid x \\in \\mathcal{N}_1 (v) \\}\\!\\}$ is a  a nested-aggregation. Consider the vertices in Figure 2 left, we refer to the top vertices as $a, b$ and the bottom vertices as $x, y$. Then Figure 2 is meant to be read as the blue vertex receives the colors $\\{\\!\\{ \\{\\!\\{ c_a^t, c_b^t \\}\\!\\}, \\{\\!\\{ c_x^t, c_y^t \\}\\!\\} \\}\\!\\}$.
>
> > Line 94 the notation $2^{(U^l)}$ was unknown to me so I had to google it. It means the space of all mappings/functions $a: U^l \\to 0,1$, right? Perhaps some detail/definition can be added.
>
> The _power set_ of a set $X$ contains all subsets $2^X=\\{Y \\mid Y \\subseteq X\\}$ of that set $X$. For a set $U$ and an integer $\\ell > 1$, the set $U^\\ell=\\{ (u_1, \\ldots, u_\\ell) \\mid u_1 \\in U, \\ldots, u_\\ell \\in U \\}$ is the set containing all tuples of length $\\ell$ built from elements of $U$. Thus, for a set of objects $U$, the set $2^{(U^\\ell)}$ consists of all possible sets built from $\\ell$-tuples of $U$. For example, $U = \\{x, y \\}$ and $\\ell = 2$. Then, $U^\\ell = \\{ (x,x), (x,y), (y,x), (y,y) \\}$ and $2^{(U^l)} = \\{\\emptyset, \\{  (x,x) \\}, \\{  (x,y) \\}, \\{  (y,x) \\}, \\{  (y,y) \\}, \\{  (x,x), (x,y) \\}, \\ldots, \\{(x,x), (x,y), (y,x), (y,y) \\} \\}$. Finally, the neighborhood function $\\mathcal{N}$ assigns one set in the domain $2^{(U^l)}$ to each element $u\\in U$. We will add some details about this to the paper.
>
>
> > Are there any general recommendations for how to derive the graph transform to simulate non-standard MPNNs, or is this a case by case analysis? E.g., it is still unclear to me how one should derive/define R.
>
> We provide an automated way of deriving the graph transformations with Algorithm 4 (strong simulation) and Algorithm 6 (weak simulation). We see this as one of the key strengths of our paper: it is no longer necessary to design the graph transformations by hand for every architecture and to prove their correctness. Instead, our algorithms automatically yield the required graph transformations to simulate the given non-standard MPNN. For strong simulation, the algorithm first creates a vertex for every object and then adds edges between two vertices if the corresponding two objects exchange messages in the simulated algorithm. The main difference for weak simulation are the cases with non-pairwise message passing and nested aggregations where we need to add dummy vertices (see proof sketch of Theorem 5.2).

---

> > ### Comment · Reviewer_sQdR · 2023-08-21
> > **Still in support of acceptance**
> >
> > Thank you for your reply, and sorry for the late reply (holidays...). I appreciate the clarifications and have confidence the (otherwise already nice) paper will be further improved with these minor adjustments.
> >
> > I keep my score at 8 (strong accept)

---

### Official Review · Reviewer_r6Zx · 2023-07-10

**Soundness:** 4 excellent
**Presentation:** 4 excellent
**Contribution:** 3 good
**Rating:** 7
**Confidence:** 3

**Summary:**

The authors introduce methods for simulating certain graph neural networks (GNNs) using standard message-passing algorithms composed with graph transformations. To do so, the authors introduce a class of nonstandard message-passing algorithms they call "augmented message passing" (AMP) algorithms, demonstrate that many standard GNNs are composed of AMP layers, and explicitly construct graph transformations such that standard message passing on the transformed graph input is at least as expressive as a given AMP algorithm. The authors also discuss "weak simulation," and consider more specifically existing techniques that can be weakly simulated via the methods introduced by the authors. Finally, the authors numerically evaluate the performance of their introduced methods.

**Strengths:**

The authors contribute a novel method for constructing message-passing neural networks (MPNNs) on transformed graph inputs that are at least as expressive as a wide class of GNNs with nonstandard message passing. This gives a standard way of viewing GNNs composed of AMP layers, which encompasses many state-of-the-art GNNs. This also formalizes previous work that considered graph transformations composed with MPNNs as a way to make MPNNs more expressive; simultaneously, it gives motivation for constructing GNNs that do not fit within this framework to construct more expressive GNNs This mapping between certain GNNs with nonstandard message passing and MPNNs acting on graph transformations also gives a potentially simpler framework for analyzing properties of such GNNs, though the authors are saving such study for future work.

**Weaknesses:**

One weakness is that it is not immediately apparent what practical advantages the introduced simulation methods yield over direct implementations of the GNNs being simulated. The authors mention one advantage being that standard libraries for implementing MPNNs can be used once a given GNN architecture has been shown to be simulable via an MPNN acting on a graph transformation, but that is a relatively minor advantage.

**Questions:**

I think this work is very nicely done, and only recommend some further exposition on the potential utility of the authors' work.

**Limitations:**

The authors have adequately discussed the limitations of their work.

---

> ### Author Rebuttal · Authors · 2023-08-09
>
> We thank the reviewer for the very encouraging and positive feedback!
>
> On the reviewers remark:
>
> > I think this work is very nicely done, and only recommend some further exposition on the potential utility of the authors' work.
>
> We will provide further information on the utility of this work in the final version of this paper. Our work has three main utilities: (1) theory of expressivity, (2) design of new GNNs and (3) implementing GNNs with graph transformations.
>
> **(1) theory of expressivity:** Simulation allows to investigate the expressivity of different GNNs through a unified lens by analyzing the corresponding graph transformation. For example, it should be possible to apply the results of Geerts and Reutter (2022) to obtain this result: Any GNN that can be weakly simulated whose graph-to-structure and structure-to-graph encoding can be written in a tensor language with $k+1$ indices is upper bounded in expressivity by $k$-WL. Similarly, it should be possible to apply the results of Morris et al (2023) to obtain VC bounds for any weakly simulated GNN.
>
> **(2) design of new GNNs:** As noted by the reviewer, our theorems indicate that certain constructions (nested aggregations and non-pairwise message passing) **cannot** be strongly simulated and are thus fundamentally different from the message passing paradigm. Thus, to build GNNs that go beyond MPNNs in expressivity it seems promising to investigate such constructions.
>
> **(3) implementing GNNs with graph transformations:** Instead of implementing a GNN with non-standard message passing it can be implemented as a graph transformation together with an MPNN. We see the main advantage of this method as it being easier to implement a graph transformation than non-standard message passing and that the resulting method is:
>
> - Framework agnosticism: graphs can be stored independent from the used library for example as plain text.
> - Ease of use: to use a GNN implemented via a graph transformation it is only necessary to apply the graph transformation to the data and use a standard MPNN. Thus, it is not necessary to implement a complex model, incorporate it into an existing codebase and get it to work.
>
> This utility is also relevant to the only weakness raised by the reviewer:
>
> > One weakness is that it is not immediately apparent what practical advantages the introduced simulation methods yield over direct implementations of the GNNs being simulated. The authors mention one advantage being that standard libraries for implementing MPNNs can be used once a given GNN architecture has been shown to be simulatable via an MPNN acting on a graph transformation, but that is a relatively minor advantage.
>
> For many GNNs it is often not straightforward to use the publicly available implementations. Consider the examples of the GNNs we used in this paper: CWN and DSS.
> CWN is compatible with PyTorch 1.7 and CUDA 10.2 [(source)](https://github.com/twitter-research/cwn) which is too old to be compatible with RTX 3080 GPUs as they are based on the Ampere architecture [(source)](https://www.nvidia.com/en-us/geforce/graphics-cards/30-series/) which requires CUDA 11 [(source)](https://www.nvidia.com/en-us/geforce/graphics-cards/30-series/) (and trying to get it to work with newer versions failed for us, we actually had to find old hardware to run the model).
> DSS is implemented for PyTorch Geometric 1.7.2 which by now is almost 2 years out of date. In version 2.0.2 of PyTorch Geometric the behavior of  attributes with `batch` in the name in PytorchGeometric changes  (line 420 [here](https://github.com/pyg-team/pytorch_geometric/blob/2.0.2/torch_geometric/data/data.py)) compared to 1.7.2 (line 190 [here](https://github.com/pyg-team/pytorch_geometric/blob/1.7.2/torch_geometric/data/data.py#L178)) which causes issues with the `subgraph_batch` attribute in the implementation of DSS. This means that DSS requires a fix to work for newer versions of PyTorch. This shows that it is non-trivial to use existing implementations of GNNs with non-standard message passing. We believe that having a simplified way of running many GNNs would move the field forward by making it easier to use existing GNNs.
>
> #
>
> (Geerts and Reutter 2022): Floris Geerts and JL. Reutter; Expressiveness and Approximation Properties of Graph Neural Networks; ICLR, 2022
>
> (Morris et al 2023): Morris Christopher, Martin Grohe, Jan Martin Tönshoff, and Floris Geerts; WL meets VC; ICML, 2023

---

> > ### Comment · Reviewer_r6Zx · 2023-08-10
> >
> > Thanks for the reply. Including the above in the final version of the paper would address my concern.

---

### Official Review · Reviewer_GmCj · 2023-07-12

**Soundness:** 3 good
**Presentation:** 3 good
**Contribution:** 3 good
**Rating:** 7
**Confidence:** 2

**Summary:**

The paper formally introduces the notions of MPNN simulating GNN with graph transformation. With the definitions, the authors further investigate which class of GNN can be simulated by MPNN.

**Strengths:**

The work presents the first systematic theoretical investigation toward understanding which GNN can be simulated by MPNN.

**Weaknesses:**

The simulation is based on expressiveness equivalence, so more empirical investigations may be required to understand the equivalence in practice.

**Questions:**

It would be nice to see more empirical investigations under the notions of simulations.

**Limitations:**

Limitations have been adequately addressed.

---

> ### Author Rebuttal · Authors · 2023-08-09
>
> We thank the reviewer for their positive feedback and appreciate that the reviewer acknowledges the `first systematic theoretical investigation toward understanding which GNN can be simulated by MPNN'. Concerning the weakness and question:
> > (Weakness) The simulation is based on expressiveness equivalence, so more empirical investigations may be required to understand the equivalence in practice.
>
> > (Question) It would be nice to see more empirical investigations under the notions of simulations.
>
> Our experiments indicate that simulation achieves similar performance as the simulated algorithms. We compare simulation via MPNNs against 3 different non-standard message passing GNNs on 11 different datasets (see Tables 1, 2, 4, 5).
>
> We agree that additional empirical investigations would further our understanding of GNNs and simulations in practice. While this is highly interesting, a fair comparison with a large number of higher-order GNNs requires that all higher-order GNNs are evaluated with a fair selection of hyper-parameters (and bounded run-time). This goes well beyond the scope of the current paper (we are not aware of any such study) and is an avenue for future work. Note that for transformations+simulation only, the fair comparison will be much easier as the same MPNN implementation and structure can be used.

---

### Official Review · Reviewer_5K9U · 2023-07-12

**Soundness:** 3 good
**Presentation:** 2 fair
**Contribution:** 2 fair
**Rating:** 6
**Confidence:** 4

**Summary:**

This work presents a simulation theory/method for efficiently approximating non-standard GNN functions via standard MPNNs plus graph transformers. It starts from the cases that can be strongly simulated and extends to weak simulation for a comprehensive conclusion. A simulation algorithm is also proposed and verified with experimental results.



**Strengths:**

   1. The problem is new (especially focused on non-standard GNNs), clearly defined and rigorously investigated. The motivation is quite practical (if I understand correctly).
   2. Experiments on graph classification tasks verify the effectiveness of the proposed algorithms, including performance on AUC, time cost, RMSE, etc.

**Weaknesses:**

   1. The structure of the paragraphs dramatically reduces the readability.
   2. See the questions below.

**Questions:**

A. Major

      1. Lines 21-22 should give more examples of 'non-standard' message passing and 'standard' MPNNs for better readability at the beginning. Better to use a table.
      2. As I understand it, the core of the work is to transfer these 'non-standard' MP functions to MPNNs plus graph transformation functions. Why can these graph transformation functions not also be understood as MPNNs? If this is not the case, is this the essential contribution, rather than just giving them a different name?
      3. It seems that strongly simulated GNNs are equivalent to higher-order WL architectures. For example, subgraph-based GNNs.
      4. For the experiments, the most important aspect is the time cost, while Table 6 mixes the convergence epochs and the computational complexity. More statistics on FLOPs and convergence analysis are recommended.


B. Minor

      1. Is the expressiveness of MPNNs in this paper about the 1-WL test? Or the authors should provide different explanations for the word "expressivity" each time it is used. For example, in line 503 of the appendix,  should it read 1-WL test？
      2. Some errors in citation notation, e.g., lines 42-45.
      3. In line 94, the definition of the neighborhood function, should it be $\mathcal{N}_l$? Since $l$ is a parameter of the function, or two functions with different settings of $l(\mathcal{N})$ can never know. There may be some misunderstanding of line 95.

**Limitations:**

N/A，

---

> ### Author Rebuttal · Authors · 2023-08-09
>
> We thank the reviewer for the positive review and appreciate that the reviewer acknowledges our rigorous investigation of a practically motivated novel problem.
>
> The reviewer mentions that the "structure of paragraphs" reduces readability. Could you give us more details so we can fix it? Do you mean the inline equations and/or rather long paragraphs? We will improve readability in the camera-ready version using the extra page.
>
> ### Concerning the Questions
> **Major**
>
> >   1. Lines 21-22 should give more examples of 'non-standard' message passing and 'standard' MPNNs for better readability at the beginning. Better to use a table.
>
> We will add more examples to this section. With standard message passing we refer to MPNNs - the most common type of GNNs, which aggregate messages over each node's neighborhood and expressivity bounded by $1$-WL (Morris et al., 2019, Xu et al., 2019). With non-standard message passing we mean any other GNN, typically with expressivity surpassing $1$-WL.
>
> >   2. As I understand it, the core of the work is to transfer these 'non-standard' MP functions to MPNNs plus graph transformation functions. Why can these graph transformation functions not also be understood as MPNNs? If this is not the case, is this the essential contribution, rather than just giving them a different name?
>
> On the one hand, graph transformations take graphs as input and output a modified graph. We use them as pre-processing. On the other hand MPNNs are differentiable functions that take graphs as input and output either node or graph embeddings (in Euclidean space). Furthermore, graph transformations cannot be seen as MPNNs: graph transformations can perform operations that are fundamentally impossible to do for MPNNs alone. For example, the transformation in Figure 1 transforms induced cycles in the graph to specially labeled vertices. As it is known (Chen et al., 2020) that MPNNs cannot detect induced cycles it follows that this transformation cannot be based on an MPNN.
>
> >  3. It seems that strongly simulated GNNs are equivalent to higher-order WL architectures. For example, subgraph-based GNNs.
>
> It depends on what we regard as "higher-order": $k$-WL specifically or more generally some variants of WL. Strongly simulatable GNNs do not necessarily correspond (as far as we know) to some $k$-WL test and might have expressivity incomparable to it. However, any strongly or weakly simulatable GNN can be loosely seen as a variant of WL, potentially increasing expressivity. In particular, replacing the final "function application (S4)" step in augmented passing with any injective function (i.e., the identity function) will result in a color update function, which can be viewed as a variant of WL. Hence, our work makes WL / MPNNs at least as expressive as any strongly simulatable GNN through graph transformations.
>
> >   4. For the experiments, the most important aspect is the time cost, while Table 6 mixes the convergence epochs and the computational complexity. More statistics on FLOPs and convergence analysis are recommended.
>
> Fair empirical convergence analysis in realistic settings is difficult as the different architectures often use different hyperparameters and performance dependence strongly on them. An extensive empirical comparison of time vs performance over different hyperparameters or tuning strategies goes well beyond the scope of this paper (we are not aware of any paper presenting such an analysis).
>
> Having said that, we like the idea of adding more statistics like FLOPS to the final version of the paper and will do so as long as time permits.
>
> **Minor**
>
> >   1. Is the expressiveness of MPNNs in this paper about the 1-WL test? Or the authors should provide different explanations for the word "expressivity" each time it is used. For example, in line 503 of the appendix,  should it read 1-WL test？
>
> When talking about about expressivity (or expressiveness) we mean the ability of a GNN (or any function defined on graphs) to distinguish non-isomorphic graphs i.e. its ability to map different graphs to different embeddings. MPNNs are at most as expressive as the $1$-WL test and there are specific MPNNs (such as GIN) that are exactly as expressive as $1$-WL (Morris et al., 2019, Xu et al., 2019). Thus, when we compare the expressivity of some GNN to an MPNN we also compare it to the $1$-WL test. Line 503 reads: _This lead to the development of new GNNs that have higher expressivity than MPNNs._ Hence, line 503 equivalently also means: _This lead to the development of new GNNs that have higher expressivity than the $1$-WL graph isomorphism test._
>
> > 2. Some errors in citation notation, e.g., lines 42-45.
>
> Thank you, we will fix this.
>
> > 3. In line 94, the definition of the neighborhood function, should it be $\mathcal{N}_\ell$? Since $\ell$ is a parameter of the function, or two functions with different settings of $\ell(\mathcal{N})$ can never know. There may be some misunderstanding of line 95.
>
> We do not fully understand the question. Can you please explain more exactly what you mean? We agree that it might be useful to make the parameter $\ell$ explicit in $\mathcal{N}_\ell$. However, this would clash with with other notation we are already using. We will add further explanations to clarify the definition of neighborhood function.
>
> #
>
> (Morris et al., 2019): Morris et al.; Weisfeiler and Leman Go Neural: Higher-order Graph Neural Networks; AAAI, 2019
>
> (Xu et al., 2019): Xu et al.; How Powerful are Graph Neural Networks?; ICLR, 2019
>
> (Chen et al., 2020): Chen et al.; Can Graph Neural Networks Count Substructures?; NeurIPS, 2020

---

> > ### Comment · Reviewer_5K9U · 2023-08-15
> > **Thank you for your rebuttal and the clarifications.**
> >
> > Some of my concerns were addressed. Here, I would like to clarify some questions:
> >
> > The "structure of paragraphs" reduces readability:
> >
> > --  The paper's organization in the intro and background sections can be optimized. In the intro, we want to understand some background, issues, and the motivation for the paper's work.
> >
> > -- Readers want to see more than just a simple list of works in the related work. Instead, we should present an organized summary of existing work;
> >
> > -- The contribution section should be more concise, clear, and organized.
> >
> > The definition of the neighborhood function： Your understanding is correct, I still believe that $ℓ (\mathcal{N})$ is not a good representation.

---

> > > ### Author Response · Authors · 2023-08-16
> > >
> > > Thank you for the clarifications. We will take them into account for the final version of our paper.

---

### Official Review · Reviewer_YhPG · 2023-07-13

**Soundness:** 4 excellent
**Presentation:** 3 good
**Contribution:** 3 good
**Rating:** 6
**Confidence:** 3

**Summary:**

The paper proposes a novel approach to simulate state-of-the-art graph neural networks (GNNs) using standard message passing. The authors introduce graph transformations that preserve the expressivity of GNNs and allow for better code optimization and competitive predictive performance on various molecular benchmark datasets. The paper presents two types of simulation, weak and strong, and evaluates their performance empirically.

**Strengths:**

- The paper is highly original and presents a novel approach to simulating GNNs.
- The authors provide a thorough evaluation of the proposed method on 10 benchmark datasets.
- The paper is well-written and easy to follow, with clear explanations of the proposed method and its evaluation.
- The proofs in the supplementary are comprehensive. I am impressed by the detailed step-by-step illustrations


**Weaknesses:**

Lack of Clarification.
- The intro is not logically coherent. For example. it lacks a formal definition of “simulation” before expanding on it (e.g. simulation is using graph transformations so that standard message passing can reach comparable performances as state-of-the-art graph neural networks? In the introduction, no formal definition of “simulation” is given before expanding on “strongly/weakly simulated” MP algorithms.
- Likewise, how do you define graph transformations and what are all possible types of graph transformations? Only until Section 3 were formal definition given

There are claims in the introduction that seem not grounded, which require justification and explanation:
- Line 19-20: This is due to their limited expressivity: for all MPNNs, there are 20 pairs of non-isomorphic graphs that get the same embedding


**Questions:**

- How sensitive is the performance of the proposed method to the choice of hyperparameters?

---

> ### Author Rebuttal · Authors · 2023-08-09
>
> We thank the reviewer for the positive review and appreciate that the reviewer acknowledges our paper novel approach, thorough evaluation, and highly original paper which is `well-written and easy to follow'. The reviewer raises only some lack of clarification as weakness which we address below.
>
> ### Concerning the Weaknesses
> > Lack of Clarification.
>
>
> > The intro is not logically coherent. For example. it lacks a formal definition of “simulation” before expanding on it (e.g. simulation is using graph transformations so that standard message passing can reach comparable performances as state-of-the-art graph neural networks? In the introduction, no formal definition of “simulation” is given before expanding on “strongly/weakly simulated” MP algorithms.
>
> >Likewise, how do you define graph transformations and what are all possible types of graph transformations? Only until Section 3 were formal definition given
>
> To address the points raised, we will provide some intuition and give a high-level definition of our central concepts simulation and graph transformation in the introduction.
>
> > There are claims in the introduction that seem not grounded, which require justification and explanation:
>
> > Line 19-20: This is due to their limited expressivity: for all MPNNs, there are pairs of non-isomorphic graphs that get the same embedding
>
> We will clarify in the camera ready version that this follows from the fact the MPNNs are bounded by $1$-WL in their expressivity (Morris et al., 2019, Xu et al., 2019).
>
> ### Concerning the Question
> > How sensitive is the performance of the proposed method to the choice of hyperparameters?
>
> In our experiments our approaches (simulating other GNNs) were similarly sensitive to hyperparameters as the original GNNs. In particular the number of message passing layers and graph pooling operations are important to tune.

---

### Author Rebuttal · Authors · 2023-08-09

Dear reviewers, we thank you very much for your detailed comments and respond to your reviews individually below. We appreciate that you acknowledge the novelty, originality, the systematic theoretical investigation, and the well-conducted experiments.

Dear all, to further extend the generality of our approach we have taken a quick look at some recent papers and could prove that two additional GNNs can be strongly simulated: shortest path networks [Abboud et al., 2022] and generalized distance weisfeiler-leman [Zhang et al., 2023]. We have added a list of utilities that our work provides to the review of r6Zx and look forward to the discussion with you.

Best,

The authors of Expressivity-Preserving GNN Simulation



#


[Abboud et al., 2022]: Abboud et al.; Shortest Path Networks for Graph Property Prediction; Learning on Graphs Conference, 2022

[Zhang et al., 2023]: Zhang et al.; RETHINKING THE EXPRESSIVE POWER OF GNNS VIA
GRAPH BICONNECTIVITY; ICLR, 2023

---

### Comment · Area_Chair_RsCu · 2023-08-15
**Author-reviewer discussion**

Dear all,

The author-reviewer discussion period has now started. It will continue for one more week, until August 21.

@authors: Please respond to the comments or questions reviewers may further have. Remain short and to the point.

@reviewers: Please read the author's responses and ask any further questions you may have. To facilitate the decision by the end of the process, please also acknowledge that you have read the responses and indicate whether you want to update your evaluation.

- You can update your evaluation positively (if you are satisfied with the responses) or negatively (if you are not satisfied with the responses or share other reviewers' concerns). Please note that major changes are a reason for rejection.
- You can also keep your evaluation unchanged. In this case, please indicate that you have read the responses and that you do not have any further comments.

Best regards,
The AC

---

### Decision · Program_Chairs · 2023-09-21

**Decision:**

Accept (poster)

**Comment:**

The reviewers are overall quite positive about the paper (6-6-7-7-8), except for Reviewer sDHK leaning towards rejection (4). The author-reviewer discussion has been constructive and has led a number of recommendations to improve the paper. The novelty of the paper is debated by Reviewer sDHK, who argues the results are mostly implicitly known to the community. Nevertheless, provided that shortcomings are addressed, Reviewer sDHK is also not against acceptance. For these reasons, we request the authors to address as much as possible the reviewers' comments in the final version of the paper.